# DUACS DT2014 : the new multi-mission altimeter dataset reprocessed over 20 years

**M.-I. Pujol[1*], Y. Faugère[1*], G. Taburet[1], S. Dupuy[1], C Pelloquin[1], M. Ablain[1], N. Picot[2]**

[1]{Collecte Localisation Satellites, Toulouse, France}

[2]{Centre National Etudes Spatiales, Toulouse, France}

[*]{M.-I. Pujol and Y. Faugere contributed equally to this paper}

Correspondence to: M.-I. Pujol (mpujol@cls.fr)

**Abstract**

The new DUACS DT2014 reprocessed products have been available since April 2014. Numerous innovative

changes have been introduced at each step of an extensively revised data processing protocol. The use of a new

20-year altimeter reference period in place of the previous 7-year reference significantly changes the SLA

patterns and thus has  a strong user impact. The use of up to date altimeter standards and geophysical corrections,

reduced smoothing of the along-track data, and refined mapping parameters, including spatial and temporal

correlation scale refinement and measurement errors all contribute to an improved high quality DT2014 SLA

dataset. Although all of the DUACS products have been upgraded, this paper focuses on the enhancements to the

gridded SLA products over the global ocean. As part of this exercise, 21 years of data have been homogenized

allowing us to retrieve accurate large scale climate signals such as global and regional MSL trends, interannual

signals, and better refined mesoscale features.

An extensive assessment exercise has been carried out on this dataset, which allows us to establish a

consolidated error budget. The errors at mesoscale are about 1.4cm² in low variability areas, increase to an

average of 8.9cm² in coastal regions, and reach nearly 32.5cm² in high mesoscale activity areas. The DT2014

products, compared to the previous DT2010 version, retain signals for wavelengths lower than ~250 km,

inducing SLA variance and mean EKE increases of respectively +5.1% and +15%. Comparisons with

independent measurements highlight the improved mesoscale representation within this new dataset. The error

reduction at the mesoscale reaches nearly 10% of the error observed with DT2010. DT2014 also presents an

improved coastal signal with a nearly 2 to 4% mean error reduction. High latitude areas are also more accurately

represented in DT2014, with an improved consistency between spatial coverage and sea ice edge position. An

error budget is used to highlight the limitations of the new gridded products, with notable errors in areas with

strong internal tides.

**1.  Introduction**

Since its inception in late 1997, the DUACS (Data Unification and Altimeter Combination System) system has

produced and delivered high quality along track (L3) and multimission gridded (L4) altimeter products that are

used by a large variety of users for different applications. The data are available both in Near Real Time (NRT),

with a delay of a few hours to one day, and in a Delayed Time (DT) mode with a delay of few months. A

complete reprocessing of the DT products is done every four years approximately. Over the last two decades,

successive papers have described the evolution of the DUACS system and its associated products (Le Traon et

al, 1992; 1995; 1998; 1999; 2003; Ducet et al, 2000; Pujol et al, 2005; Dibarboure et al., 2011). The quality of

DUACS products is affected by several factors, such as the altimeter constellation used for input (Pascual et al,

2006; Dibarboure et al, 2011), the choices of altimeter standards (Dibarboure et al, 2011; Ablain et al, 2015), and

improvements in data processing algorithms (Ducet et al, 2000; Dussurget et al, 2011; Griffin et al, 2012;

Escudier et al, 2013).

This paper addresses a new global reprocessing that covers the entire altimeter period and allows us, for the first

time, to generate a gridded time series of more than 20 years, identified here as DT2014. The period starts at the

beginning of the altimeter era and ranges from 1993 to 2013. Measurements from 10 altimeter missions (repeat

track and geodetic orbits) have been used: the TOPEX/Poseidon (TP) and Jason series (Jason-1 (J1) & OSTM/Jason-2 (J2)), ERS-1, ERS-2 and ENVISAT (EN), Geosat Follow On (GFO), Cryosat-2 (C2), Saral/AltiKa (AL) and Haiyang-2A (HY-2A). DT2014 represents a major upgrade of the previous version, DT2010 (Dibarboure et al., 2011), but pursues the same objectives that comprise the generation of time series that are homogeneous in terms of altimeter standards and processing with an optimal content at both mesoscales and large scales. To achieve this objective, various algorithms and corrections developed by the research community and through different projects and programs as the French SALP/Aviso, the European Myocean2, and the European Space Agency (ESA) Climate Change Initiative projects are used. The development of regional experimental DUACS products in the framework of scientific oceanographic campaigns such as KEOPS-2 (d'Ovidio et al, 2015) was also valuable for local assessments of the improvements, prior to the implementation and release of the global product. However, one of the main priorities was to improve the monitoring of the mesoscales in the global ocean. Indeed, recent papers (Dussurget et al, 2011, Chelton et al, 2011, Escudier at al. 2013) have shown that despite the accuracy of the DT2010 gridded products, the interpolation of mesoscale signals is limited by the anisotropy of the altimetry observing system. Finally finer scale signals contained in the altimeter raw measurements are not really exploited and provided in the higher level DUACS products (L3 and L4). In addition to these mesoscale retrieval improvements and to satisfy the needs of different Aviso users, the new DT2014 reprocessing product also benefits from climate standards and corrections that do not degrade the mesoscale signals. Thus, the different choices and trade-offs that have been made in the generation of the DT2014 reprocessing are described in detail in this paper.

The DT2014 reprocessing is characterized by important changes in terms of altimeter standards, data processing and formats. The main changes consist of referencing the SLA products to a new altimeter reference period, taking advantage of the 20 years of measurements that are currently available and optimizing along-track random noise reduction, which affected a large part of the physical signal in the DT2010 version. These changes make a significant impact on the physical content of the SLA and derived products. The gridded SLA products are constructed using more accurate parameters (e.g., correlation scales, error budgets), and computed directly at the 1/4°x1/4° Cartesian grid resolution. Other changes that have been implemented allow us to correct a number of different anomalies that were detected in the previous DT2010 product suite. The resulting quality of the sea surface height estimate is improved. In this paper we introduce DT2014, the latest version of the Aviso SLA product and evaluate its improvements with respect to the previous version.

The paper is organized as follows: details of the L3/L4 altimeter data processing used for the generation of the DT2014 products, is presented in section 2. In section 3, results obtained from the DT2014 SLA reprocessed products are compared with equivalent DT2010 results, focusing on the mesoscales and coastal areas. In the same section, for the first time, we make an estimate of the L4 SLA product errors. Finally, a summary of the key results obtained are given in section 4.

## 2. Data Processing

### 2.1. Altimeter standards

The altimeter standards used for DT2014 were selected taking advantage of the work performed during the first phase of the Sea Level Climate Change Initiative (SL_cci) led by the European Space Agency in 2011-2013. The objective of this project was to generate optimal reprocessed products for climate applications, notably global and regional mean sea level trends. As part of this exercise, a rigorous selection process was put in place. This process, as well as all the selected standards, is described by Ablain et al. (2015). As recommended by the SL_cci project, several major standards were implemented in the DT2014 products, compared to DT2010. The details of the altimeter standards used in the DT2014 products are given in Tab. 1.

One of the most dramatic improvements comes from the use of ERA-interim reanalysis (from the European Centre for Medium-Range Weather Forecasts -ECMWF-; Dee et al., 2011) instead of operational ECMWF fields for the calculation of the dry tropospheric and other dynamical atmospheric corrections. Important improvements have been observed over the first altimetry decade (1993-2003) at the mesoscale and, especially, at high latitudes, allowing a better estimation of long-term regional mean sea level trends (Carrere et al., 2016). However, the evaluations also showed that the use of this correction slightly degraded the variance of the signal in shallow water areas for the second altimetry decade. To ensure an optimal description of these signals for Aviso/Myocean-2 users, the Operational ECMWF fields were used from 2001 onwards

Another major improvement has been achieved by using new orbit solutions for different altimeter missions: REAPER combined orbit solutions (Rudenko et al., 2012) for ERS-1 and ERS-2, CNES GDR-D orbit solutions (Couhert et al., 2015) for the J1, J2 and EN missions. Significant effects were observed on regional sea level trends, in the range 1-2 mm/yr, with large patterns at hemispheric scale when using static and time variable Earth gravity field models for orbit computation. Thanks to cross-comparisons between altimetry missions (Ollivier et al., 2012) and with in-situ measurements (Valladeau et al., 2012), these new orbit solutions have been demonstrated to dramatically improve the regional sea level trends.

In addition to these major improvements, other new altimeter standards were also selected, although their impact on sea level estimates was lower. These mainly concern the radiometer-based corrections that use combined estimates from valid on-board MWR values and Global Navigation Satellite System (GNSS) measurements (Fernandes et al., 2015) and the ionospheric correction with the use of the NIC09 (NOAA Ionosphere Climatology) model for ERS-1 (Scharroo et al., 2010).

## 2.2. Overview of the DUACS DT2014 processing

The DUACS DT processing includes different steps as described by Dibarboure et al (2011). The steps consist of acquisition, homogenization, input data quality control, multi-mission cross-calibration, along-track SLA generation, multi-mission mapping and final quality control. Here we present the DT2014 processing system and evolution compared to the DT2010 version.

### 2.2.1. Acquisition

60+ cumulative years of different datasets were acquired over the 21-year period [1993-2013]. They include measurements from 10 different altimeters: ERS-1 (repeat 35-day and geodetic 168-day period orbits), ERS-2, EN (repeat track and geodetic orbits), TP (historical repeat orbit and new interleaved orbit, i.e. on the midway of

its historical ground tracks), J1 (repeat track orbit, interleaved and geodetic end of life orbit), J2, GFO, C2, AL and HY-2A. The different periods covered by the different altimeters is summarized in Fig. 1. The main differences from DT2010 are the introduction of the year 2011 for C2 and the first cycles of the J1 geodetic orbit (cycle 500 to 505, May to mid June 2012).

### 2.2.2. Input data quality control

The detection of invalid measurements involves various algorithms, from the simplest, such as threshold selection for the different parameters, to more complex (e.g., SLA selection with splines). It was based on the same approach developed for DT2010, detailed in Dibarboure (2011). Details of threshold editing can be found in the handbook of each altimeter mission [e.g. Aviso/SALP 2013; 2015b] as well as Cal/Val reports and publications [e.g. Aviso/SALP (2015a), Ablain et al 2010].

For the DT2014 processing, a specific procedure was established specifically for non-repeat track and the new repeat track orbit missions inducing a more restrictive data selection. As these new missions are able to sample the ocean surface in areas never reached before by older altimeters, their data are usually contaminated by the reduced quality of Mean Sea Surface (MSS) in these specific areas. Such anomalies were observed in the DT2010 along-track SLA fields, and were responsible for the introduction of anomalies to the gridded fields, especially in coastal and high latitude areas. In order to avoid this problem in the DT2014 products, the criteria used for the detection of erroneous measurements along non-repeat tracks and the new repeat tracks was strongly restricted in coastal areas. Indeed, the measurements along the ERS-1 (during geodetic phase), EN geodetic, J1 geodetic, C2, and HY-2A orbits are systematically rejected when closer than 20 km to the coast. In the same way, the poor quality of the MSS in the Laptev Sea leads to systematic rejection of the measurements along non repeat track orbits in this area. The use of a MSS to generate SLA along non repeat track orbits is discussed in Sect. 2.2.4..

### 2.2.3. Multi-mission homogenization and cross-calibration

The first homogenization step consists of acquiring altimeter and ancillary data from the different altimeters that are a priori as homogeneous as possible. The data should include the most recent standards recommended for altimeter products by the different agencies and expert groups such as OSTST, ESA Quality Working groups or ESA SL_cci project. The up to date standards used for DT2014 are described and discussed in Sect.2.1.

Although the raw input L2 GDR datasets are properly homogenized and edited (see Sect. 2.2.2), they are not always coherent due to various sources of geographically correlated errors (instrumental, processing, orbit residuals errors). Consequently, the multi-mission cross-calibration algorithm aims to reduce these errors in order to generate a global, consistent and accurate dataset for all altimeter constellations.

The second homogenization step, crucial for climate signals, consists of ensuring mean sea level continuity between the three altimeter reference missions. The DUACS DT system uses, first, TP from 1993 to April 2002, then J1 until October 2008 and, finally, J2 that covers the end of the period. This processing step consists of reducing the global and regional biases for each transition (TP-J1 and J1-J2), using the tandem phase of the J1 and J2 altimeters where the altimeters follow the same orbit with a few minutes phase offset. The methodology is described in ESA SL_cci (2015). After removing the mean global bias observed, the regional biases are

estimated in two steps. First a polynomial along-track adjustment allows reduction of the latitude dependent biases between the two successive reference missions. A second adjustment consists of reducing regional long wavelength residual biases. As illustrated in Fig. 2, this adjustment permits removal of large spatial pattern (basin scale) errors of the order of 1-2 cm.

Next, a cross-calibration process consists of reducing orbit errors through a global minimization of the crossover differences observed for the reference mission, and between the reference and other missions also identified as complementary and opportunity missions (i.e. TP after April 2002, J1 after Oct. 2008, ERS-1, ERS-2, EN, GFO, AL, C2 and HY-2A). The methodology, used also for DT2010 dataset is described by Le Traon and Ogor (1998).

The last step consists of applying the long wavelength error reduction algorithm. This process reduces geographically-correlated errors between neighboring tracks from different sensors. This optimal-interpolation based empirical correction (Appendix B) also contributes to reduction of the residual high frequency signal that is not fully corrected by the different corrections that are applied (mainly the Dynamic Atmospheric Correction and Ocean tides). This empirical processing requires an accurate description of the variability of the error signal associated with the different altimeter missions. The variance of the correlated long-wavelength errors used in the DT2014 processing is described in Sect. 2.2.6.

### 2.2.4. Along-track (L3) SLA generation

In order to take advantage of the repeat characteristics of some altimeter missions, and to facilitate use of altimeter products by the users, the measurements are co-located onto theoretical positions, allowing us to estimate a precise Mean Sea Surface (MSS) along these tracks, also referred to as the Mean Profile (MP). The MPs are time averages of the co-located Sea Surface Height (SSH) measured by the altimeters with repeating orbits. The DT2014 reprocessing includes the reprocessing of these MPs along the TP/J1/J2, TP-interleaved/J1-interleaved, ERS-1/ERS-2/EN/AL and GFO tracks. The MPs need to be consistent with the altimeter standards (see Sect.2.1), and the MSS that is used for the non repeat track orbit missions. MP reprocessing includes specific attempts to improve accuracy and extend the estimates into the high latitudes areas. One of the main changes included in the new MPs reprocessing is the use of a new 20-year [1993-2012] altimeter reference period, as more fully explained in Sect. 2.3. Additionally, the precision of the different MPs was improved by combining altimeter data that are on the same orbit. In this way, TP, J1 and J2 measurements are all used to define the corresponding MP; TP interleaved and J1 interleaved or ERS-2 and EN are also merged. ERS-1 measurements were not used in the MP computation. We indeed considered that the temporal period covered by ERS-2 and EN was long enough and allow us to discard the reduced quality ERS-1 35-day repetitive measurements over year 1993. This processing leads to an improved definition of the MPs with, in particular, a gain of defined positions near the coast. The number of points defined within 0-15 km from the coast in the new MPs is twice (three times) the number observed in the previous MP version, along respective TP and TP interleaved theoretical tracks. In the same way, an additional 15 to 20% more points are defined near the coasts along the GFO and EN theoretical tracks in the new MPs. The MP along EN theoretical tracks is also more accurately defined in the high latitude areas, taking advantage of increased ice melt since 2007 (Fig. 3).

In the case of the non repeating missions (i.e., ERS-1 during the geodetic phase; EN after the orbit change; J1 on the geodetic phase; C2), or recent missions following the newest theoretical track (i.e., HY-2A), the estimation of a precise MP is not possible. In this case, the SLA is estimated along the real altimeter tracks, using a gridded MSS as a reference. The latter is the MSS_CNES_CLS_11 described by Schaeffer et al (2012), and corrected in order to be representative of the 20-year [1993-2012] period (see also Sect. 2.3).

The SLA, obtained by subtracting the MP or MSS from the SSH measured by the altimeter, is affected by measurement noise. A Lanczos low pass along-track filtering allows us to reduce this noise. Two different filtering parameterizations are used, according to the application. For the generation of the L3 along-track SLA, the cut-off wavelength was revisited in the DT2014 in order to reduce random measurement noise as much as possible whilst retaining the dynamic signal. More details are given in the following section. For the generation of the L4 gridded SLA, the filtering is also intended to reduce small scale dynamical signals that cannot be accurately retrieved. Details are given in Sect. 2.2.6.

### 2.2.5. Along track (L3) noise filtering

The gridded product processing parameters are a trade-off between the altimeter constellation sampling capability and the signal to be retrieved. For DT2010 the processing and, in particular, the along track noise filtering were set up in accordance with this objective. Consequently, the global DT2010 along-track SLA products were low-pass filtered with a Lanczos cut-off filter with wavelengths depending on latitude (250 km near the equator, down to 60 km at high latitudes). This technical choice was mostly linked to the ability of the TP altimeter mission to capture ocean dynamic mesoscale structures (Le Traon and Dibarboure 1999). However it strongly reduced the along-track resolution that can be useful and beneficial for modeling and forecasting systems. For this reason a dedicated along track product that preserves the along track 1 Hz short wavelengths signals has been developed in the frame of the DT2014 reprocessing. The main inputs come from the study by Dufau et al. (2016).

An SLA power spectrum density analysis was used in order to determine the wavelength where signal and error are of the same order of magnitude. It represents the minimum wavelength associated with the dynamical structures that altimetry would statistically be able to observe with a signal-to noise ratio greater than 1. This wavelength has been found to be variable in space and time (Dufau et al., 2016). The mean value was found to be nearly 65 km. It was defined with a single year of Jason-2 measurements, over the global ocean, excluding latitudes between 20°S and 20°N (due, in part, to the limit of the underlying Surface Quasi-Geostrophic turbulence in these areas). In the end the cut-off length of 65 km in the DT2014 along-track low-pass filtering processing was retained. It is considered as the minimal low-pass cut-off length that can be applied to along-track SLA in order to reduce noise effects and preserve as much as possible the physical signal. This however cannot be defined as a perfect noise removal operation since, in practice, a signal-to-noise ratio of 2 to 10 (cut-off with wavelength of 100-150 km or more) would be required to obtain a noise-free topography.

The filtered along-track products are subsampled before delivery in order to retain every second point along the tracks, leading to a nearly 14 km distance between successive points. Because some applications need the full resolution data, the non-filtered and non-sub-sampled products are also distributed in DT mode.

| | |
|---|---|
| 1 | ## 2.2.6. **Gridded product (L4) generation: multimission mapping** |
| 2 | Before the multi-mission merging into a gridded product, the along-track measurements are also low-pass |
| 3 | filtered in view of the mapping process. In this case, the aim of the filtering is also to reduce the signature of the |
| 4 | short scale signals that cannot be properly retrieved mainly due to limitations of the altimetry spatial and |
| 5 | temporal sampling. Indeed, the altimeter inter-track diamond distances and the revisit time period limit the |
| 6 | observation of mesoscale structures. Previous studies (Le Traon and Dibarboure, 1999; Pascual et al, 2006) |
| 7 | underscore the necessity for a minimum of a 2-satellite constellation for the retrieval of mesoscale signals. Thus, |
| 8 | in view of the mapping process, the along-track SLA are low-pass filtered by applying a cut-off wavelength that |
| 9 | varies with latitude in order to attenuate SLA variability with wavelengths shorter than nearly 200 km near the |
| 10 | equator, and nearly 65 km for latitudes higher than 40°. Finally, a latitude dependent sub-sampling is applied in |
| 11 | order to be commensurate with the filtering. |
| 12 | The objective of the mapping procedure is to construct a SLA field on a regular grid by combining |
| 13 | measurements from different altimeters. The DUACS mapping processing mainly focuses on mesoscale signal |
| 14 | reconstruction. It uses an Optimal Interpolation (OI) processing as described in Appendix B. This methodology |
| 15 | requires a description of the observation errors and of the characteristics of the physical signal that we want to |
| 16 | map. The parameters used for the mapping procedure are a compromise between the characteristics of the |
| 17 | physical field we focus on, and the sampling capabilities associated with the altimeter constellation. The |
| 18 | parameters used in the DT2014 OI processing were optimised. |
| 19 | An important improvement implemented in DT2014 is the use of more accurately defined correlation scales for |
| 20 | the signal we want to map, and a more precise estimation of the error budgets associated with the different |
| 21 | altimeter measurements. These two parameters indeed have a direct impact on mapping improvements as |
| 22 | underscored by previous studies (Fieguth et al, 1998; Ducet et al 2000; Leben et al 2002; Griffin et al, 2012, |
| 23 | among others). The spatial variability of the spatial and temporal scales of the signal (see Dibarboure et al., |
| 24 | 2011) is better accounted for. Both the spatial and temporal scales are defined as functions of latitude and |
| 25 | longitude. The spatial correlation scales however stay mainly dependent on latitude. Evolution of the zonal and |
| 26 | temporal correlation scales with latitude is given in Fig. 4. The zonal (meridional) correlation scales range |
| 27 | between 80 (80) km and slightly more than ~400 (300) km. The larger values are observed in the low latitude |
| 28 | band (±15°N) where they are mainly representative of the equatorial wave signature. A global reduction of the |
| 29 | correlation scales is observed in the poleward direction. At mid-latitudes (between 20° and 40°), the typical |
| 30 | values observed range between 100(100) km and 200(150) km for zonal (meriodional) scales. Poleward of 60°, |
| 31 | local increases up to 200 km of the correlation can be observed. Temporal scales are more dependent on both |
| 32 | longitude and latitude position. Shorter temporal scales are fixed at 10 days. The longer scales are observed at |
| 33 | mid latitudes (20 to 60°) where maximum observed values range between 30 and 45 days. Propagation speeds |
| 34 | are also taken into account. They are mainly westward oriented with extreme values ranging to nearly 30 cm s$^{-1}$ |
| 35 | for latitudes around 5° to a few cm s$^{-1}$ at high latitudes. Eastward propagations of a few cm s$^{-1}$ are also observed |
| 36 | close to the equator and in the circumpolar jet. |
| 37 | Observation errors are defined with an uncorrelated component and an along-track long-wavelength correlated |
| 38 | component (see Appendix B). The variance of the uncorrelated errors is defined assuming an 1 Hz initial |

measurement noise of nearly 3 cm for TP, J1, J2 and AL. Nearly 4 cm is used for the other altimeters. The effect of the filtering and sub-sampling that is applied to the measurement is taken into account and modulates the initial noise estimation. In addition to this noise effect, nearly 15% of the signal variance is used to take account of small scale variability, which cannot be retrieved (see discussion in Le Traon et al., 2001). Additional errors induced by the geodetic characteristics of some orbits (and also the use of a gridded MSS, rather than a more precise MP, as explained in Sect. 2.2.4) are taken into account. In the same way, additional variance is included in the altimeter error budget for which the absence of dual-frequency and/or radiometer measurements leads to the necessity for a model correction for the ionospheric and wet-troposphere signal corrections. The variance associated with along-track long-wavelength correlated errors corresponds to the residual orbit errors, as well as tidal and dynamic atmospheric signal correction errors. In the DT2014 products, the long-wavelength residual ionosphere signal, that can be observed when this correction is obtained from a model (typically for missions with mono-frequency measurements), is taken into account for ERS-2, C2 and HY-2A. In the same way, geodetic missions, for which no precise mean profile is available (see Sect. 2.2.4), present additional long-wavelength errors induced by the use of a global gridded MSS for the SLA computation. These additional MSS errors are taken into account in the reprocessed products for C2, J1 geodetic phase, EN on it geodetic orbit and HY-2A. In the end, the variance of long-wavelength errors represents between 1 to 2% of the signal variance in high variability areas (e.g., the Gulf Stream, Kuroshio, …) and up to 40% in low variability areas and in high ionospheric signal areas for missions without dual-frequency measurement.

Other important changes of the mapping process consist of computing the maps with a daily sampling (i.e., a map is computed for each day of the week, while only maps centered on Wednesdays were computed for DT2010). The reader should, however, note that the time scales of the variability that is resolved in the DT2014 dataset are not substantially different from DT2010; these time scales are imposed by the temporal correlation function used in the OI mapping procedure. A second important change is the definition of the grid points with a global Cartesian 1/4°x1/4° resolution. This choice was mainly driven by user requests since Cartesian grid manipulation is simpler than working on a Mercator projection. The effects of this change are discussed in Appendix C. Note, however that the grid resolution does not correspond to the spatial scales of the features that are resolved by the DT2014 SLA field. These spatial scales are about the same (perhaps slightly smaller) than in the DT2010 fields; they are imposed by the spatial correlation function used in the OI mapping procedure. In addition to the grid standards change, the area defined by the global product was extended towards the poles in order to take into account the high latitude sampling offered by the more recent altimeters such as C2 (i.e. up to ±88°N).

As previously stated, two gridded SLA products are computed, using two different altimeter constellations. The all-sat-merged products take advantage of all the altimeter measurements available. This allows an improved signal sampling when more than 2 altimeters are available (Fig. 1). The mesoscale signal is indeed more accurately reconstructed during these periods (Pascual and al, 2006), when omission errors are reduced by the altimeter sampling. In the same way, high latitude areas can be better sampled by at least one of the available altimeters. These products are however not homogeneous in time, leading to interannual variability of the signal that is directly linked to the evolution of the altimeter sampling. Pascual and al (2006) indeed observed SLA rms differences between 5 and 10 cm when comparing the 2 and the 4 altimeters configurations in high variability

areas. In order to avoid this phenomenon, two-sat merged products are also made available. These are a merging of data from two altimeters following the TP and ERS-2 tracks (e.g., TP, then J1 then J2 merged with ERS-1 then ERS-2 then EN then AL (or C2 when neither EN nor AL are available)) in order to preserve, as much as possible, the temporal homogeneity of the products. Excepting for the differences in altimeter constellations, the mapping parameters are the same for the all-sat-merged and two-sat-merged products.

### 2.2.7. Derived product generation

Derived products are also disseminated to the users. These consist of the Absolute Dynamic Topography (ADT) (maps and along-track) and maps of geostrophic currents (absolute and anomalies).

The ADT products are obtained by adding a Mean Dynamic Topography (MDT) to the SLA field. The MDT used in the DT2014 reprocessing is the MDT CNES/CLS 2013 (Mulet et al, 2013), corrected to be consistent with the 20-year reference period used for the SLA.

The geostrophic current products disseminated to users are computed using a 9-point stencil width methodology (Arbic et al, 2012) for latitudes outside the ±5°N band. Compared with the historical centered difference methodology, the stencil width methodology allows us to correct the anisotropy inherent to the Cartesian projection. It also leads to slightly higher current intensities. In the equatorial band, the Lagerloef methodology (Lagerloef et al, 1999) introducing the β plane approximation is used, with various improvements compared to the previous DT2010 version. Indeed, the meridional velocities are introduced into the β component. Moreover, filtering of the β component is reduced, leading to more intense currents and improving the continuity of the currents within the latitudes ±5°N. The reader should however note that this paper is focused towards a quality description of the SLA products. With this objective, the geostrophic currents used for different diagnostics presented within this paper are obtained using the same methodology (centered differences) for DT2014 and DT2010 datasets.

### 2.2.8. Product format and nomenclature

The DT2014 SLA products and derived products are distributed in NetCDF-3CF format convention with a new nomenclature for file and directory naming. Details are given in the user handbook (Aviso/DUACS, 2014).

### 2.3. Reference period and SLA reference convention

Due to incomplete knowledge of the geoid at small scales and to ease the use of the altimeter DUACS products, the altimeter measurements are co-located onto theoretical tracks and a time average is removed (Dibarboure et al 2011, Sect 2.2.4). Consequently, the sea level anomalies provided in the L3 and L4 DUACS products are representative of variations of the sea level relative to the given period, called the altimeter reference period. Since 2001, the SLA have been referenced to a 7-year period [1993-1999]. In 2014, with more than 20 years of altimeter measurements available, it was of high interest to extend the altimeter reference period to 20 years [1993-2012].

Changing from a 7 to 20 year reference period leads to more realistic oceanic anomalies, in particular at interannual and climate scales. Indeed, the change of reference period from 7 to 20 years not only integrates the evolution of the sea level in terms of trends, but also in terms of interannual signals at small and large scales

(e.g., El Niño/La Niña) over the 13 last years. Fig. 5 (b) shows an example of this impact on a specific track from J2 over the Kuroshio region. It clearly underscores the different SLA signature of the amplitude of the current. The reference period change from 7 to 20 years induces global and regional Mean Sea Level (MSL) variations, as plotted in Fig. 5 (a). It also includes the adjustment of the SLA bias convention. The latter consists of having a mean SLA null over the year 1993. The use of this convention for the SLA leads to the introduction of an SLA bias between the DT2014 products and the former version. In Delayed time, this bias is estimated to be nearly 0.6 cm. The Fig. 5 (a) represents the change that users will observe in the DT2014 version of the product compared to DT2010

The altimeter reference period change also impacts the Mean Dynamic Topography (MDT) field. Indeed, as long as the MDT is combined with the SLA in order to estimate the Absolute Dynamic Topography (ADT), the reference period the MDT refers to must be coherent with the reference period that the SLA refers to. The latest MDT_CNES/CLS 2013 (Mulet et al., 2013) available from Aviso is based on a 20-year reference period, consistent with the DT2014 SLA products.

The Appendix A gives an overview of the relationship between SLA and MDT over different reference periods.

## 3. DT2014 products analysis

### 3.1. Mesoscale signals in the along-track (L3) products

The unique cut-off length of 65 km used for the along-track product low-pass filtering (see Sect. 2.2.5) drastically changes the content of the SLA profiles, especially in low latitudes areas where wavelengths from nearly 250 km (near the equator) to 120 km (near ±30°N) were filtered in the DT2010 products. Higher resolution SLA profiles are now provided.

Spectral analysis applied to the new products confirms the addition of energy in the mesoscale dynamics band at low latitudes: The new along-track SLA preserves the energy of the unfiltered data for length scales greater than 80 km in the equatorial band, and also in the mid latitudes high variability areas although the impact of the filtering change is less. Fig. 6 shows the variance of the short wavelength signal removed (by low-pass filtering) from J2 along-track products over year 2012, both for DT2010 and DT2014. The figure shows a large variance in the mid latitudes areas and equatorial regions. The variance is directly linked to the 1 Hz altimeter measurement error that is highly respectively correlated with the significant wave height and inhomogeneities within the altimeter footprint induced for instance by surface roughness changes or rain cells (Dibarboure et al, 2014; Dufau et al, 2016). In the DT2010 dataset, the filtered wavelength signal is clearly more important in the latitudes ranging between ±40°N, underlining part of the physical signal that is also reduced by the filtering applied.

### 3.2. Mesoscale signals in the gridded (L4) products

#### 3.2.1. DT2010 and DT2014 gridded product intercomparison methodology

In order to be compared with DT2014, the DT2010 products were first processed in order to ensure consistency in resolution and physical content. In this way,

- The DT2010 products considered correspond to the ¼°x1/4° Cartesian resolution products previously identified as "QD" products. These products were obtained from the native DT2010 grid layout (1/3°x1/3° Mercator grid, see Sect. 2.2.6) using bilinear interpolation.
- The DT2010 SLA was referenced to the 20-year altimeter reference period (see Sect. 2.3).

The DT2014 and DT2010 SLA gridded products were compared over their common period [1993, 2012].

### 3.2.2. Additional signal observed in DT2014 compared to DT2010

The mapping process optimization (see Sect. 2.2.6) directly affects the SLA physical content observed within the gridded products. The differences between DT2014 and DT2010 temporal variability of the signal for the period [1993-2012] are shown in Fig. 7. The figure shows additional variability in the DT2014 products. The global mean SLA variance is now increased by nearly +3.5 cm² within the latitude band ±60°N. This represents 5.1% of the variance of the DT2010 "QD" products. This increase is mainly due to the mapping parameters including two main changes in the DT2014 products. The first one, that explains +3.6% of the variance increase, is the change of the native grid resolution. DT2014 was computed directly on the 1/4°x1/4° Cartesian grid resolution (see Sect. 2.2.6), while the DT2010 "QD" product was interpolated linearly from the native 1/3°x1/3° Mercator resolution product (see Sect. 3.2.1). This interpolation process slightly smoothes the signal, and directly contributes to reduction of the variance of the signal observed in DT2010. The second change implemented in the DT2014 products is the use of improved correlation scales, associated with the change of the along-track low-pass filtering presented in Sect. 2.2.6). This change contributes to an increase in the SLA variance of +1.5%. Finally, additional measurements (e.g., C2 in 2011) that were not included in the DT2010 products also contribute to improvements in the signal sampling, and thus increase the variance of the gridded signal.

The additional signal observed in the DT2014 products is not uniformly distributed as shown in Fig. 7. Indeed, the main part of the variance increase (from +50 to more than +100 cm²) is observed in the higher variability areas and coastal areas. It is an expression of the more accurate reconstruction of the mesoscale signal in the DT2014 products, as discussed below. In some parts of the ocean we however observe a decrease of the SLA variance. The improved standards used (see Sect. 2.1) indeed contribute to local reductions of the SLA error variance. The main reduction is observed in the Indonesian area with amplitudes ranging 2 to 3 cm². The SLA error variance is also reduced in the Antarctic area (latitudes < 60°) with the higher local amplitudes. The improved DAC correction using ERA-Interim reanalysis fields over the first decade of the altimeter period is a significant contributor to the variance reduction (Carrere et al., 2015).

Analysis of the spectral content of the gridded products over the Gulf Stream area (Fig. 8) shows that all of the DT2014 products are impacted at small scales, i.e., wavelengths lower than 250-200 km. For "all-sat-merged" as well as "two-sat-merged" products the energy observed in DT2014 for wavelengths around 100 km is twice as high as that observed in the DT2010 gridded SLA products, both in the zonal and meridional directions. The maximum additional signal is observed for wavelengths ranging between 80-100 km. For these wavelengths, the DT2014 products have 2 to 4 times more energy than the DT2010 versions. Nevertheless, the energy associated with these wavelengths falls drastically for both DT2014 and DT2010 SLA products, meaning that DT2014 still misses a large part of the dynamic signal at these wavelengths, as discussed in Sect. 4.

Compared to the DT2010 products, the new DT2014 version has more intense geostrophic currents. This has a direct signature on the eddy kinetic energy (EKE) that can be estimated from the two different versions of the product. Fig. 9 shows the spatial differences of the mean EKE computed from the DT2014 and DT2010 products. As previously observed with the SLA variance, the EKE is higher in the DT2014 products. An additional 400 cm²/s² in levels of EKE are observed in the DT2014 products in high variability areas. This represents a 20% EKE increase compared to DT2010. Proportionally, the EKE increase observed in the DT2014 products is quite large in low variability areas and Eastern boundary coastal currents where it reaches up to 80% of the DT2010 EKE signal, as underscored by Capet et al (2014). The global mean EKE increase, excluding the equatorial band and high latitude areas ($> \pm 60°N$), represents nearly 15% of the EKE observed in the DT2010 products. As previously observed with the SLA variance, the change of the native grid resolution and the change of the correlation scales and along-track filtering explain respectively +10% and +6% of the EKE increase. The change of the altimeter standards rather contribute to slightly reduce the EKE in the DT2014.

### 3.2.3. Impact of the altimeter reference period on EKE

Fig. 10 shows the temporal evolution of the mean EKE over the global ocean both for DT2014 and DT2010. We first note the nearly 15% additional mean EKE in the DT2014 product as previously discussed. We also note a significant difference in the EKE trend between DT2014 and DT2010, where the latter is on the 7-year altimeter reference period (Sect. 2.3). Indeed, the mean EKE trend is nearly -0.027 (-0.445) $cm²s^{-2}$/year when DT2010_ref7y (DT2014) products are considered. On the other hand, when DT2010 is referenced to the 20-year period, the EKE trend (-0.369 $cm²s^{-2}$/year) is comparable to the DT2014. This result clearly emphasizes the sensitivity of the EKE trend estimation to the altimeter reference period. Indeed, the use of the 20-year reference period leads to a minimized signature of the SLA signal over this period. Conversely, the SLA gradients are artificially higher after 1999 when the historical [1993-1999] reference period is used. As a consequence, after 1999, the EKE from the DT2010 products (on the 7-year reference period) is higher than the EKE from the DT2014 products (we do not consider here the global mean EKE bias observed between the two products).

### 3.2.4. DT2014 gridded product error estimates at the mesoscale and error reduction compared to DT2010

The accuracy of the gridded SLA field is estimated by comparing SLA maps with independent along-track measurements. Maps produced by merging of only two altimeters (i.e., "two-sat-merged" products; see Sect. 2.2.6) are compared with SLA measured along the tracks from other missions. In this way, TP interleaved is compared with a DT2014 gridded product that merges J1 and EN over the years 2003-2004. The variance of the SLA differences is analyzed for the wavelengths ranging between 65-500 km, characteristics of medium and large mesoscale signals. The same comparison is done using the previous DT2010 version of the products in order to estimate the improved accuracy of the new DT2014 gridded SLA fields. The results of the comparison between gridded and along-track products are shown in Fig. 11 and summarized in Tab. 2.

The gridded product errors for mesoscale wavelengths usually range between 4.9 (low variability areas) and 32.5 cm² (high variability areas) when excluding coastal and high latitude areas. They can, however, be lower, especially over very low variability areas such as the South Atlantic Subtropical gyre (hereafter "reference area") where the observed errors nearly 1.4 cm². It is important to note that these results are representative of the

quality of the "two-sat-merged" gridded products. These can be considered to be degraded products for mesoscale mapping since they use minimal altimeter sampling. On the other hand the "all-sat-merged" products, during the periods when three or four altimeters were available, benefit from improved surface sampling. The errors in these products should thus be lower than those observed in the products that merge only two altimeters.

Compared to the previous version of the products, the gridded SLA errors are reduced. Far from the coast, and for ocean variances lower than 200 cm², the processing/parameter changes included in the DT2014 version lead to a reduction of 2.1% of the variance of the differences between gridded products and along-track measurements observed with DT2010. The reduction is higher when considering high variability areas (> 200 cm²), where the impact of the new DT2014 processing is maximum. In this case, it reaches 9.9%. On the other hand, some slight degradation is observed in tropical areas, especially in the Indian Ocean. In that region, up to 1 cm² increased variance of the differences between grids and along-track estimates is observed. This can be directly linked to the change of the processing in these latitudes, especially the reduction of the short wavelength filtering applied before the mapping process, as explained in Sect. 2.2.6.

### 3.3. geostrophic current quality

The improved mesoscale mapping also affects the quality of the geostrophic current estimation, which is directly linked to SLA gradients. Geostrophic currents computed from ADT altimeter gridded products were compared with geostrophic currents measured by drifters. Surface drifters distributed by the AOML (Atlantic Oceanographic & Meteorological Laboratory) over the period 1993-2011 were processed in order to extract the absolute geostrophic component only. In this way, they were corrected for the Ekman component using the model described by Rio et al. (2011). Drifter drogue loss was detected and corrected using the methodology described by Rio et al. (2012). A low-pass 3-day filter is applied in order to reduce inertial wave effects. Finally, the absolute geostrophic currents deduced from altimeter "all-sat-merged" SLA grids using centered differences methodology are interpolated to the drifter positions for comparison.

The distribution of the speed of the current (not shown), shows a global underestimation of the current in the altimeter products compared to the drifter observations, especially for currents with medium and strong intensities (> 0.2 m/s). However, in both cases, the DT2014 current speeds are still closer to the drifter distribution. The rms of the zonal and meridional components of the currents are also increased in the DT2014 dataset and hence closer to the observations. Taylor skill scores (Taylor, 2001), that take into account both correlation and rms of the signal, are given in Tab. 3. Outside the equatorial band, the Taylor score is 0.83 (0.83) for the zonal (meridional) component. Compared to the DT2010 products, this is an increase of 0.01 (0.02).

Variance reduction of the differences between altimetry and drifter zonal and meridional components is shown in Fig. 12. Collocated comparisons of zonal and meridional components show that this improvement is not consistent in space, and that errors in the position and shape of the structures mapped by altimeter measurements are still observed in the DT2014 products. Outside the equatorial regions (±15°N), the variance reduction observed with the DT2014 product is nearly -2.1 (-1.2) cm²/s², i.e., -0.55 (-0.34)% of the drifter variance for the zonal (meridional) component. Locally, this reduction can reach more than -10%. Such is the case, for instance, in the Gulf of Mexico and tropical Atlantic Ocean. In contrast, local increase of the variance of the differences between altimetry and drifter measurement (ranging from 2 to 15% of the drifter variance) is observed within the

tropics. This increase is especially significant in the Pacific (zonal component), North Indian Ocean, and north of Madagascar. These areas correspond quite well with regions with high amplitudes of the M2 internal tide that are still present in the altimeter measurements and affect the non-tidal signal at wavelengths near 140 km (Dufau et al, 2016). The increase of the variance of the differences between altimetry and drifter measurement seems to underscore a noise-like signal in the SLA gridded products. This could correspond to the signature of the internal tidal signal, which is more prominent in the DT2014 gridded products as shown by Ray et al (2015). This is certainly reinforced by reduced filtering and the smaller temporal/spatial correlation scales used in this version (Sect. 2.2.6).

### 3.4. Coastal areas and high latitudes

As described in Sect. 2.2.6, processing in coastal regions has also been improved. The most visible change is the increased spatial coverage of the grid in coastal areas. The DT2014 grid more closely approximates the coastline, as illustrated in Fig. 13 (c, d). This is achieved both by tuning of the grid definition near the coast, and by the improved definition of the MPs close to the coast (see Sect. 2.2.4) that allow improved data availability in these nearshore areas.

Spatial grid coverage is also greatly improved in the Arctic region, as illustrated in Fig. 13 (a, b). As above, the tuning of the SLA mapping parameters and availability of MPs in this region directly contribute to this result. Additionally, the reduced errors that contribute to reduction of the SLA variance as shown in Fig. 7, are also a result of a more finely tuned data selection process and the more precise MPs (along ERS-1, ERS-2 and EN tracks) used in the DT2014 product (see Sect. 2.2.1 and 2.2.4). The SLA variance reduction is significant in the Laptev Sea, where it reaches up to 100 cm²

The quality of the gridded SLA products near the coast (0-200 km) was estimated by comparison with independent along-track measurements as explained in Sect. 3.2.4. Results are shown in Fig. 11 and Tab. 2. The mean error variance reaches 8.9 cm². It can be larger in areas of high coastal variability, where up to more than 30 cm² can be observed (Indonesian/Philippine coasts, Eastern Australian coasts, North Sea coasts and coasts located in proximity to the western boundary currents). The DT2014 processing resulted in a global reduction of these differences compared to the DT2010 products. They reach 4.1% of the error variance observed in the DT2010 products. However, local degradations are observed, such as along the Philippine coasts.

The comparison between gridded SLA products and monthly mean tide gauge (TG) measurements from the PSMSL (Permanent Service for Mean Sea Level) database also emphasizes a global improvement in the DT2014 products in coastal areas. TG with a long lifetime (> 4 years) were used. The TG data processing is described by Valladeau et al (2012) and Prandi et al. (2015). The Sea Surface Height measured by the TGs is compared to the monthly mean SLA field given by altimeter gridded products merging all the altimeters available (i.e. "all-sat-merged" products). Data collocation is based on a maximum correlation criterion. The variance of the differences between sea level observed with DT2014 gridded altimetric SLA fields and TG measurements is compared with the results obtained using the DT2010 gridded SLA fields. The results (Fig. 14) show a global reduction of the variance of the differences between altimetry and TGs when DT2014 products are used. This reduction is quite clear at the northern coast of the Gulf of Mexico, along the eastern Indian coasts, and along the US coasts (reduction of up to 5 cm², i.e., from 2 and up to 10% of the TG signal). The Western Australian sea

level is also more accurately represented in the DT2014 products (reduction of up to 2.5 cm², i.e., 1 to 2% of the TG signal). In contrast, a local degradation of the comparison between altimetry and TGs is observed in the north Australian and Indonesian area (increase of up to 2 cm², with local values reaching up to 5 cm²) where it represents, however, less than 4% of the TG signal. The local improvements seen via TG results are consistent with the conclusions from other diagnoses, such as the comparisons between SLA grids and independent along-track measurements over the same coastal areas.

### 3.5. Climate scales

Different processing and altimeter standards changes were defined in accordance with the SL_cci project (Sect. 2.1), and thus also have an impact on MSL trend estimation, especially at regional scales.

The Global MSL trend measured with the DT2014 gridded SLA products over the [1993-2012] period is 2.94 mm/year (no glacial isostatic adjustment applied). The comparison between DT2014 and DT2010 products (Fig. 15, b) does not exhibit any statistically relevant differences. Although no impact is detected on the Global MSL trend, differences are observed at inter-annual scales (1-5 years). The main improvement is the ERS-1 calibration during its geodetic phase (i.e., from April 1994 to March 1995). The nearly 3 mm/year differences observed between DT2010 and DT2014 during this period show an improvement in the DT2014 products. Indeed, a nearly 6 mm bias between ERS-1 and TP was observed in the DT2010 product and this was not entirely reduced when merging both of the altimeter measurements. This was corrected in the DT2014 version. Fig. 15 (b) also shows a global 5.5 mm mean bias difference between the mean SLA from DT2014 and DT2010. This bias is directly linked to the global SLA reference convention used in the DT2014 version, as explained in Sect. 2.3.

The regional MSL trend differences between DT2014 and DT2010 (Fig. 15, a) are similar to the differences shown by Philipps et al, (2013a and 2013b) and Ablain et al (2015) between the SL_cci and DT2010 products (see Fig 6 of the paper). As explained by the authors, the change of orbit standards solution mainly explains the east/west dipole differences.

In order to highlight the improved MSL trend estimation between the eastern and western hemispheres with the DT2014 product, the trend computed from the altimeter products was compared to the trend computed from in-situ quality controlled Temperature/Salinity (T/S) profiles from the CORIOLIS Global Data Assembly Center. The T/S profiles processing used in this paper is the same as described by Valladeau et al (2012) and Legeais et al (2016). The Dynamic Height Anomalies (DHA) deduced from T/S profiles (reference depth 900 dbar) are compared to the SLA fields from gridded "all-sat-merged" products. As discussed by Legeais et al (2016), the DHA are representative of the steric effect above the reference depth, while SLA is representative of both barotropic and baroclinic effects affecting the entire water column. In spite of this difference of physical content, the relative comparison between altimeter SLA and in-situ DHA is sufficient to detect differences between two SLA altimeter products. This comparison was done during the [2005-2012] period when a significant number of in-situ measurements are available. One would expect consistent differences between altimeter and in-situ measurements in both hemispheres. This is the case for the DT2014 products for which the MSL trend differences reach nearly 1.56 (1.68) mm/year in the eastern (western) hemisphere. Conversely, an inconsistency can be observed with DT2010 since the MSL trend differences with in-situ measurements are 2.02 (1.05) mm/year, showing the nearly 1 mm MSL trend differences between the hemispheres.

As presented by Ablain et al (2015), the regional MSL trend comparison also show differences at smaller scales. Here again, the change of standards is directly responsible for these differences. The use of the ERA-Interim reanalysis meteorological fields in the DAC solution (see Sect.2.1) mainly affects the regional MSL trend estimation in the southern high latitude areas, with for instance, impacts higher than 1 mm/yr in the South Pacific Ocean below 50°S latitude as underscored by Carrere et al. (2015). The same meteorological forcing used in the wet troposphere correction slightly contributes to the regional improvement of the MSL trend, especially for the first altimetry decade (Legeais et al 2014). A portion of the smallest regional scale differences are also induced by the improved inter-calibration processing in the DT2014 products, that more accurately take account of the regional biases from one reference mission to another (see Sect. 2.2.3).

Some of the improvements implemented in the DT2014 version also impact the interannual signal reconstruction at regional scales. The more accurate estimation of the long wavelength errors associated with the ionospheric signal correction (see Sect. 2.2.6) leads to a reduced signature of these errors in the products, especially during periods of high solar activity. This was the case in 2000, when ERS-2 is available. The later is indeed a mono-frequency altimeter, preventing us from making a precise ionospheric correction. Additional long wavelength errors in the magnetic equator band, induced by the use of a less precise model solution, are taken into account in the DT2014 products. Comparisons of the regional mean SLA from ERS-2 measurements with TP (for which a precise ionospheric correction is available) over the year 2000 (Fig. 16) underscore a residual ionospheric signal that locally reaches 5 mm. The same comparison done with DT2010 products shows that this residual error was almost twice as high than in the DT2014 version with a more than 1 cm local bias between ERS-2 and TP measurements.

## 4. Discussions and Conclusions

More than 20 years of L3 and L4 altimeter SLA products have been entirely reprocessed and delivered as the DT2014 version. This reprocessing takes into account the most up-to-date altimeter standards, and also includes important changes in different parameters/methods involved at each step of the processing. The implemented changes impact the SLA signals at different spatial and temporal scales, from large to mesoscales and from low to high frequencies.

One important change that will have an impact on users is the referencing of the SLA products to a new altimeter reference period, taking advantage of the 20 years of available measurements and leading to a more realistic interannual SLA record. The variability of the SLA, as well as the EKE deduced from SLA gradients is thus changed compared to the DT2010 dataset, especially after 1999. This change is visible in the mean EKE trend over the 20 year period; it was overestimated in DT2010. This result suggests that previous estimates of EKE trends from altimeter products (e.g., Pujol et al, 2005; Hogg et al., 2015) should be reviewed, taking into account the altimeter reference period.

Other changes were implemented in the DT2014 processing. They consist of using up to date altimeter standards and geophysical corrections, reduced smoothing of the along-track data, and refined mapping parameters, including spatial and temporal correlation scale definitions and measurement errors. This paper focuses on the description of the impact of these changes on the SLA gridded fields, through comparisons with independent measurements.

The SLA variability of the DT2014 dataset is more energetic than DT2010. The variance of the SLA is increased by 5.1% in the DT2014 products, implying additional signals for wavelengths lower than ~250 km. A global 15% EKE increase (equatorial band excluded; latitudes poleward 60° excluded) is also observed with DT2014. This increase is higher in low variability and eastern coastal areas where it reaches up to 80%. The interpolation process that is applied to the DT2010 SLA grids (see Sect. 3.2.1) explains nearly 2/3 of the variability/energy decrease compared to the DT2014 signal. The other 1/3 is directly linked to the improved parameterization of the DT2014 mapping procedure. In contrast to the DT2010 reprocessing (Dibarboure et al, 2011), the effect of the new altimeter standards is moderate in comparison with the effect of the processing changes. The improved accuracy of the along-track signal, that is a result of the use of more accurate altimeter standards (see Sect.2.1), should contribute to a reduction of the SLA error variance observed with gridded products. This was the case when comparing DT2010 with previous DT2007 gridded products (Dibarboure et al, 2011). The DT2010 products did not include significant changes in the mapping processing, and the reduction of the SLA error variance, larger in the Indonesian area, was mainly explained by the use of improved altimeter GDR-C standards. However, the amplitude of this error variance reduction is almost 10 times smaller than the effect of the mapping procedure changes implemented in the DT2014 products.

The additional signal observed in DT2014 is the signature of the improved SLA signal reconstruction, especially at mesoscales, as previously demonstrated by Capet et al (2014) in the eastern boundary upwelling systems. The DT2014 SLA product quality was estimated at global scales using comparisons with independent measurements (altimetry and in-situ) which allowed us to establish a refined mesoscale error budget for the merged gridded products. The DT2014 SLA product errors for the mesoscale signal in the open ocean is estimated to be between 1.4 cm² in low variability areas, and up to 32.5cm² in high variability areas where the altimeter sampling does not allow a full observation of the SLA variability. Compared to the previous version of the products, this error is reduced by a factor up to 9.9% in high variability areas.

Globally, geostrophic currents are slightly intensified in the DT2014 products, becoming closer to the surface drifter observations. The geostrophic current are, however, still underestimated compared to the in-situ observations. Outside the tropical band, the variance of the differences between altimeter products and in-situ observations is reduced almost everywhere. This reduction can reach more than 10% of the in situ variance. In contrast, geostrophic currents estimated with DT2014 products have a lower correlation with in-situ observations within the tropics. This degradation represents up to 15% of the in-situ variance.

DT2014 SLA products were also improved in coastal and high latitude areas. The main improvements are visible in the spatial coverage, refined in coastal areas and improved in Arctic regions with a more precise definition of the coastline and sea ice edge. The SLA gridded product errors in the coastal areas (< 200 km) are estimated at 8.9 cm², with higher values in high variability coastal areas. This error is globally reduced by 4.1% compared to the previous version of the products. Consistency with TG measurements is improved, especially in different areas such as the northern coast of the Gulf of Mexico, along the Indian eastern coasts and along the US coasts. In this case the reduction of variance of the differences between altimetry and TGs ranges between 2 and up to 10 % of the TG signal, when compared to the results obtained with DT2010 products. In some other coastal areas, degradation is however observed. This is the case in the north Australian and Indonesian areas where it reaches less than 4% of the TG signal.

As for the global products, mapping was also improved at regional scale with a positive impact in coastal areas, as presented by Marcos et al (2015) and Juza et al (2016) in the Mediterranean Sea.

Climate scales are also improved with DT2014, taking advantage of the altimeter standards and processing defined in line with the SL_cci project. The global MSL trend estimation is nearly unchanged in the DT2014 products compared to DT2010. However, significant improvements are observed at regional scales, with a reduction of the ±1mm/year dipole error observed in DT2010 between eastern and western hemispheres. Additionally, the residual ionospheric errors, previously observed in altimeter measurements without dual-frequency, are reduced by up to 50% in the DT2014 products.

The assessment of the quality of the DT2014 SLA products at mesoscales underlines the limits of the products.

First, the spectral content of the gridded SLA fields clearly shows that part of the small signal is missing in the gridded products. Although small wavelengths can be resolved with 1 Hz along-track products (up to nearly 80-100km in eastern basins where SLA signal to noise ratios limit observations of smaller wavelengths; satellite and seasonally dependent) (Dufau et al, 2016), the temporal and spatial across-track sampling of the dynamical structures at these wavelength is, however, limited. They are difficult to interpolate onto a 2D grid, especially with a two-altimeter constellation (Pascual et al, 2006, Pujol et al. 2005), and with conventional mapping methods (Escudier et al, 2013; Dussurget et al, 2011). The spatial grid resolutions used for the DT2010 and DT2014 products, as well as the parameters used for map construction (e.g., along-track low pass filtering, correlation scales, measurement errors) are a result of a compromise between the altimeter sampling capability and the physical scales of interest. They are not adapted to resolve the small mesoscales. The resulting mean spatial resolution of the DT2014 global gridded SLA is comparable to the DT2010 resolution. It was estimated to be nearly 1.7°, i.e., slightly less than 200 km at mid latitudes (Chelton et al, 2011, 2014). The comparison with the spectral content computed from full resolution Saral/AltiKa 1Hz along-track measurements (not shown) shows that nearly 60% of the energy observed in along-track measurements at wavelengths ranging from 200-65km is missing in the SLA gridded products. In other words, nearly 3/5 of the small-mesoscale variability is missing in the DT2014 gridded products. This is clearly linked to the mapping methodology, combined with altimeter constellation sampling capability.

The second limitation of the DT2014 gridded SLA fields is the additional non mesoscale signal that is observed. It is characteristic of the residual M2 internal tide, visible in both along-track (Dufau et al, 2016) and gridded products (Ray et al, 2015). The presence of this signal leads to local degradation of DT2014 quality in specifics areas. The signature of internal waves is on the same wavelengths as the mesoscale signal that the DUACS SLA products focus on, making reduction of this signal without affecting the mesoscale signal a non-trivial procedure.

In spite of these limitations, the quality and accuracy of the DUACS products makes them valuable for many applications. They are currently used for derived oceanographic product generation such as ocean indicators (e.g., regional MSL; ENSO; Kuroshio among others; http://www.aviso.altimetry.fr). They are also currently used

for the generation of Lagrangian products, for which the precision of the current can strongly affect the results
(d'Ovidio et al, 2015).

In order to ensure the best consistency and quality, the DUACS DT SLA products will be regularly reprocessed
for all missions, taking advantage of new altimeter standards and improved L3/L4 processing. The next
reprocessed version of the products will be undertaken as part as the new European Copernicus Marine
Environment Monitoring Service (CMEMS) and is expected for release in 2018.

**Appendix A: How to change the reference period**

The gridded SLA products can be referenced to another reference period following Eq. (1), where P and N are
two different reference periods and $\langle SLA \rangle_X$ is the temporal mean of the SLA over the period X. In the same way,
MSS and MDT can be referenced to different reference periods following eq (2) and (3).

$SLA_P = SLA_N - \langle SLA_N \rangle_P$ (1)

$MSS_P = MSS_N + \langle SLA_N \rangle_P$ (2)

$MDT_P = MDT_N + \langle SLA_N \rangle_P$ (3)

By definition, the ADT is independent of the reference period. ADT is obtained combining SLA and MDT
defined over the same reference period (eq. 4)

$ADT = SLA_N + MDT_N = SLA_P + MDT_P$ (4)

**Appendix B: Description of the OI mapping methodology**

The mapping methodology is a global suboptimal space-time objective analysis which takes into account along-
track correlated errors, as described in many previous publications (see for instance Ducet et al., 2000; Le Traon
et al., 2003).

The best least squares linear estimator $\theta_{est}$ and the associated error field $e^2$ are given by Bretherton et al. (1976).

$$\theta_{est} = \sum_{i=1}^{n} \sum_{j=1}^{n} A_{ij}^{-1} C_{xj} \Phi_{obs}$$

$$e^2 = C_{xx} - \sum_{i=1}^{n} \sum_{j=1}^{n} C_{xi} C_{xj} A_{ij}^{-1}$$

24 Where $\Phi_{obs}$ is the observation, i.e., the true SLA $\Phi_i$ and its observation error $\varepsilon_i$. A is the covariance matrix of the
25 observation and C is the covariance between observation and the field to be estimated.

$$A_{ij} = \langle \Phi_{obs} \Phi_{obs} \rangle = \langle \Phi_i \Phi_j \rangle + \langle \varepsilon_i \varepsilon_j \rangle$$

$$C_{xi} = \langle \theta(x) \Phi_{obs} \rangle = \langle \theta(x) \Phi_i \rangle$$

1  The spatial and temporal correlation scales (zero crossing of the correlation function) and propagation velocities

2  characteristic of the signal to be retrieved are defined by the function $C(r,t)$ as in Arhan and Colin de Verdière

3  (1985).

$$C(r,t) = \left[1 + ar + \frac{1}{6}(ar)^2 - \frac{1}{6}(ar)^3\right] e^{-ar} e^{-t^2/T^2}$$

4  Where

$$a = 3.337$$

$$r = \sqrt{\left(\frac{dx - C_{px}dt}{L_x}\right)^2 + \left(\frac{dy - C_{py}dt}{L_y}\right)^2}$$

dx, dy and dt define the distance in space (zonal and meridional directions) and time to the point under

consideration. The spatial and temporal correlation scales are defined as the first zero crossing of C. T is the

temporal correlation radius, $L_x$ and $L_y$ are the spatial correlation radii (zonal and meridional directions), and $C_{px}$

and $C_{py}$ are the propagation velocities (zonal and meridional directions). The values of the different correlation

scales are presented in Sect. 2.2.6

For each grid point where SLA is estimated, the altimeter measurements are selected in a spatial and temporal

subdomain defined as 3 times the prescribed spatial and temporal correlation scales. Measurements located

outside the smaller subdomain, defined by the spatial and temporal correlation scales, are used to correct for long

wavelength errors, enabling us to separate long wavelength errors from the ocean signal. In order to limit the size

of the matrix to be inverted, the SLA measurements are subsampled when located outside the smaller

subdomain. In that case only one point out of four is retained. Additionally, the matrix A is constructed on a

coarse-resolution grid of 1°x1°. The same matrix is used to compute the SLA and associated errors in the

surrounding points located on the ¼°x1/4° grid.

The selected measurements are centered. The removed mean is computed using weights corresponding to the

long wavelength error variance defined along each altimeter track. The removed mean SLA value is then added

back after the analysis.

The observation errors that are considered consist of two components. First, an uncorrelated component is

evaluated. Its variance b² contributes to the $\langle \varepsilon_i \varepsilon_j \rangle$ diagonal matrix. Then, long-wavelength correlated errors are

also considered. In this case, the corresponding variance $E_{LW}$ is added to the non diagonal terms of the

$\langle \varepsilon_i \varepsilon_j \rangle$ matrix, as follows:

$\langle \varepsilon_i \varepsilon_j \rangle = \delta_{i,j} b^2 + E_{LW}$ for points i and j that are on the same track and in the same cycle.

$\delta_{i,j}$ is the Kronecker delta.

The variances b² and $E_{LW}$ are described in Sect. 2.2.6

**Appendix C: Change of the grid spatial resolution between DT2010 and DT2014**

Compared to the historical $1/3°x1/3°$ Mercator native resolution, the Cartesian $¼°x1/4°$ projection leads to a higher grid resolution between latitudes in the band $±41.5°N$, as illustrated in Fig. C1. These latitudes include the bulk of the high variability mesoscale regions, such as the Gulf Stream, Kuroshio, Agulhas current and north of the confluence area. Above these latitudes, the meridional grid resolution is reduced in the Cartesian projection.

As discussed in Sect. 2.2.6, the grid resolution does not correspond to the spatial scales of the features that are resolved by the DT2014 SLA field.

**Acknowledgements**

The DT2014 reprocessing exercise has been supported by the French SALP/CNES project with co-funding from the European MyOcean-2 and MyOcean Follow On projects. The datasets are available from the Aviso website (http://aviso.altimetry.fr/) and the CMEMS web site (http://marine.copernicus.eu/). Level 2 (GDR) input data are provided by CNES, ESA, NASA. The altimeter standards used in DT2014 were selected taking advantage of the work performed during the first phase of the Sea Level Climate Change Initiative (SL_cci) led by ESA in 2011-2013.

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

Table 1: Altimeter standards used in DT2014. Standard changes compared with the DT2010 solution are underlined in bold format.

| | J2 | J1 | TP | ERS-1 | ERS-2 | EN | GFO | C2 | AL | H2 |
|---|---|---|---|---|---|---|---|---|---|---|
| **Product standard ref** | **GDR-D** | **GDR-D** | GDR-C | **OPR** | | **GDRV2.1+** | GDR (NOAA) | CPP CNES | GDR-T patch2 | GDR (NSOAS) |
| **Orbit** | **Cnes POE (GDR_D for cycles ≤253 and GDR-E afterwards)** | **Cnes POE (GDR_D)** | GSFC (ITRF2005, Grace last standards) | **Reaper [Rudenko et al., 2012]** | | **Cnes POE (GDR-D)** | GSFC** (ITRF2005, Grace last standards) | Cnes POE (GDR-D for cycle ≤66 and GDR-E afterwards) | Cnes POE (GDR-D for cycle ≤23 and GDR-E afterwards) | Cnes POE (GDR-D) |
| **Ionopheric** | **dual-frequency altimeter range measurements** | | **dual-frequency altimeter range measurements (Topex), Doris (Poseidon)** | **Reaper (NIC09 model, Scharro and Smith, 2010)** | Bent model (cycle ≤ 36), GIM model (cycle > 36) [Iijima et al., 1999] | **dual-frequency altimeter range measurement (cycle 6-64) and GIM model >cycle 65 [Iijima et al., 1999] corrected from 8 mm bias** | GIM model [Iijima et al., 1999] | | | |
| **Dry troposphere** | Model computed from ECMWF Gaussian grids (new S1 and S2 atmospheric tides are applied) | Model computed fromECMWF rectangular grids (new S1 and S2 atmospheric tides are included) | **Model computed from ERA Interim Gaussian grids (new S1 and S2 atmospheric tides are applied)** | | | Model computed from ECMWF Gaussian grids (new S1 and S2 atmospheric tides included) | Model computed fromECMWF rectangular grids (new S1 and S2 atmospheric tides included) | Model computed from ECMWF Gaussian grids (new S1 and S2 atmospheric tides included) | Model computed from ECMWF Gaussian grids (new S1 and S2 atmospheric tides included) | |
| **Wet troposphere** | **JMR radiometer (replacement product) ≥50km from the coast + ECMWF between 10-50 km from the coast** | **AMR radiometer (enhancement product)** | TMR radiometer [Scharoo et al. 2004] | MWR radiometer | MWR corrected for 23.6Ghz TB drift [Scharoo et al. 2004] before Neutral Network algorithm | **MWR ≥50km from the coast + ECMWF between 10-50 km from the coast (cycle ≤94); MRW (cycle >94)** | GFO radiometer | ECMWF model | WMR radiometer | ECMWF model |
| **DAC** | MOG2D High Resolution forced with ECMWF pressure and wing fields (S1 and S2 were excluded) + inverse barometer computed from rectangular grids . | | **MOG2D High Resolution forced with ERA Interim pressure and wing fields (S1 and S2 were excluded) + inverse barometer computed from rectangular grids .** | | | MOG2D High Resolution forced with ECMWF pressure and wing fields (S1 and S2 were excluded) + inverse barometer computed from rectangular grids . | | | | |

| | | | | | | | | | | |
|---|---|---|---|---|---|---|---|---|---|---|
| **Ocean tide** | **GOT4v8 (S1 and S2 are included)** | | | | | | | | | |
| **Pole tide** | [Wahr, 1985] | | | | | | | | | |
| **Solid earth tide** | Elastic response to tidal potential [Cartwright and Tayler, 1971], [Cartwright and Edden, 1973] | | | | | | | | | |
| **Loading tide** | **GOT4v8 (S1 parameter is included)** | | | | | | | | | |
| **Sea state bias** | **Non parametric SSB [Tran, 2012] (using J2 cycles 1 to 36 with GDR-D standards)** | **Non parametric SSB [Tran, 2012] (using J1 cycles 1 to 111 with GDR-C standards and GDR-D orbit)** | Non parametric SSB [N. Tran and al. 2010] (using cycles 21 to 131 with GSFC orbit for TP-A; cycles 240 to 350 with GSFC orbit for TP-B) | BM3 | Non parametric SSB (using cycles 70 to 80 with DELFT orbit and equivalent of GDR-B standards) | **Non parametric SSB [Tran, 2012] compatible with enhanced MWR** | Non parametric SSB [N. Tran et al. 2010] (using cycles 130 to 172 with GSFC orbit) | Non parametric SSB from J1, with unbiased sigma0 | Hybrid SSB from R. Scharroo et al (2005) | Linear model |
| **Mean Sea Surface** | **CNES_CLS_2011 referenced to the 1993-2012 period** | | | | | | | | | |

Table 2: Variance of the differences between gridded DT2014 two-sat-merged products and independent TP interleaved along-track measurements for different geographic selections (unit = cm²). In parenthesis: variance reduction (in %) compared with the results obtained with the DT2010 products. Statistics are presented for wavelengths ranging between 65-500 km and after latitude selection (|LAT| < 60°).

|  | TP [2003-2004] |
| --- | --- |
| **Reference area*** | 1.4 (-0.7%) |
| **Dist coast > 200km & variance < 200 cm²** | 4.9 (-2.1%) |
| **Dist coast > 200km & variance > 200 cm²** | 32.5 (-9.9%) |
| **Dist coast < 200km** | 8.9 (-4.1%) |

*The reference area is defined by [330,360°E]; [-22,-8°N]*

Table 3: Taylor skill scores for the comparison of the geostrophic currents computed from altimetry or measured by drifters. Results obtained with DT2014 (2010) products are in bold (parentheses).

|  | Zonal | Meridional |
| --- | --- | --- |
| Outside the equatorial band | **0.83** (0.82) | **0.62** (0.63) |
| Inside the equatorial band | **0.87** (0.85) | **0.83** (0.81) |

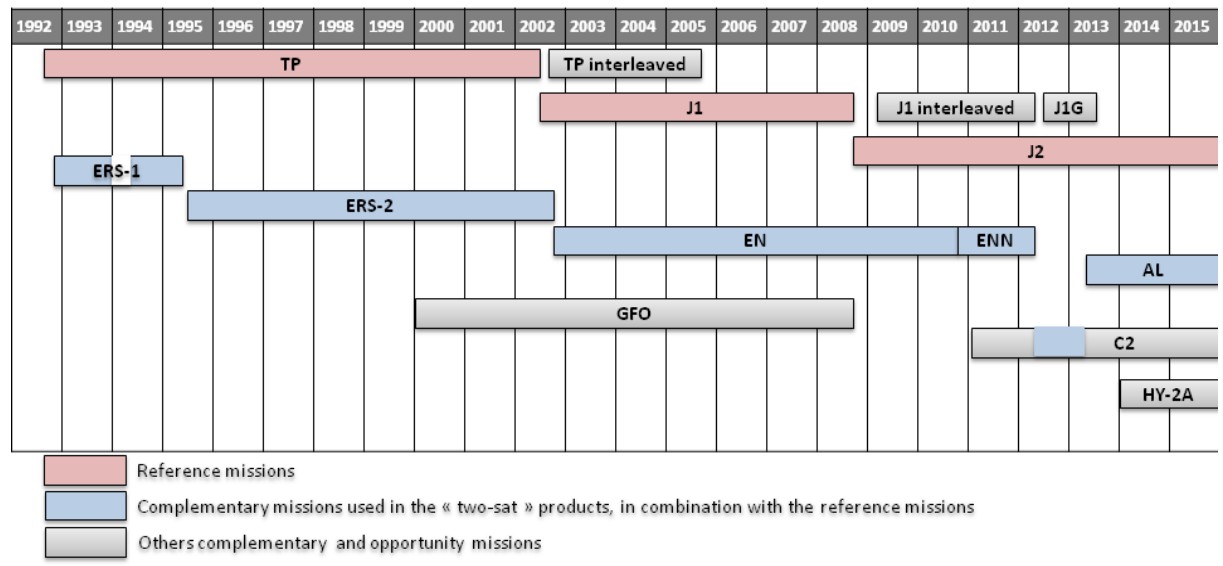

Figure 1: Timeline of the altimeter missions used (or expected) in the multi-mission DUACS DT system.

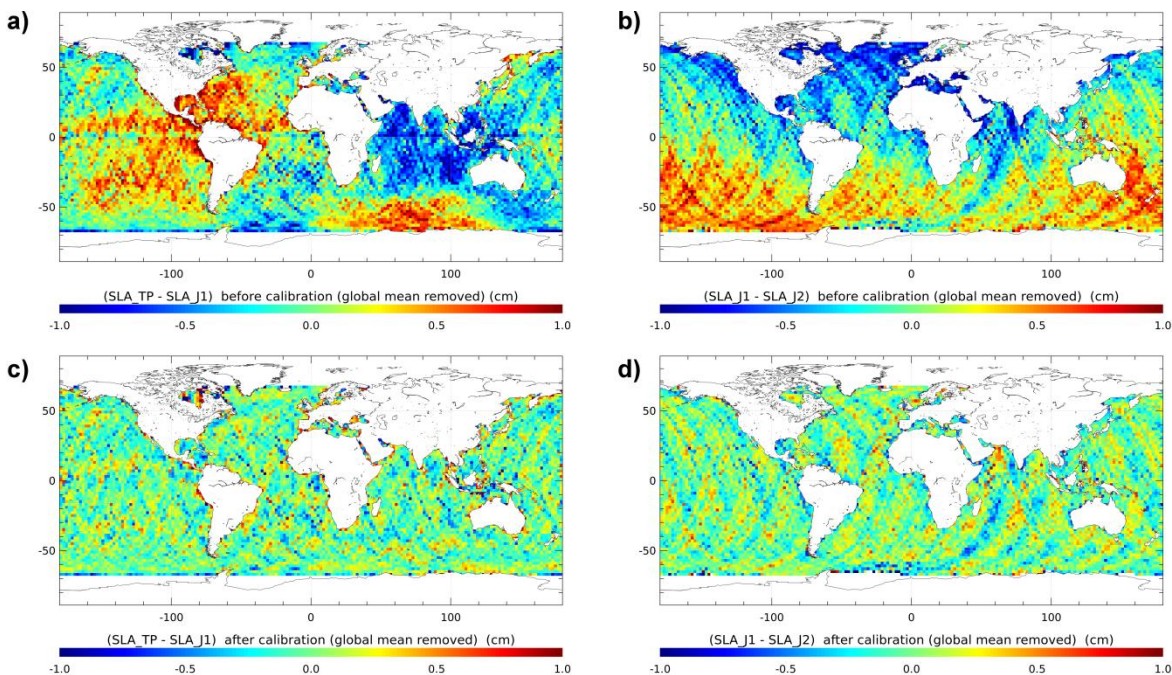

Figure 2: Regional SLA biases observed between TP and J1 during the cycles 1 to 21 of J1 before (a) and after (c) reduction of biases. Regional SLA biases observed between J1 and J2 during cycles 1 to 21 of J2, before (b) and after (d) reduction of biases.

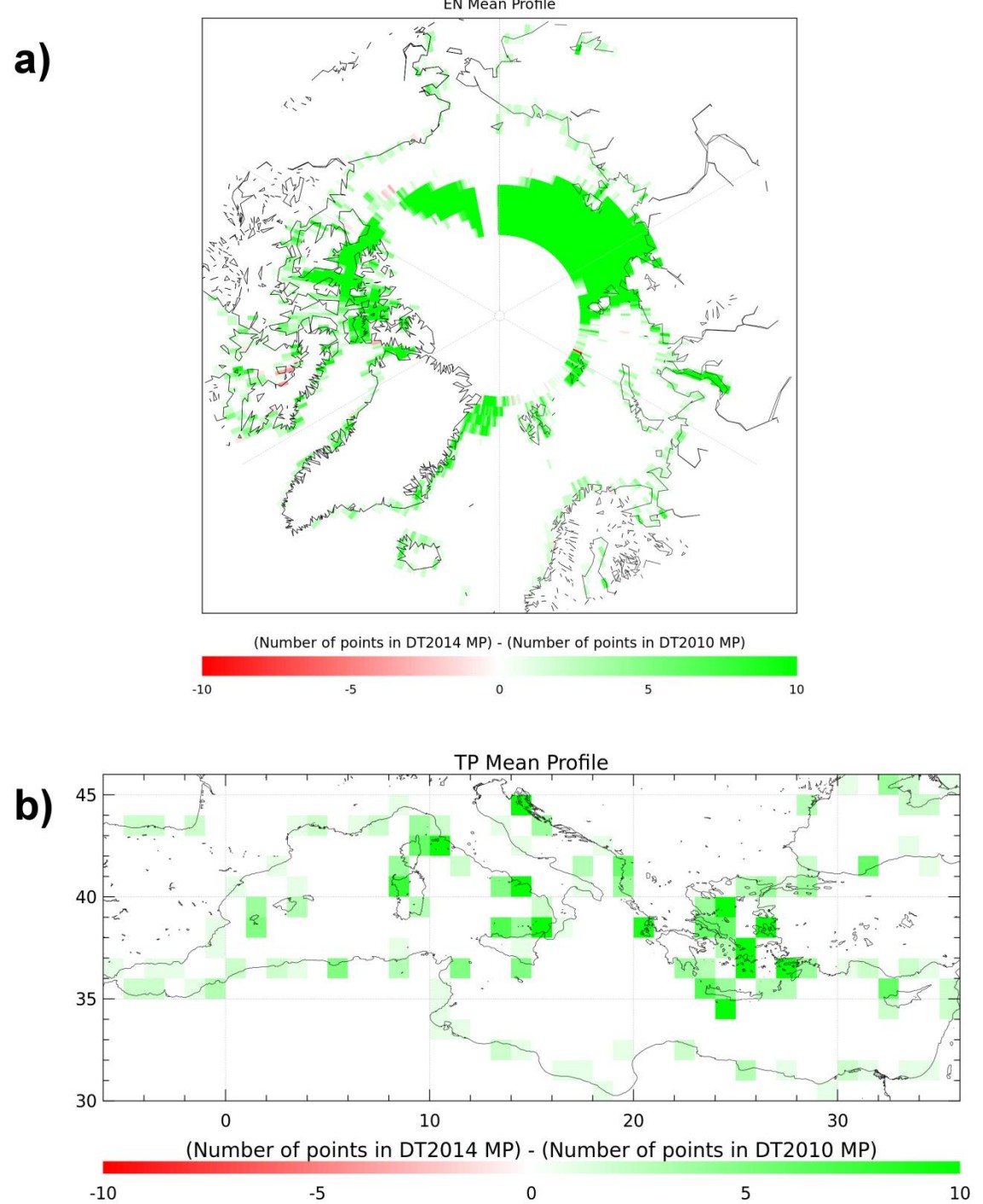

Figure 3: Differences of the number of points defined along the DT2014 and DT2010 versions of the Mean Profile defined along theoretical EN (a) and TP (b) tracks. Statistics done in 1°x1° boxes.

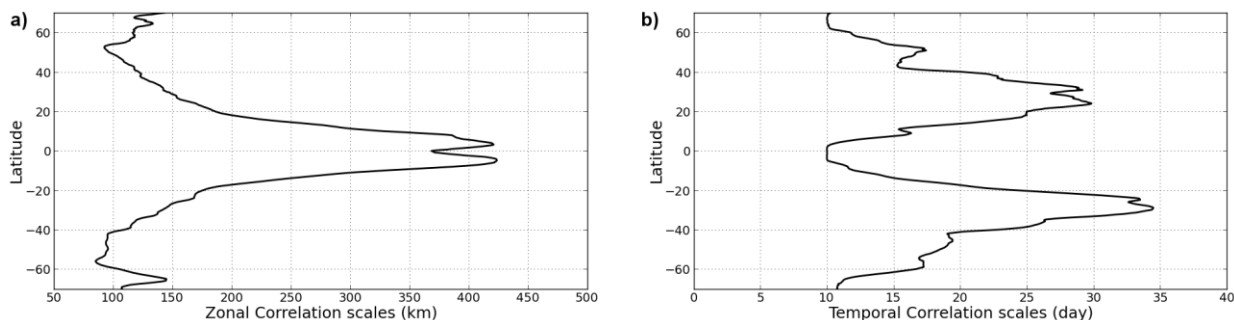

Figure 4: Evolution of the mean zonal (left) and temporal (right) correlation scales according to the latitude.

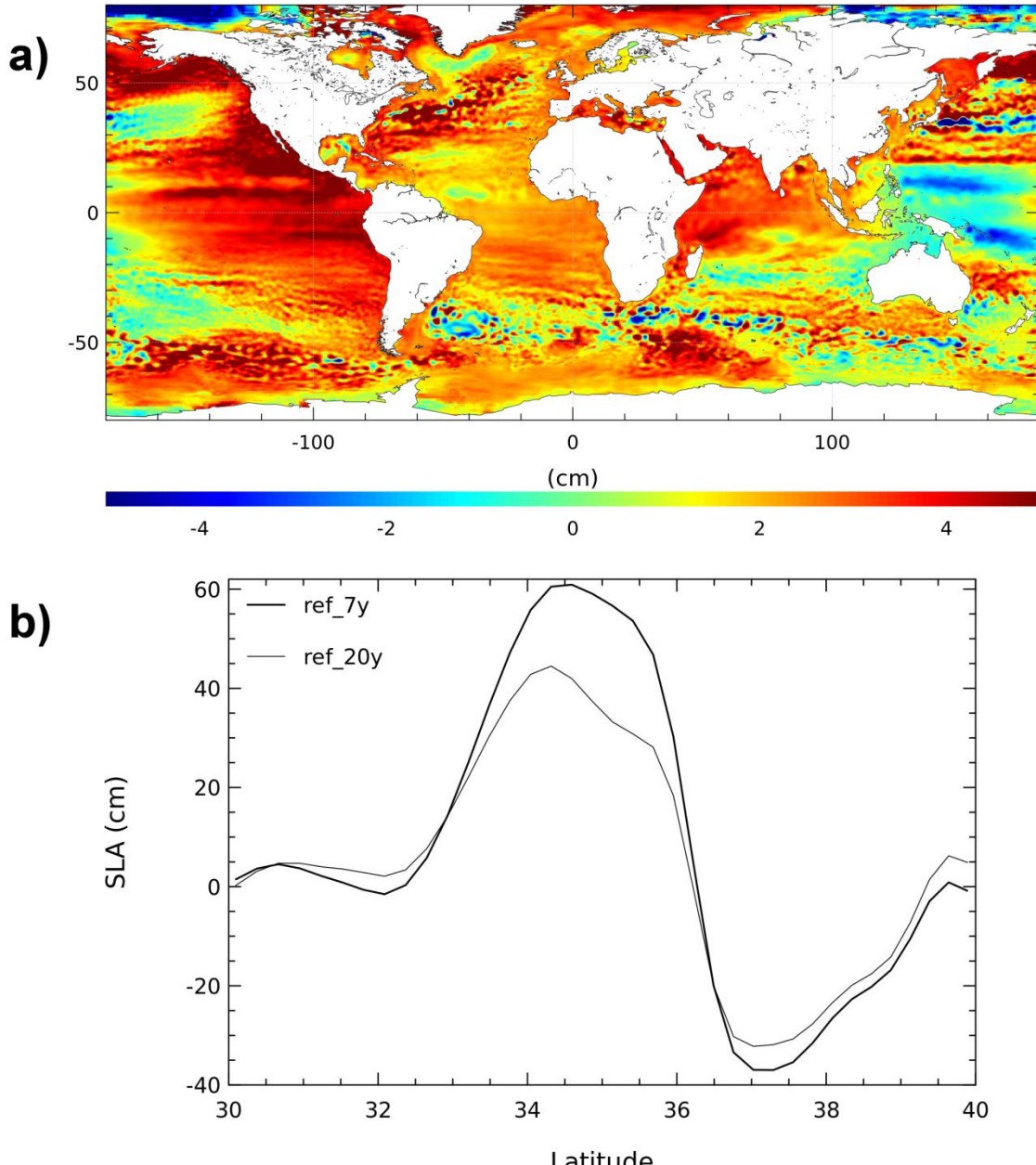

Figure 5: Impact of the change of reference period. a) regional MSL variation differences when considering the 7-year or the 20-year period. b) SLA along a J2 track crossing the Kuroshio, referenced to the 7-year (thick line) and 20-year (thin line) period.

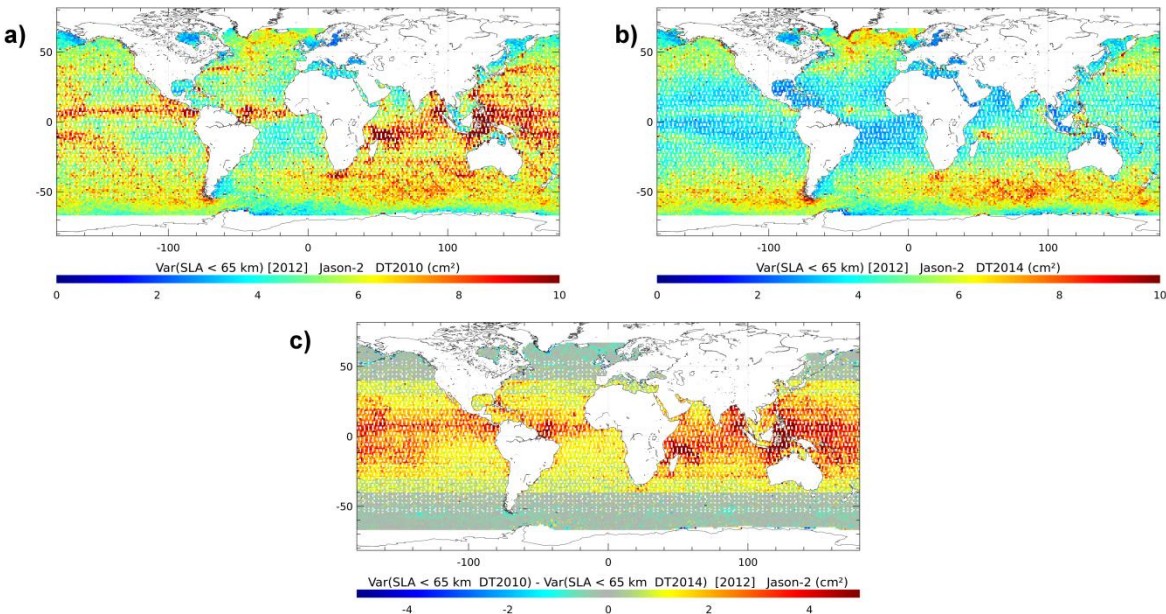

Figure 6: Variance of the short wavelength signal removed (by low-pass filtering) on L3 along-track J2 SLA in the DT2010 (a) and DT2014 (b) versions. c) Difference between a) and b). Statistics done over year 2012.

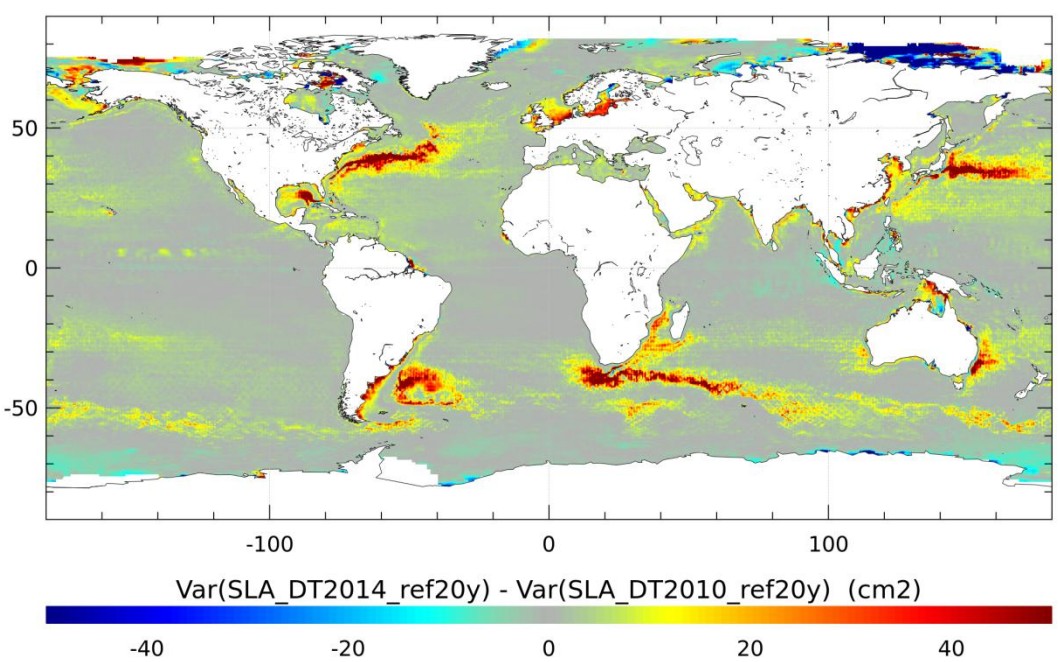

Figure 7: Difference between SLA variance observed with DT2014 gridded products and SLA variance observed with DT2010 products over the [1993-2012] period. Gridded products merging all the altimeters available are considered (i.e., "all-sat-merged" in DT2014; "UPD" in DT2010). DT2010 products were referenced to the 20-year altimeter reference period and interpolated onto the ¼°x1/4° Cartesian grid for comparison with DT2014.

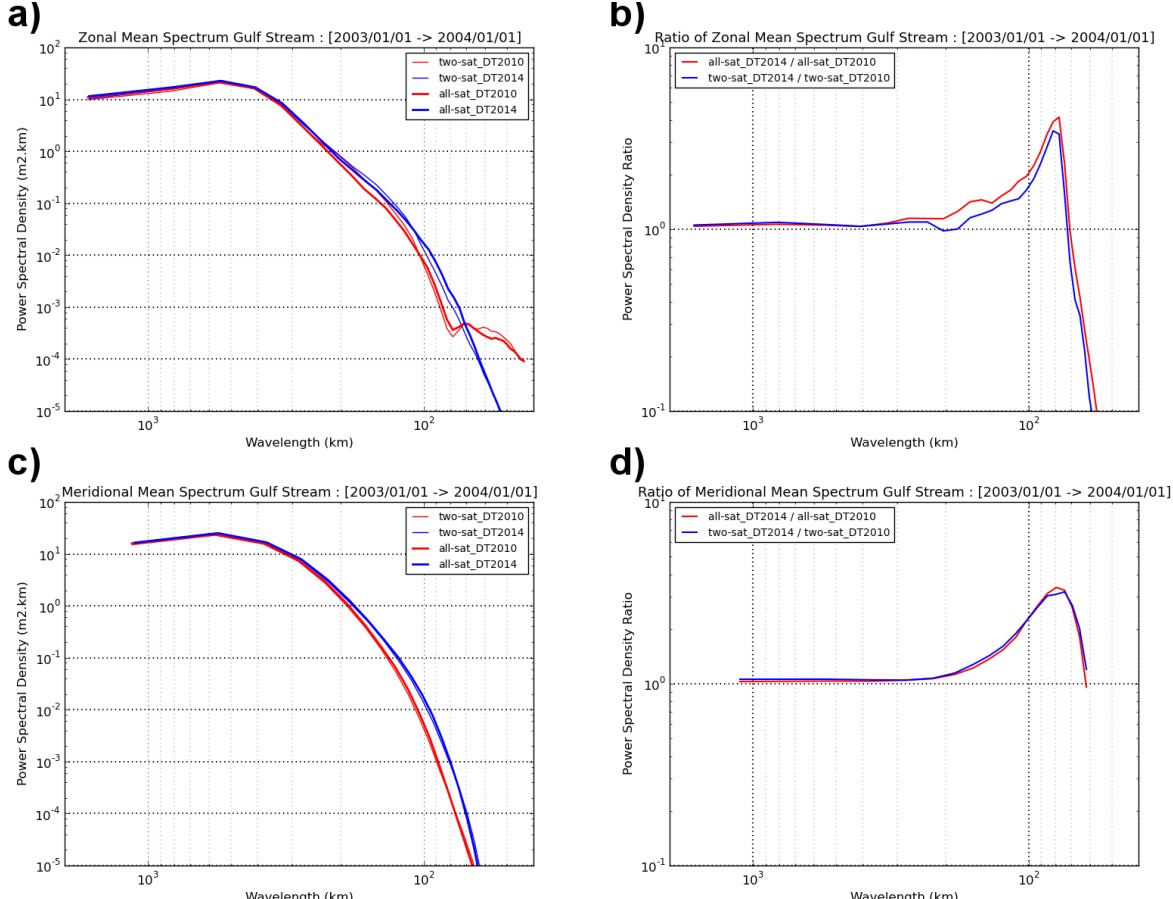

Figure 8: Mean zonal (a) and meridional (c) power spectral density (PSD) computed from gridded DT2014 (blue) and DT2010 (red) all-sat-merged (UPD; thick line) and two-sat-merged (REF; thin line) SLA fields over the Gulf Stream area during 2003 (when the constellation included J1, TP interleaved, GFO and EN). Ratio between DT2010 and DT2014 PSD when all-sat-merged (UPD; red line) and two-sat-merged (REF; blue line) are considered: zonal (b) and meridional (d) components.

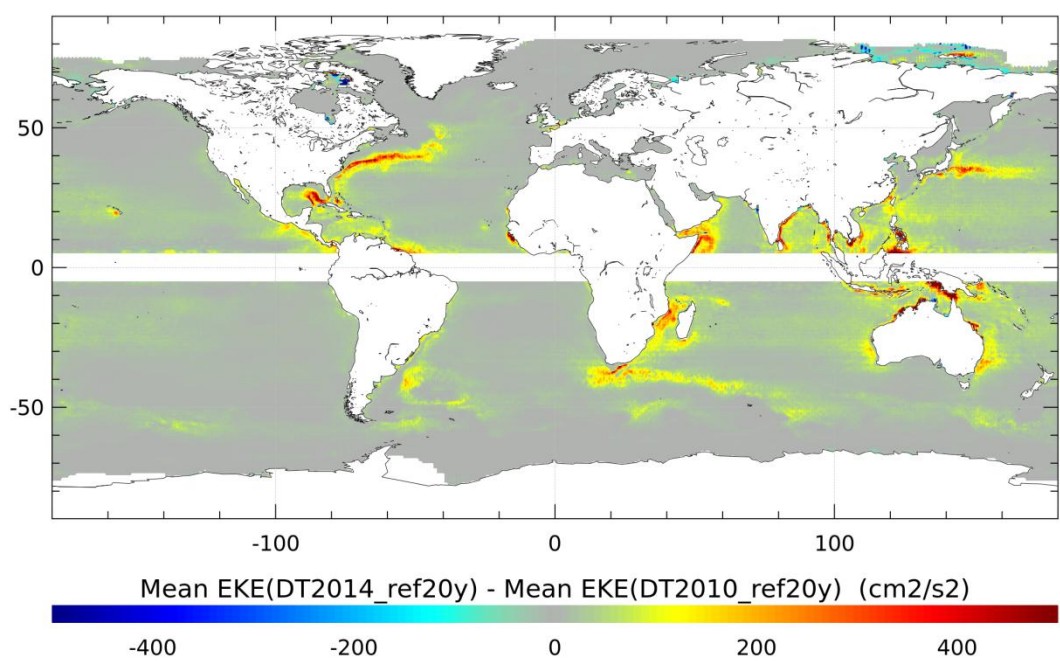

Figure 9: Difference of the mean EKE computed from DT2014 and DT2010 SLA over the [1993-2012] period. Gridded SLA merging all the altimeters available are considered (i.e., "all-sat-merged" in DT2014; "UPD" in DT2010). DT2010 SLA was referenced to the 20-year altimeter reference period and interpolated onto a ¼°x1/4° Cartesian grid for comparison with DT2014. The same methodology (centered differences) was used for geostrophic current computations for DT2010 and DT2014.

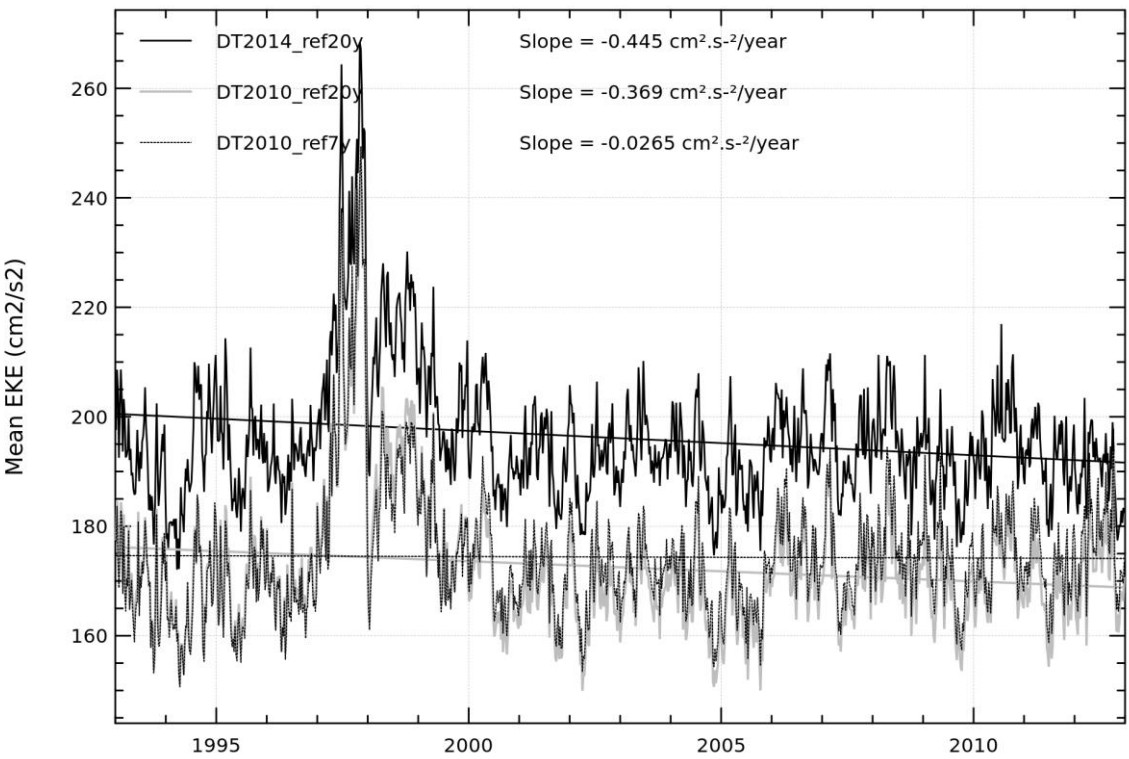

Figure 10: Evolution of the mean EKE over the global ocean (selection of latitudes lower than 60°), computed from the DT2014 (black line) and DT2010 SLA gridded products referenced to the 20-year period (black dotted lines) or to the 7-year period (grey lines). The same methodology (finite differences) was used for the geostrophic current computation for DT2010 and DT2014.

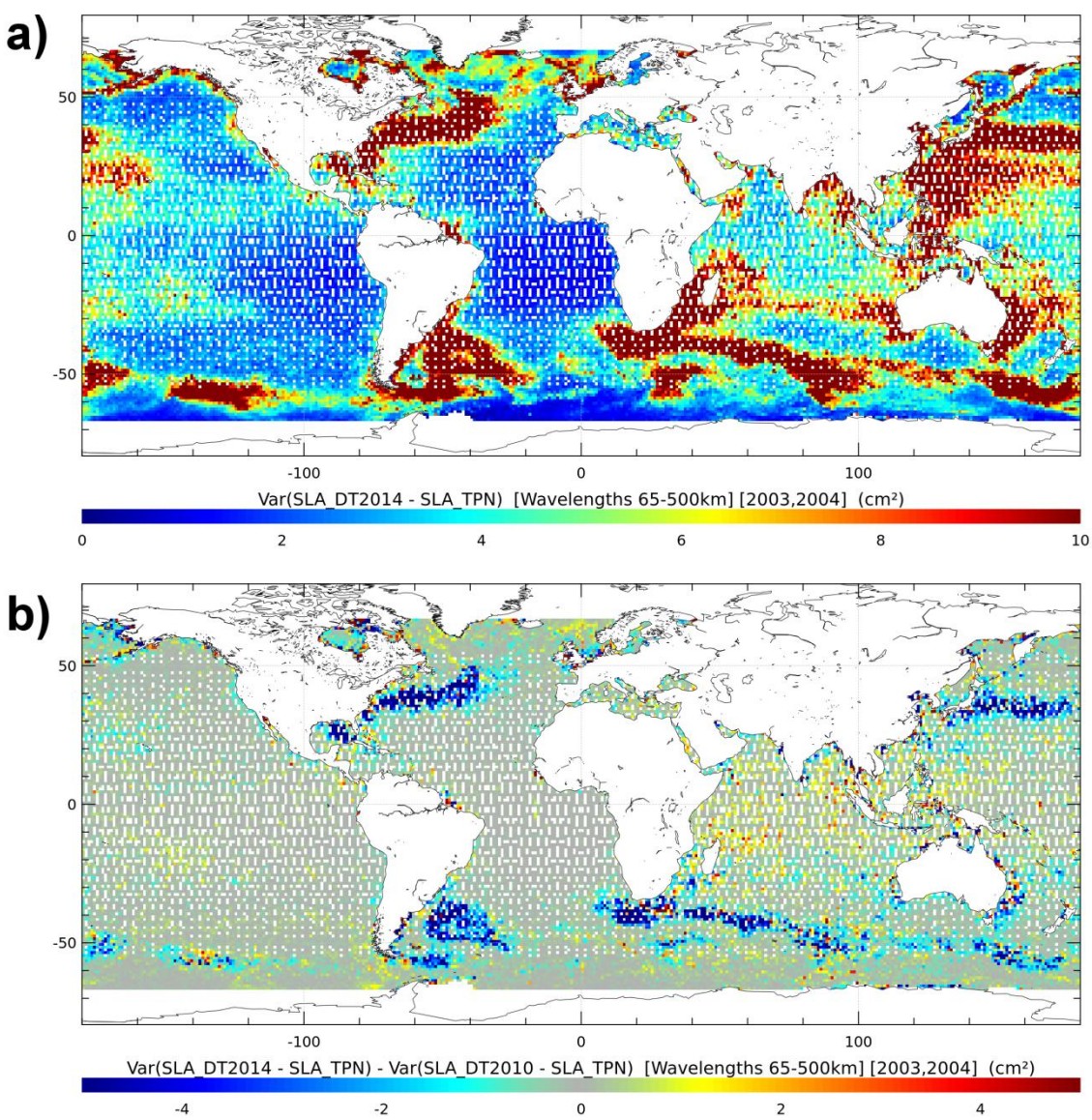

Figure 11: a) Variance of the differences between gridded DT2014 two-sat-merged SLA and independent TP interleaved along-track SLA measurements. Statistics are presented for wavelengths ranging from 65-500 km. (unit = cm²). b) Differences with the results obtained with the DT2010 SLA products. Negative values indicate a reduction of the differences between gridded and along-track SLA when DT2014 products are considered.

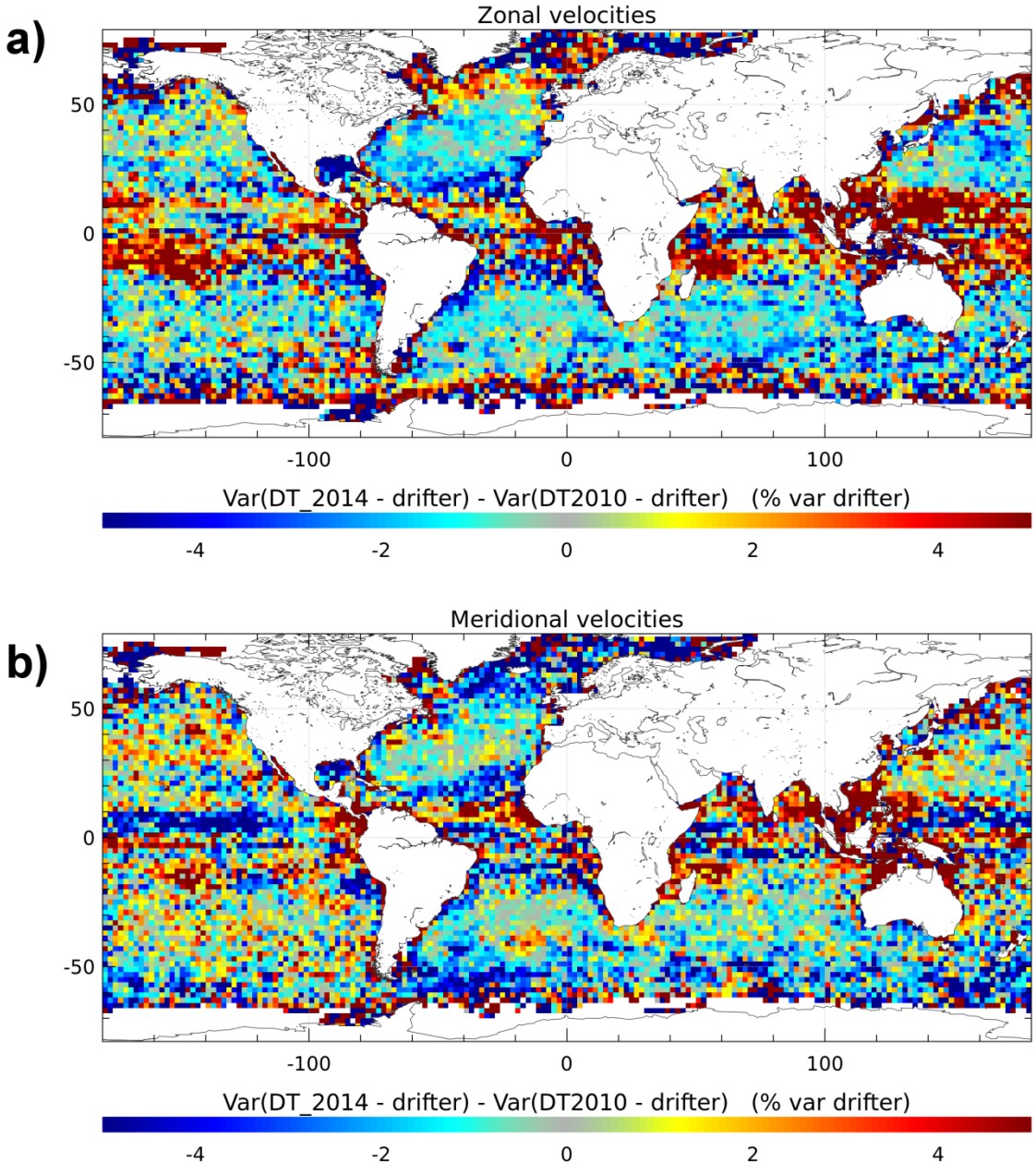

Figure 12: Maps of the difference of the variances of the altimeter geostrophic currents minus drifter measurement differences, using successively DT2014 and DT2010 SLA gridded products. The difference of variance is expressed in % of the drifter variance. Zonal (a) and meridional (b) component differences. Negative values mean that the variance of the differences between geostrophic currents from altimetry and from drifter measurement is reduced when considering the DT2014 product.

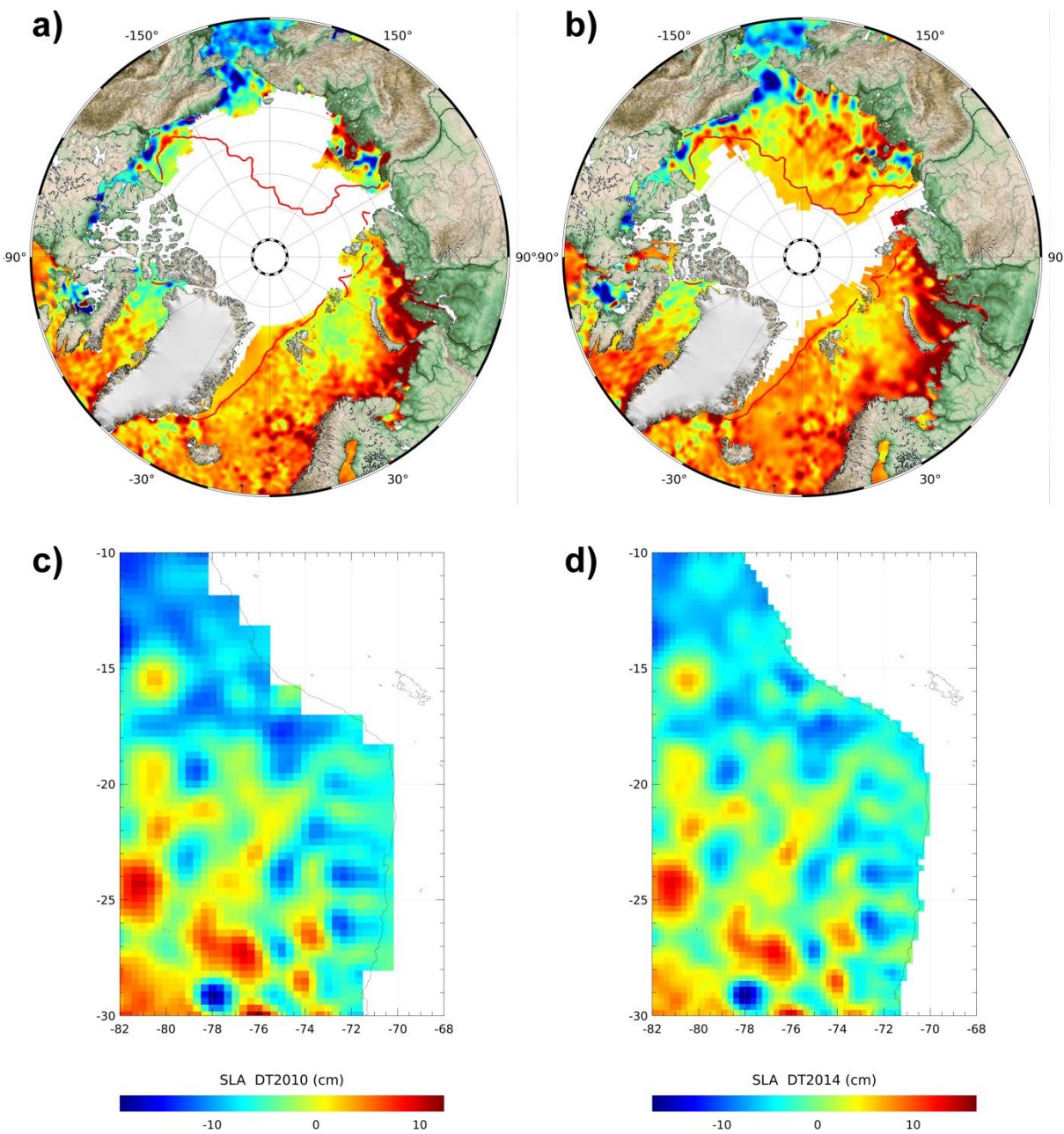

Figure 13: Coverage improvement associated with the DT2014 reprocessing. Map of SLA for day 2011/10/17 over the Arctic Ocean observed with the DT2010 (a) and DT2014 (b) products. Sea ice extent is shown with red line (OSISAF product). Same map along the western South American coast with DT2010 (c) and DT2014 (d).

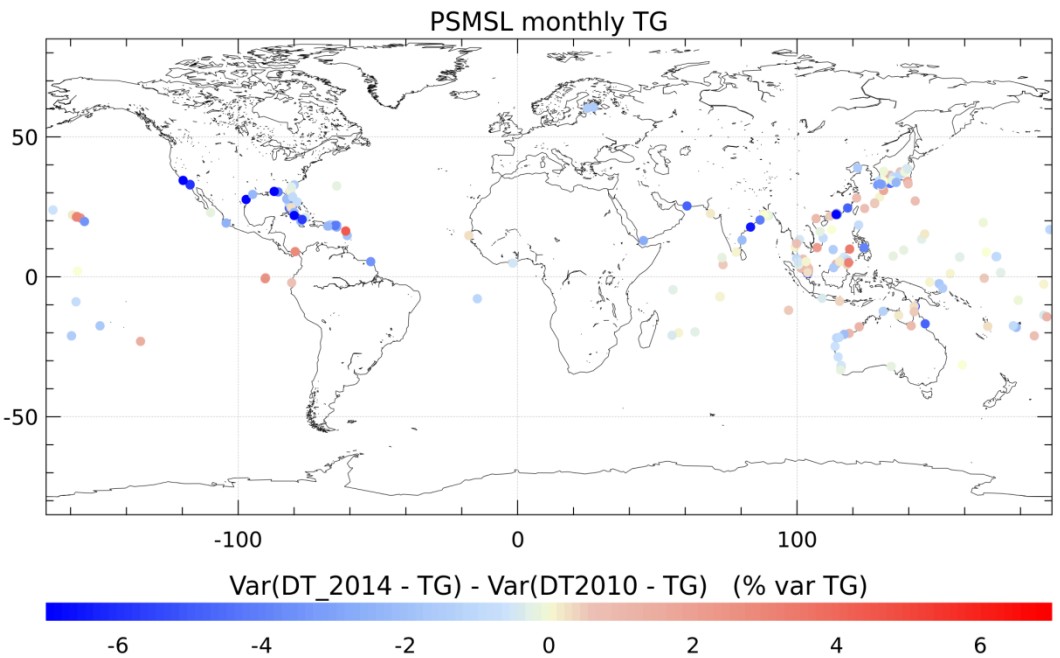

Figure 14: Difference of the variance of the altimeter SLA minus TG SLA differences, using successively DT2014 and DT2010 SLA gridded products. Monthly TG from PSMSL. Negative values mean that the SLA differences between altimetry and TGs is reduced when considering DT2014 products.

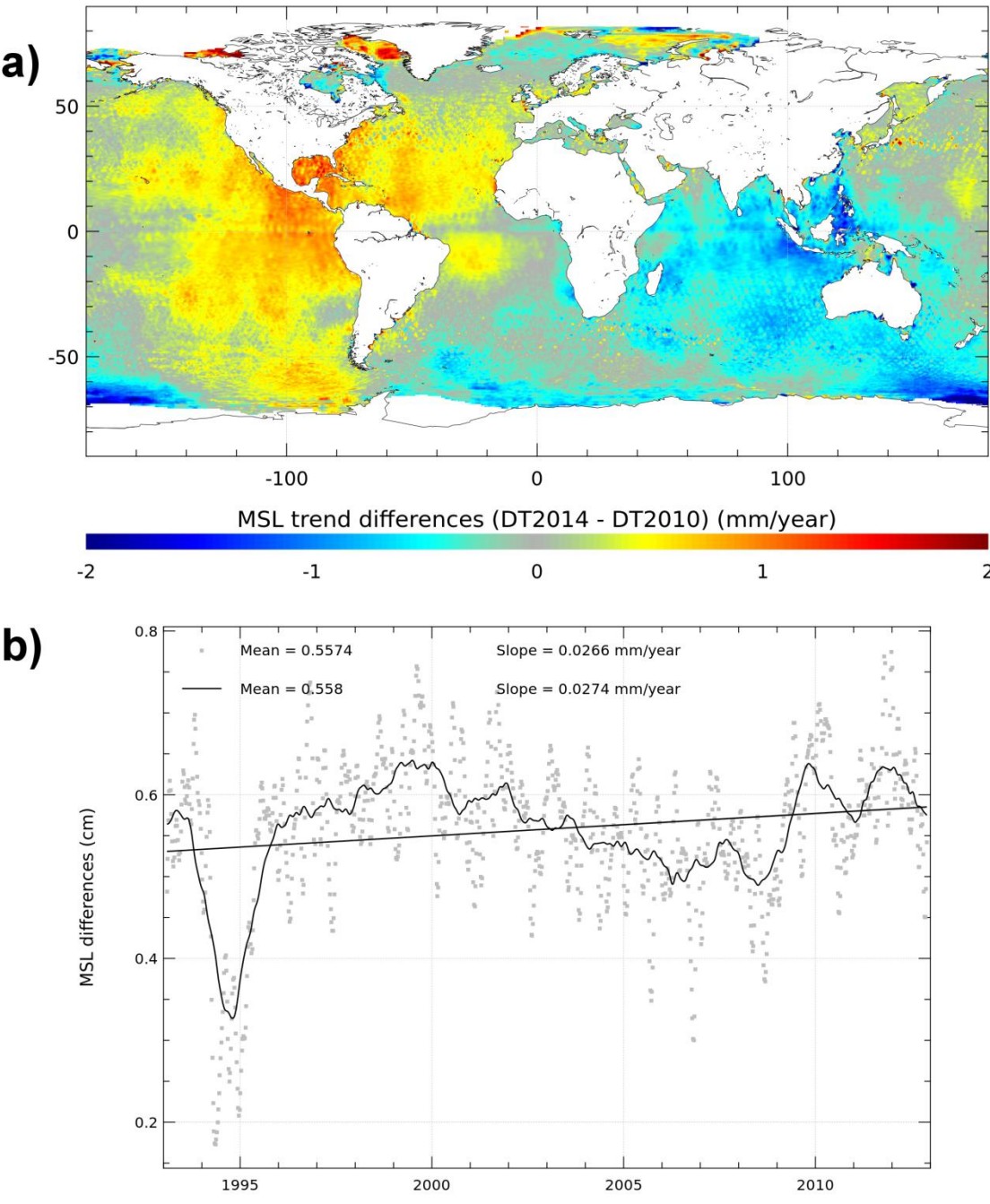

Figure 15: a) Map of the differences of the local MSL trend estimated from the DT2014 and DT2010 gridded

SLA products. MSL estimated over the [1993-2012] period. b) Temporal evolution of the differences of the

global MSL estimated from DT2014 and DT2010.

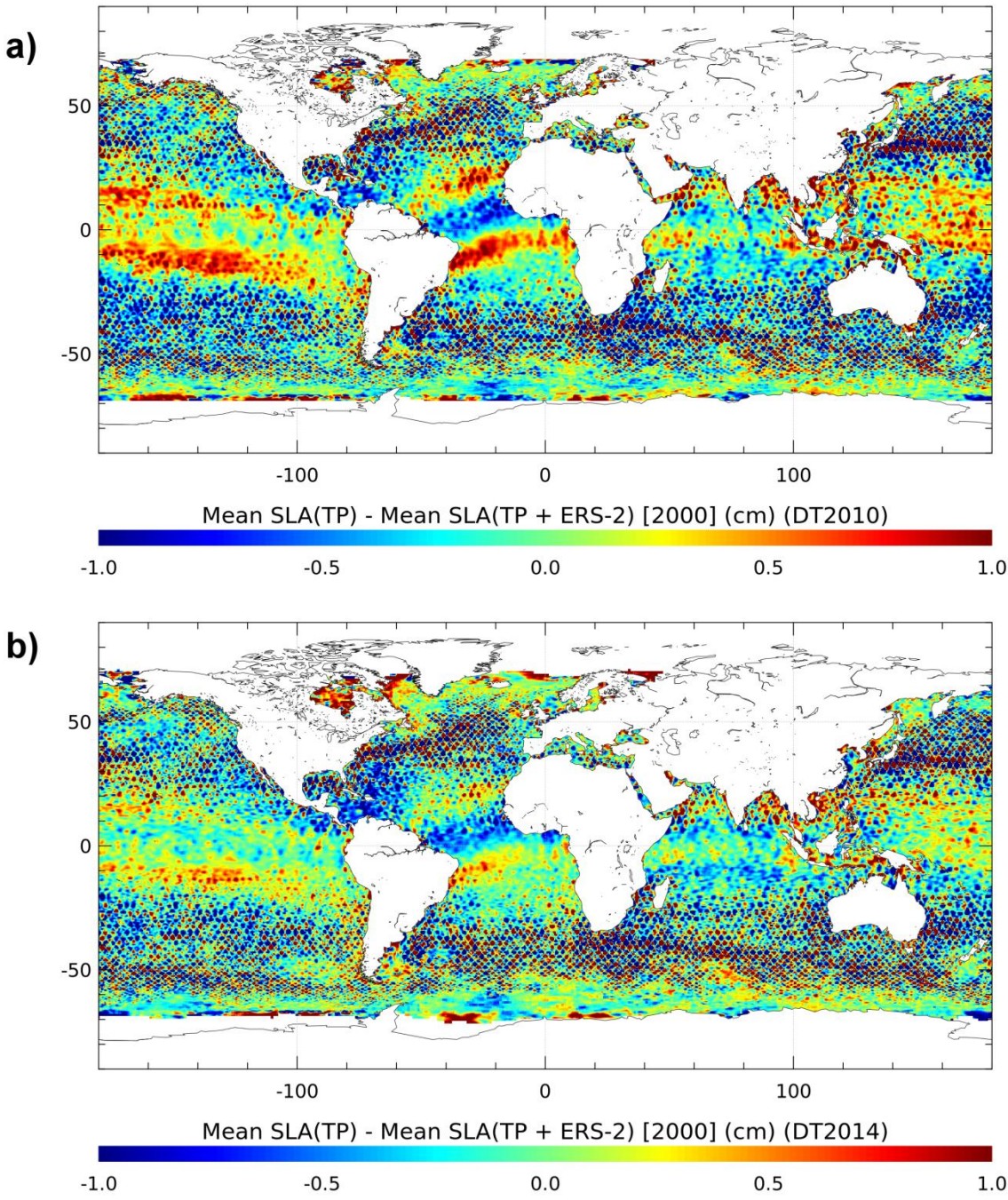

Figure 16: Difference of the mean SLA over the year 2000, measured with TP only, and with the merged TP+ERS-2 product. Comparison done for the DT2010 (a) and DT2014 (b) products.

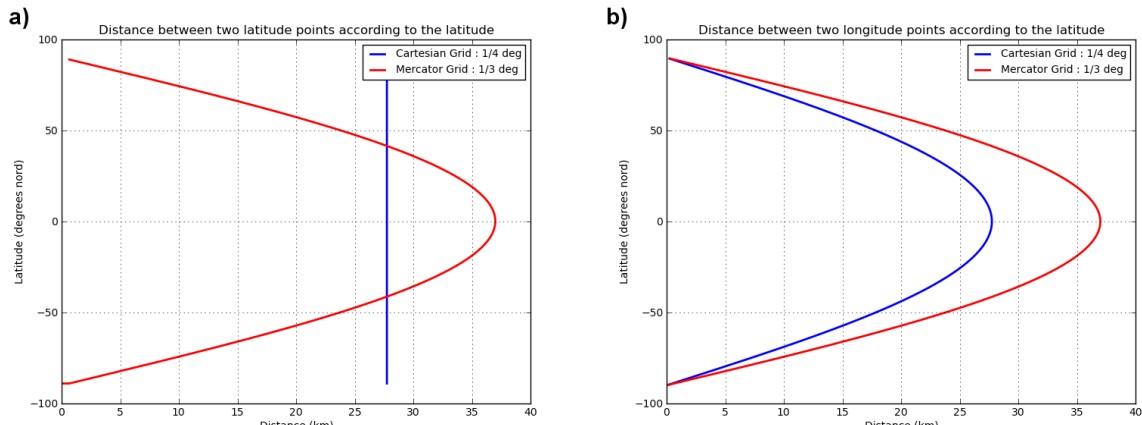

Figure C1 : a) Difference between two successive grid points on a meridional section as a function of latitude, at 1/4°x1/4° Cartesian resolution (blue) and 1/3°x1/3° Mercator resolution (red). b) same as left but for a zonal section.