# Peer review of "DUACS DT2014 : the new multi-mission altimeter dataset"

_Ocean Science, 2015_

## Referee Comment (RC1) · Anonymous Referee #1 · 3 Feb 2016

This paper describes the latest revision (DUACS DT2014) of the processing system behind the very popular satellite altimeter-based gridded sea level product - known to many researchers as the 'AVISO sea level data'. Indeed, this gridded altimetry product is frequently used in preference to the along-track source data, so many users have lost sight of the many issues associated with the processing of the along track data to the SLA stage, or the multimission gridding process.

In this context, the release of a new version of this data set is a very significant announcement, and is an opportunity to inform and educate the users. The paper describing the production and assessment of the data set, and how and why it differs from its predecessor, is bound to be heavily cited. In my opinion, however, the present draft of this paper is not yet satisfactory for this heavy responsibility.
[Figure]

I think this paper needs to be extensively re-written. It is very light on specifics, but still manages to be 24pp of text, which takes far too long to read for the amount of detail that is presented. The 2nd sentence of the abstract "Numerous and impacting evolutions have been implemented at each step of this new data processing" is indicative of the poor English and imprecise style of the whole paper. Reviewing criterion 11 "is the language fluent and precise?" is certainly not met.

There are many details of the processing that should be explained, but are not. Indeed, the abstract mentions only "The main one is the use of...". The other 'evolutions' are not listed in the abstract. Review criterion 9 is therefore not met.

Section 2 on "Data processing" says the "altimeter standards" were chosen by a rigourous selection process (described by Ablain et al 2015). Altimetry specialists will know what is meant by this but a little more explanation would help the target audience of this paper, so that they can understand the cause of the differences between the resulting product and its predecessor (which will have impacts on analyses and interpretations of the older dataset). For example, on p23 line 26 there is mention of the east-west dipole error that has now disappeared because of the new 'standards'. This needs to be much better explained.

Page 8 discusses the changes made to the 'multimission mapping' or gridding process. A critical change is a 'better defined correlation scale' which needs a more quantitative explanation. Conversely, the change to a finer spatial (0.25 deg) and temporal (1 day instead of 7 day) sampling could be much more compactly described.

Page 13 says that dynamic height anomalies from T/S profiles are compared with the 'equivalent' field from altimetry. There should be mention here that the third term in this equation is the eustatic change of sea level.

The results of the comparisons of the new data set with its predecessor, and with independent data (tide gauges, drifters) are an interesting part of the paper, but the description of these results needs to be much clearer and informative. For a start,

there is too much use of the word 'products' instead of geophysical quantities. See the caption to Fig. 5, for example. Wording in many places needs to be much clearer, for example page 17 line 25 refers to the 'DT2014 currents intensity' and then the 'variability'. My guess is that the 'intensity' is a measure of the time-mean, but I am not sure.

Some qualitative conclusions could also be better justified. The page 19 line 25 statement "thus reinforce our confidence in these good results" came as a surprise to me after reading of changes that are near zero, or a mix of positive and negative values.

The English of the paper (grammar, vocabulary, spelling, phrasing) certainly needs to be improved throughout (as well as the imprecise style). In some places the poor English is understandable "..error was quite two time more stronger.." but in many places it is not (e.g. "where belong +400cm2/s2 are observed"). I had to consult a dictionary to find out what 'restitute', 'conduce' and 'traduce' all meant. The word 'underline' is idiomatically correct but heavily over-used, as is 'impacts'. "From" is spelled "form" in several places.

The Figure captions all need to be improved. Example 1: it is ambiguous to describe Fig. 12b as showing the 'variance reduction'. The values are mostly negative. Is a negative reduction a reduction or an increase? It is clearer to say 'change'. Example 2: Fig. 15a shows the "MSL trend difference between DT2014 and DT2010" while 15b shows the "MSL differences..". The M in MSL usually stands for 'mean', sometimes a time-mean, but sometimes a spatial mean. In 15a it can be neither, while in 15b it must be the spatial mean.

---

## Referee Comment (RC2) · D. Chelton (Referee) · 12 Feb 2016

Comments on "DUACS 1 DT2014 : the new multi-mission altimeter dataset reprocessed over 20 years" by M.-I. Pujol, Y. Faugere, G. Taburet, S. Dupuy, . Pelloquin, M. Ablain and N. Picot Ocean Science Discussions Manuscript #os-2015-110

**Reviewer: Dudley Chelton**

This paper is a very valuable contribution to the altimeter data user community and is likely to be heavily cited because of its comprehensive summary of the improvements made to the latest version of the DUACS altimeter datasets (more commonly referred to colloquially as the "AVISO dataset"). In addition to a general overview of all of the changes that were made between the old altimeter dataset (named DT2010 in this paper) and the new DT2014 dataset, the two most important contributions of this paper are the documentation of the estimated errors of the gridded products in Sec. 3.2.3, and the extensive validation effort by comparisons with in situ data in Secs. 3.2 and 3.4 (monthly mean tide gauge sea level; dynamic height derived from temperature-salinity profiles; and surface drifter data).

While the value of this paper is indisputable, the present version needs a lot of work. I list below two major issues, followed by numerous specific comments, questions and suggestions and, lastly, a list of minor comments, mostly editorial in nature. My comments are mostly focused on the gridded products, which are what I have used almost exclusively for my own research.

My comments that follow are embarrassingly extensive, especially for a journal that posts reviews in their entirety in a Discussion section. The level of detail of my comments reflects the amount of time I've devoted in the past to thinking about these issues with the old DT2010 dataset. This paper is an opportunity to put all of those issues to rest once and for all for any future researchers using the DT2014 altimeter dataset.

**Major Comments**

1. Users have long been frustrated that it is so difficult to track down the details of the OI procedure used by DUACS to generate the gridded SSH fields. The complicated paper trail that users must claw their way through to understand all of these details is evident from lines 10-12 on page 2 of this manuscript in which the authors list a sequence of 7 publications (to which I would add Le Traon and Ogor, 1998) that describe the evolution of the DUACS system and associated products. My own frustration with this is summarized near the beginning of Appendix A2 of Chelton et al (2011, *Progress in Oceanography*) where it is stated that

[...] The details of this procedure have evolved somewhat over the years and an upto-date and comprehensive documentation of the procedure is not available. Many of the details are described in a sequence of published papers: Le Traon et al. (1995, 1998, 2003), Le Traon and Ogor (1998), and Ducet et al. (2000). [...] we summarize here our understanding of the procedure as it has most recently been implemented. In addition to the above publications, this summary is based on personal communication in December 2010 with G. Dibarboure at CLS in Toulouse, France who presently oversees the AVISO processing.

Rather than include the subsequent description of all of the details, I would have much preferred to point to a single publication in which users would find all of the details in a single place. Not including the details in this paper would perpetuate the difficulty for users to understand how the parameters of the OI procedure impact the interpretation of the data. One consequence of this is that many (probably even most) users believe that the new DT2014 dataset resolves features with spatial scales of  $\frac{1}{4} \times \frac{1}{4}$ ° and temporal scales of 1 day. This is, of course, not even vaguely close to the true spatial and temporal scales of the features that can be resolved. But without easy access to the specifications of the spatial and temporal autocorrelation functions used in the OI analysis, it is understandable that users will make such gross misinterpretation of the resolution capabilities of the DUACS/AVISO dataset. This paper is an opportunity to provide users with "one-stop shopping" for everything they need to know about the dataset.

2. The English grammar and spelling are in need of extensive work by a copy editor. There are also a lot of instances where the authors used incorrect words. These English problems are understandable since English is not the native language of any of the authors. And their English is infinitely better than my French. But the extent of the problem is quite extreme. I did not carry out a careful statistical analysis, but I have a sense that more than half of the sentences have an English problem of one sort or another, albeit many times only a minor problem. Much of the time, the reader can guess what the authors are trying to say. But there are quite a few cases where I just couldn't figure it out. I draw attention to some of those in the Minor Comments below.

**Specific Comments**

1. Page 1, lines 14-15: It is stated here and in several other places in the paper (e.g., page 3, lines 12-13, and somewhat less forcefully on Page 21, lines 23-25) that the main improvement in the DT2014 dataset compared with DT2010 is in the use of a 20-year reference period rather than the 7-year reference period used previously. I think this is actually a relatively minor point. Indeed, the authors explain how to change the reference period in a very short appendix at the end of the paper. The main improvements in the DT2014 dataset are the reduced smoothing of the along-track data, improved orbit accuracy, improved ionospheric and atmospheric corrections and the refined scales of the correlation functions used in the OI procedure (the details of which the authors relegate to a reference to Dibarboure et al., 2011, instead of

including them in this paper).

- 2. Page 6, lines 13-14: "Reference mission" is defined on lines 2-4 on this page, but "secondary mission" is not clearly defined anywhere. The informed reader will know that the sequence of secondary missions is ERS-1, ERS-2, ENVISAT, Cryosat, AltiKA and HY-2A, but this should be spelled out here to avoid ambiguity.
- 3. Page 8, lines 2-3: I'm not certain I understand what is meant by "The wavelengths ranging nearly 200 to 65 km are filtered". I think that the short end of this may be imposed by the along-track smoothing applied to the track data. But from page 11, lines 11-12, I think that 65 km is a global average for that, rather than a fixed number everywhere. And furthermore, the wavelengths shorter than 65 km are also attenuated, so there should not be a lower range of 65 km for attenuation by filtering. Is the 200 km upper end of the range something that is imposed explicitly in the OI procedure? In any case, the wording is awkward. Perhaps say something like "SSH variability with wavelengths shorter than about 200 km is attenuated."
- 4. Page 8, lines 8-15: This is the place to include the specifics of the parameters for the OI mapping procedure. In my opinion, it is not enough to simply say that the parameters used previously for DT2010 were refined for DT2014 and then send the readers to Dibarboure et al. (2011) to find those details.
- 5. Page 8, lines 16-18: I would follow this first sentence of this paragraph with a qualifying sentence like, "But note that the time scales of the variability that are resolved in the DT2014 dataset are not substantially different from DT2010; these time scales are imposed by the temporal correlation function used in the OI mapping procedure."
- 6. Page 8, lines 21-22: Change "an improved resolution" to "a higher grid resolution" so that readers don't draw the incorrect conclusion that the smaller grid spacing using essentially the same OI correlation parameters somehow magically resolves smaller-scale variability.
- 7. Page 8, line 22: Prior to the sentence that begins "These latitudes include the main part…", insert a qualifying sentence like "Note, however, that the spatial scales of the features that are resolved in the DT2014 fields are about the same (perhaps slightly smaller) than in the DT2010 fields; these spatial scales are imposed by the spatial correlation function used in the OI mapping procedure."
- 8. Page 8, lines 25-26: I'm not sure I agree with the statement that the new gridding "reduces the capability of the gridded products to accurately represent the mesoscale signal in high latitude areas." The  $\frac{1}{4}^{\circ} \times \frac{1}{4}^{\circ}$  is much finer than the spatial correlation scales used in the OI mapping procedure. I therefore do not expect there to be much

difference in the feature resolution capability of the new  $\frac{1}{4}^{\circ} \times \frac{1}{4}^{\circ}$  gridded product compared with the old  $\frac{1}{3}^{\circ} \times \frac{1}{3}^{\circ}$  Mercator gridded product.

- 9. Page 8, line 29: I am wondering what happens at the highest latitudes in the DT2014 dataset at times when data are not available "from the recent altimeters like C2". Are the gridded values flagged as missing, or are they filled with some other value such as the mean?
- 10. Page 9, line 8: After this first paragraph summarizing the processing for SSH, it would be helpful to include a summary of the details of the calculation of geostrophic velocity. For example, it is my understanding that the DT2014 processing used a wider stencil as suggested by Arbic et al. (2012) to compute the SSH derivatives. Arbic et al. (2012) is included in the reference list, but I could not find it cited anywhere in the paper. The authors apparently intended to discuss the changes in the derivative calculations but either forgot to include that text or inadvertently deleted it from an earlier draft of the paper.
- 11. Page 9, line 12: The phrase "The mesoscale signal is indeed better reconstructed..." in the discussion of the all-sat-merged product compared with the two-sat-merged product is ambiguous. The meaning of "better" should be defined. In lines 20-21 on this same page, it is stated that the OI parameters are the same for both products. The all-sat product therefore doesn't have any better space-time resolution than the two-sat product. So "better" evidently means "more accurate". This is an important point because virtually all users believe that the all-sat product has higher spatial resolution, which is not the case since it uses the same correlation parameters in the OI mapping procedure. But the all-sat product is presumably more accurate.
- 12. Page 11, lines 5-7: Defining the resolution of the along-track data as the point where the spectra of signal and noise intersect is too liberal. With a signal-to-noise ratio of 1, it is impossible to distinguish between signal and noise. In my experience, based on visual assessment of various fields with simulated noise added, a more reasonable choice is a S/N variance ratio of 10, which corresponds to a signal standard deviation that is about 3 times larger than the noise standard deviation. At a minimum, a S/N standard deviation ratio of 2 is needed to distinguish signal from noise (i.e., a S/N variance ratio of 4).
- 13. Page 11, line 7: It should be clarified that the 55 km mesoscale resolution capability corresponds to wavelength resolution. And the apparent contradiction between 55 km on line 7 and 65 km on line 12 should be clarified.
- 14. Page 14, lines 18 and 25: I am very surprised, and indeed skeptical, that changing the gridding from the original  $1/3^{\circ} \times 1/3^{\circ}$  to  $1/4^{\circ} \times 1/4^{\circ}$  is a bigger factor than the improved processing in determining the increased SSH variance of DT2014 compared with

DT2010. I would have thought that the primary explanation for the increased variance in DT2014 is the reduced smoothing applied to the along-track data that are used in the OI procedure. I wonder if the authors can somehow assess the impact of the different along-track smoothing independently of other changes to the gridded products.

- 15. Page 15, lines 11-21: I find the discussion in this paragraph to be rather confusing. But in any case, I disagree with the statement on line 14 that wavelength scales of 80-100 km are "fully observable". With DT2010, SSH is fully resolved only for scales longer than 200 km (Chelton et al., 2011, *Progress in Oceanography*). It is asserted later (Page 24, line 3) that DT2014 has a wavelength resolution of 150 km (but see comment #21 below). The statement on lines 11-21 that scales of 80-100 km are fully resolved is therefore incorrect.
- 16. Page 15, lines 29-30: It is not intuitive that the variance in DT2014 can be 20% smaller than the variance in DT2010 in low variability areas and near the eastern boundaries. It would be good to include some discussion of why this happens.
- 17. Page 16, lines 2-5: Same comment as #14 above for SSH variance: I am very surprised that the finer gridding has a bigger impact on geostrophic current variance than do all of the other changes in the processing.
- 18. Page 16, lines 27-31: The error estimates are stated as 4.9 cm2 in low variability areas, 32.5 cm2 in high variability areas and 1.4 cm2 in very low-variability areas. These numerical values seem to differ from the numbers given elsewhere in the manuscript. For example, the three numbers that are stated on lines 21-23 of the abstract on page 1. There are perhaps similar inconsistencies elsewhere in the paper.
- 19. Page 18, lines 1-15: It would probably be easier to interpret comparisons of speeds rather than velocity components since the velocity fields are very anisotropic so that analysis of each velocity component separately is difficult to interpret.
- 20. Page 22, lines 5-7: Same comment as #14 and #17 above.
- 21. Page 24, lines 3-8: This discussion of the resolution capability of DT2014 versus DT2010 is important but confusing. On line 3, the authors assert that the resolution of DT2014 is 150 km, which can be compared with the 200 km resolution estimated for DT2010 by Chelton et al. (2011, *Progress in Oceanography*). The authors cite the OSTST presentation by Chelton et al (2014) for evidence of the 150 km resolution for DT2014. If they want to include this level of detail in this paper, then I think they should include a figure showing this result, rather than pointing the readers to an abstract for the OSTST meeting. On line 6, the authors again state the resolution of the along-track data to be 65 km, but as I have noted above, I believe this is a global

average number. So they should refer to the cutoff as less than about 65 km, rather than a rigid < 65 km. In line 8, I can't figure out where the 300 km end of the range 300-65 km comes from.

**Minor Comments (mostly editorial in nature)**

- 1. The word "restitution" on page 1, line 26 and on page 4, line 10 does not seem to be the correct word. My dictionary defines "restitution" as "the restoration of something to its original state." I think "representation" is a better choice.
- 2. The word "homogeneous" is used frequently throughout the paper. While I believe I know what the authors have in mind by this, it should be defined unambiguously the first time this description is used. In most instances, I think the word "consistency" is better than "homogeneous".
- 3. Page 2, line 26: The phrase "large panel of ocean signals" is unclear.
- 4. Page 3, line 2: Chelton et al. (2011) is missing from the list of references.
- 5. Page 3, line 7: The phrase "large panel the AVISO's users" is unclear.
- 6. Page 3, line 16: I think the authors mean "derived products" rather than "derivated products". My dictionary doesn't include the word "derivated". This shows up also on page 4, line 26, and on page 24, line 19.
- 7. Page 4, line 12: Here and at various places later in the manuscript, "ERA Interim" should be replaced with "ERA Interim Reanalysis".
- 8. Page 5, line 24: The word "homogene" should be "homogeneous". But see minor comment #2 above.
- 9. Page 5, lines 15-16: I am confused about how close to land the various altimeter measurements are retained. Here it is stated that "detection of erroneous measurements was strongly restricted in coastal areas", but I can't find a clear description of what these strong restrictions are. On line 17, it is stated that non-repeat data are omitted within 20 km of land. But then on page 7, line 14, it is stated that the proximity to can range anywhere from 0 to 15 km. I guess that this is for exact repeat data. What are the details on how it is decided whether an along-track measurement near land is erroneous?
- 10. Page 8, line 4: "Keep one point over two" is awkward wording. I think the authors mean "keep every second point along track".

- 11. Page 12, line 21: The use of the word "traduce" here and in other places in the manuscript is not correct. I had never seen this word before. My dictionary defines it as "to speak badly of or tell lies about". I don't think this is what the authors have in mind, but I'm not quite sure what word to substitute. Perhaps "introduces" works, at least in some cases.
- 12. Page 13, line 10: What reference level was used for the dynamic height calculations?
- 13. Page 13, line 24: As noted above from Page 11, lines 11-12, I believe the 65 km is a global average, rather than a fixed value everywhere.
- 14. Page 14, lines 1-2: Define "short wavelength signal".
- 15. Page 15, line 11: I'm not sure what "twice more important" means here. I think the authors mean to say "twice as energetic". Same comment on page 15, line 26 where it says "EKE is more important".
- 16. Page 15, line 30: The phrase "can represent below 80%" is awkward. I think this should be "can represent less than 80%".
- 17. Page 16, line 23: The along-track data are not entirely independent of the gridded SSH fields since the mapping procedure is based on spatial and temporal smoothing of the along-track data from the various altimeters.
- 18. Page 18, line 5: Change "meridian" to "meridional".
- 19. Page 18, lines 25-26: I can't figure out what is intended by the phrase "a thinner data selection".
- 20. Page 19, line 31: Define GIA.
- 21. Page 20, lines 15 and 16: The use of the word "regional MSL trend" and "at such hemispheric scales" seems contradictory. Regional usually applies to much smaller than hemispheric.
- 22. Page 20, lines 23-29: The word "underline" appears three times here and in many other places in the manuscript. This is not the correct word, but I don't think a single word can capture every instance of "underline". In some places, "emphasize" can work. In other places, "underscore" can work.
- 23. Page 21, line 4: The acronym LWE is defined on page 6, line 16, but I had long forgotten its definition by the time I got to page 21. Since it only appears three times in the whole manuscript, two of them on page 21, maybe it doesn't need an acronym

that will require readers to go back 15 pages to find the definition.

- 24. Page 21, line 13: The phrase "was quite two times more stronger" is unclear. I think the authors mean "was not quite two times stronger".
- 25. Page 21, line 18: Change "the last altimeter standards" to "the most up-to-date altimeter standards". The standards used in this paper are most assuredly not the last standards that will ever be established.
- 26. Page 22, line 1: Describing the dataset as more energetic is not correct. It is the SSH variability in the DT2014 dataset that is more energetic than in the DT2010 dataset.
- 27. Page 22, line 16: The phrase "is quite 10 times less important than" is unclear. I'm not sure what the authors mean. Perhaps "is not quite 10 times as important as"?
- 28. Page 23, line 26: I believe that "between eastern and western basin" should be changed to "between eastern and western hemispheres."
- 29. Page 24, line 9: I think that "liked" should be "linked".

---

## Author Comment (AC1) · 18 Apr 2016

Referee: This paper describes the latest revision (DUACS DT2014) of the processing system behind the very popular satellite altimeter-based gridded sea level product - known to many researchers as the 'AVISO sea level data'. Indeed, this gridded altimetry product is frequently used in preference to the along-track source data, so many users have lost sight of the many issues associated with the processing of the along track data to the SLA stage, or the multimission gridding process. In this context, the release of a new version of this data set is a very significant announcement, and is an opportunity to inform and educate the users. The paper describing the production and assessment of the data set, and how and why it differs from its predecessor, is bound to be heavily cited. In my opinion, however, the present draft of this paper is not yet satisfactory for this heavy responsibility.

Authors: We acknowledge Rev. #1 for his/her review. All the comments and remarks have been considered and we think that they have contributed to the improvement of the manuscript. In the next paragraphs we present the reviewer's comments followed by our point-by-point reply.

————

Referee: I think this paper needs to be extensively re-written. It is very light on specifics, but still manages to be 24pp of text, which takes far too long to read for the amount of detail that is presented. The 2nd sentence of the abstract "Numerous and impacting evolutions have been implemented at each step of this new data processing" is indicative of the poor English and imprecise style of the whole paper. Reviewing criterion 11 "is the language fluent and precise?" is certainly not met.

Authors: The authors will ask for an English grammar and spelling correction service.

————

Referee: There are many details of the processing that should be explained, but are not. Indeed, the abstract mentions only "The main one is the use of...". The other 'evolutions' are not listed in the abstract. Review criterion 9 is therefore not met.

Authors: The abstract was changed in order to mention all the other changes directly impacting the SLA grids field quality. The change of the reference period is rather identified as "impacting change for users".

————

Referee: Section 2 on "Data processing" says the "altimeter standards" were chosen by a rigourous selection process (described by Ablain et al 2015). Altimetry specialists will know what is meant by this but a little more explanation would help the target audience of this paper, so that they can understand the cause of the differences between the resulting product and its predecessor (which will have impacts on analyses and interpretations of the older dataset). For example, on p23 line 26 there is mention of

the east-west dipole error that has now disappeared because of the new 'standards'. This needs to be much better explained.

Authors: We do agree with you comment and we have improved the paper describing in more details the impact of new altimeter standards.

————

Referee: Page 8 discusses the changes made to the 'multimission mapping' or gridding process. A critical change is a 'better defined correlation scale' which needs a more quantitative explanation. Conversely, the change to a finer spatial (0.25 deg) and temporal (1 day instead of 7 day) sampling could be much more compactly described.

Authors: Additional information about the new correlation scales was added in the text. The authors however considered that the description of the change of the spatial & temporal grid sampling is also important. Some users indeed asked details about this change via AVISO services. The description of this change was moved in a specific annex.

————

Referee: Page 13 says that dynamic height anomalies from T/S profiles are compared with the 'equivalent' field from altimetry. There should be mention here that the third term in this equation is the eustatic change of sea level.

Authors: The physical content of DHA and SLA are indeed not directly comparable. Mass contribution deduced from GRACE measurement need to be added to the DHA for absolute comparison with altimetry field. In this paper, we however use the T/S profiles for a relative comparison, considering two different SLA datasets. In that case, DHA may be sufficient to detect the differences between the two SLA datasets considered

The section was completed as follow: "Quality controlled Temperature/Salinity (T/S) profiles from CORIOLIS Global Data Assembly Center were used. The T/S profiles

processing used in this paper is the same as described by Valladeau et al (2012) and Legeais et al (2016). The Dynamic Height Anomalies (DHA) deduced from T/S profiles (reference depth 900dbar) are compared to the SLA field from gridded "all-sat-merged" products. As discussed in Legeais et al (2016), the DHA are representative of the steric effect above the reference depth, while SLA is representative of both barotropic and baroclinic effects affecting the entire water column. In spite of this difference of physical content, the relative comparison between altimeter SLA and in-situ DHA is sufficient to detect differences between two SLA altimeter products."

————

Referee: The results of the comparisons of the new data set with its predecessor, and with independent data (tide gauges, drifters) are an interesting part of the paper, but the description of these results needs to be much clearer and informative. For a start, there is too much use of the word 'products' instead of geophysical quantities. See the caption to Fig. 5, for example. Wording in many places needs to be much clearer, for example page 17 line 25 refers to the 'DT2014 currents intensity' and then the 'variability'. My guess is that the 'intensity' is a measure of the time-mean, but I am not sure.

Authors: The term "product" was replaced in various place of the manuscript by the physical field considered ("SLA" in the major part of the cases). Additionally, the legends of the figure were clarified (see also specific comment on figure captions). The terms "current intensity" in page 17 was replaced by the appropriate term "current speed". In the same way, "variability" was replaced by "rms of the zonal and meridional components of the current".

————

Referee: Some qualitative conclusions could also be better justified. The page 19 line 25 statement "thus reinforce our confidence in these good results" came as a surprise to me after reading of changes that are near zero, or a mix of positive and negative
values.

Authors: This sentence does not give additional information to the reader. It was removed from the manuscript.

————

Referee: The English of the paper (grammar, vocabulary, spelling, phrasing) certainly needs to be improved throughout (as well as the imprecise style). In some places the poor English is understandable "..error was quite two time more stronger.." but in many places it is not (e.g. "where belong +400cm2/s2 are observed"). I had to consult a dictionary to find out what 'restitute', 'conduce' and 'traduce' all meant. The word 'underline' is idiomatically correct but heavily over-used, as is 'impacts'. "From" is spelled "form" in several places.

Authors: The authors will ask for an English grammar and spelling correction service.

————

Referee: The Figure captions all need to be improved. Example 1: it is ambiguous to describe Fig. 12b as showing the 'variance reduction'. The values are mostly negative. Is a negative reduction a reduction or an increase? It is clearer to say 'change'. Example 2: Fig. 15a shows the "MSL trend difference between DT2014 and DT2010" while 15b shows the "MSL differences..". The M in MSL usually stands for 'mean', sometimes a time-mean, but sometimes a spatial mean. In 15a it can be neither, while in 15b it must be the spatial mean.

Authors: The figure captions were modified in order to avoid possible ambiguity.

———————————————————

---

## Author Comment (AC2) · 18 Apr 2016

Referee : This paper is a very valuable contribution to the altimeter data user community and is likely to be heavily cited because of its comprehensive summary of the improvements made to the latest version of the DUACS altimeter datasets (more commonly referred to colloquially as the "AVISO dataset"). In addition to a general overview of all of the changes that were made between the old altimeter dataset (named DT2010 in this paper) and the new DT2014 dataset, the two most important contributions of this paper are the documentation of the estimated errors of the gridded products in Sec. 3.2.3, and the extensive validation effort by comparisons with in situ data in Secs. 3.2 and 3.4 (monthly mean tide gauge sea level; dynamic height derived from temperature-salinity profiles; and surface drifter data). While the value of this paper is indisputable, the present version needs a lot of work. I list below two major issues, followed by

numerous specific comments, questions and suggestions and, lastly, a list of minor comments, mostly editorial in nature. My comments are mostly focused on the gridded products, which are what I have used almost exclusively for my own research. My comments that follow are embarrassingly extensive, especially for a journal that posts reviews in their entirety in a Discussion section. The level of detail of my comments reflects the amount of time I've devoted in the past to thinking about these issues with the old DT2010 dataset. This paper is an opportunity to put all of those issues to rest once and for all for any future researchers using the DT2014 altimeter dataset.

Authors: We warmly acknowledge Rev. #2 for his detailed review. All the comments and remarks have been considered and we think that they have contributed to the improvement of the manuscript. In the next paragraphs we present the reviewer's comments followed by our point-by-point reply.

———

Major Comments

———

1. Referee : Users have long been frustrated that it is so difficult to track down the details of the OI procedure used by DUACS to generate the gridded SSH fields. The complicated paper trail that users must claw their way through to understand all of these details is evident from lines 10-12 on page 2 of this manuscript in which the authors list a sequence of 7 publications (to which I would add Le Traon and Ogor, 1998) that describe the evolution of the DUACS system and associated products. My own frustration with this is summarized near the beginning of Appendix A2 of Chelton et al (2011, Progress in Oceanography) where it is stated that [. . .] The details of this procedure have evolved somewhat over the years and an upto- date and comprehensive documentation of the procedure is not available. Many of the details are described in a sequence of published papers: Le Traon et al. (1995, 1998, 2003), Le Traon and Ogor (1998), and Ducet et al. (2000). [. . .] we summarize here our understanding of the

procedure as it has most recently been implemented. In addition to the above publications, this summary is based on personal communication in December 2010 with G. Dibarboure at CLS in Toulouse, France who presently oversees the AVISO processing. Rather than include the subsequent description of all of the details, I would have much preferred to point to a single publication in which users would find all of the details in a single place. Not including the details in this paper would perpetuate the difficulty for users to understand how the parameters of the OI procedure impact the interpretation of the data. One consequence of this is that many (probably even most) users believe that the new DT2014 dataset resolves features with spatial scales of $\frac{1}{4}°\frac{1}{4}°$ and temporal scales of 1 day. This is, of course, not even vaguely close to the true spatial and temporal scales of the features that can be resolved. But without easy access to the specifications of the spatial and temporal autocorrelation functions used in the OI analysis, it is understandable that users will make such gross misinterpretation of the resolution capabilities of the DUACS/AVISO dataset. This paper is an opportunity to provide users with "one-stop shopping" for everything they need to know about the dataset.

Authors: The details of the DUACS DT2014 processing are given in Sect. 2, from Level2 input data acquisition, to Level4 SLA products and derivate delivery. The section 2.2.1 was completed with a description of the correlation scales and observation errors used in the OI processing (see "Multi-mission mapping" part), the clarification of the different along-track filtering applied (see specific comment # 3) and the methodology used for SLA derived products generation (see comment specific # 10). Additionally, an appendix B was added in order to summarize the OI mapping methodology.

———

2. Referee : The English grammar and spelling are in need of extensive work by a copy editor. There are also a lot of instances where the authors used incorrect words. These English problems are understandable since English is not the native language of any of the authors. And their English is infinitely better than my French. But the extent of

the problem is quite extreme. I did not carry out a careful statistical analysis, but I have a sense that more than half of the sentences have an English problem of one sort or another, albeit many times only a minor problem. Much of the time, the reader can guess what the authors are trying to say. But there are quite a few cases where I just couldn't figure it out. I draw attention to some of those in the Minor Comments below.

Authors: The authors will ask for an English grammar and spelling correction service.

———

Specific Comments

———

1. Referee : Page 1, lines 14-15: It is stated here and in several other places in the paper (e.g., page 3, lines 12-13, and somewhat less forcefully on Page 21, lines 23-25) that the main improvement in the DT2014 dataset compared with DT2010 is in the use of a 20-year reference period rather than the 7-year reference period used previously. I think this is actually a relatively minor point. Indeed, the authors explain how to change the reference period in a very short appendix at the end of the paper. The main improvements in the DT2014 dataset are the reduced smoothing of the along-track data, improved orbit accuracy, improved ionospheric and atmospheric corrections and the refined scales of the correlation functions used in the OI procedure (the details of which the authors relegate to a reference to Dibarboure et al., 2011, instead of including them in this paper).

Authors: The authors do not fully agree with the reviewer remark that considers the change of reference period as minor point.

Indeed, the change of the reference period does not directly contribute to make the SLA more accurate, but at the opposite of all the other changes, it can be strongly impacting for some users/applications. The use of a new reference period means for some users that they need to change at least the way they interpret the altimeter SLA

signal, and in some case, the field they are susceptible to combine with SLA for their applications. An example is the SLA assimilation into numerical models. Many users combine SLA with their own MDT field. They need to change their MDT (or the SLA) in order to be consistent in terms of reference period.

AVISO users feedback clearly revealed that a large number of users were not aware of the need to have a consistency between the SLA and the mean sea surface (MDT and MSS) in terms of reference period. As said by the reviewer, this paper is the opportunity to educate users and we think that it is important to highlight this reference period change, not only with a specific appendix.

Nevertheless, the authors agree with the fact that all the other changes are not highlighted as they should be since they directly contribute to improve the DT2014 products quality. The list on these changes was added in the abstract as "improving quality change". The change of the reference period is rather identified as "impacting change for users". Sect. 2 was also changed in consistency with the major comment 1).

———

2. Referee : Page 6, lines 13-14: "Reference mission" is defined on lines 2-4 on this page, but "secondary mission" is not clearly defined anywhere. The informed reader will know that the sequence of secondary missions is ERS-1, ERS-2, ENVISAT, Cryosat, AltiKA and HY-2A, but this should be spelled out here to avoid ambiguity.

Authors: The sentence was changed in order to specify the "secondary missions" term and clearly list the corresponding missions. The full spelling and corresponding acronyms of the altimeter missions is given in the second paragraph of the introduction.

———

3. Referee : Page 8, lines 2-3: I'm not certain I understand what is meant by "The wavelengths ranging nearly 200 to 65 km are filtered". I think that the short end of this may be imposed by the along-track smoothing applied to the track data. But from

page 11, lines 11-12, I think that 65 km is a global average for that, rather than a fixed number everywhere. And furthermore, the wavelengths shorter than 65 km are also attenuated, so there should not be a lower range of 65 km for attenuation by filtering. Is the 200 km upper end of the range something that is imposed explicitly in the OI procedure? In any case, the wording is awkward. Perhaps say something like "SSH variability with wavelengths shorter than about 200 km is attenuated."

Authors: Two different along-track filtering are applied:

1) One for along-track SLA, delivered as along-track products. In that case, the aims of the filtering is to reduced the measurement noise error signature, keeping as safe as possible the dynamical signal

2) The other filtering is applied on along-track SLA that are then used in the mapping (note that these filtered along-track products are not delivered to the users). In that case, the objective is also to reduce the signature of the small scale signals that cannot be correctly retrieved on map product, mainly due to the limits of the altimeter constellation sampling capabilities.

This point was not clearly explained in the manuscript. The Sect. "2.2.3 L3 Along-track noise filtering" was included in the section "2.2.1 Overview of the DUACS DT processing; Along track noise filtering". Additionally, the description of the filtering applied in view of the mapping processed was moved from section "2.2.1 Overview of the DUACS DT processing; Along track noise filtering" to "2.2.1 Overview of the DUACS DT processing; Multi-mission mapping".

The remark of the reviewer refers to the filtering described in 2). The 200 km upper end is a trade-off between the physical scales of interest and the multi-mission sampling capability (along- and across-track). The sentence describing this filtering was changed by : "in view of the mapping process, the along-track SLA are low-pass filtered applying a cut-off wavelength that varies with latitude in order to attenuate SLA variability with wavelengths shorter than nearly 200km near the equator, to nearly 65km for latitudes

higher than 40°"

———

4. Referee : Page 8, lines 8-15: This is the place to include the specifics of the parameters for the OI mapping procedure. In my opinion, it is not enough to simply say that the parameters used previously for DT2010 were refined for DT2014 and then send the readers to Dibarboure et al. (2011) to find those details.

Authors: Done. (See major comment #1)

———

5. Referee : Page 8, lines 16-18: I would follow this first sentence of this paragraph with a qualifying sentence like, "But note that the time scales of the variability that are resolved in the DT2014 dataset are not substantially different from DT2010; these time scales are imposed by the temporal correlation function used in the OI mapping procedure."

Authors: We thank the reviewer for this suggestion that will avoid misunderstanding for many users. The sentence was introduced in the paper as well as some information about temporal correlation scales (see major comment #1).

———

6. Referee : Page 8, lines 21-22: Change "an improved resolution" to "a higher grid resolution" so that readers don't draw the incorrect conclusion that the smaller grid spacing using essentially the same OI correlation parameters somehow magically resolves smallerscale variability.

Authors: done. (Note that this part of the manuscript was moved on an Appendix C)

———

7. Referee : Page 8, line 22: Prior to the sentence that begins "These latitudes include the main part. . .", insert a qualifying sentence like "Note, however, that the spatial scales of the features that are resolved in the DT2014 fields are about the same (perhaps slightly smaller) than in the DT2010 fields; these spatial scales are imposed by the spatial correlation function used in the OI mapping procedure."

Authors: These message was introduced in the text, after the sentence that begins "These latitudes include the main part. . .".

———

8. Referee : Page 8, lines 25-26: I'm not sure I agree with the statement that the new gridding "reduces the capability of the gridded products to accurately represent the mesoscale signal in high latitude areas." The $\frac{1}{4}°\frac{1}{4}°$is much finer than the spatial correlation scales used in the OI mapping procedure. I therefore do not expect there to be much difference in the feature resolution capability of the new $\frac{1}{4}°\frac{1}{4}°$gridded product compared with the old 1/3°1/3°Mercator gridded product.

Authors: We do agree with the reviewer. This part of the sentence was removed.

———

9. Referee : Page 8, line 29: I am wondering what happens at the highest latitudes in the DT2014 dataset at times when data are not available "from the recent altimeters like C2". Are the gridded values flagged as missing, or are they filled with some other value such as the mean?

Authors: The grid is defined up to the pole (i.e. highest latitude value = 89.875°N) all along the DT2014 time serie. When no valid measurement are available in the spatial&temporal selection bubble, the grid point is filled with a clearly identifiable default value as for continent (defined in the netcdf _fill_value attribute of the variable considered). Before C2, only ERS-1/2 and EN allowed a sea surface sampling in high latitudes. They are able the sample the surface up to 82°N. This means that during the important sea ice melt events, they eventually allowed an SLA interpolation up to

~83°N.

————

10. Referee : Page 9, line 8: After this first paragraph summarizing the processing for SSH, it would be helpful to include a summary of the details of the calculation of geostrophic velocity. For example, it is my understanding that the DT2014 processing used a wider stencil as suggested by Arbic et al. (2012) to compute the SSH derivatives. Arbic et al. (2012) is included in the reference list, but I could not find it cited anywhere in the paper. The authors apparently intended to discuss the changes in the derivative calculations but either forgot to include that text or inadvertently deleted it from an earlier draft of the paper.

Authors: A section "Derived products generation" summarizing the details of the calculation of geostrophic velocities was indeed present in a first draft of the paper. It was reintroduced in the reviewed version.

However, the authors finally decided to focus the paper on SLA field validation. This point was not clearly specified in the first version of the paper delivered. This was corrected (Abstract, introduction and summary). As a consequences, the geostrophic current considered in the comparison between DT2014 and DT2014 were calculated using the same methodology (here centered differences), as specified in section 2.3.1 "Altimeter gridded products intercomparison" and reminded in the new "Derived products generation" section.

————

11. Referee : Page 9, line 12: The phrase "The mesoscale signal is indeed better reconstructed..." in the discussion of the all-sat-merged product compared with the two-sat-merged product is ambiguous. The meaning of "better" should be defined. In lines 20-21 on this same page, it is stated that the OI parameters are the same for both products. The all-sat product therefore doesn't have any better space-time resolution

than the twosat product. So "better" evidently means "more accurate". This is an important point because virtually all users believe that the all-sat product has higher spatial resolution, which is not the case since it uses the same correlation parameters in the OI mapping procedure. But the all-sat product is presumably more accurate. Authors: The term "more accurately" was used as recommended by the reviewer

————

12. Referee : Page 11, lines 5-7: Defining the resolution of the along-track data as the point where the spectra of signal and noise intersect is too liberal. With a signal-to-noise ratio of 1, it is impossible to distinguish between signal and noise. In my experience, based on visual assessment of various fields with simulated noise added, a more reasonable choice is a S/N variance ratio of 10, which corresponds to a signal standard deviation that is about 3 times larger than the noise standard deviation. At a minimum, a S/N standard deviation ratio of 2 is needed to distinguish signal from noise (i.e., a S/N variance ratio of 4).

Authors: We agree with this comment and the revised manuscript tries to use a wording that is slightly more accurate.

The paragraph was reworded for clarity.

1)The description of the noise reduction by filtering was moved in Section 2.2.1 (see major comment #3)

2)The objectives of this filtering were better explained: removing noise preserving as much as possible the physical signal

3)The synthesis of the results fully described in the Dufau et al (2016) manuscript were simplified (see comment #13)

4)The wavelength obtained by the signal-to-noise ratio of 1 was defined as "the minimum wavelength associated to the dynamical structures that altimetry would statistically be able to observe with a signal-to noise ratio greater than 1". A sentence was

added in order to aware the reader that this is not a perfect noise removal since in practice a signal-to-noise ratio of 2 to 10 (cut-off with a wavelength of 100-150 km or more) would required to get a noise-free topography.

We illustrate the rationale for choosing this cut-off value in Figure A. We simulated proxies of SSH following a simple K-2 law (0.99 cm RMS) and added white noise (1.22 cm RMS) to get noisy SSH samples (1.58 cm RMS). The noise floor was set so that both PSD intersect at 10 km exactly.

Following our SLA filtering method, we used as 10 km cut-off lanczos filter (green curve). As a result the smoothed topography has RMS of 0.99 cm and we removed 1.22 cm RMS of high-wavenumber signals. As expected, the filter is not perfect: we destroyed a small fraction of signal (yellow zone, 0.16 cm RMS) and we left a small fraction of the noise (red zone, 0.12 cm RMS). Figure B shows on an arbitrary sample that the filter is not perfect but generally consistent with the truth. This is why we consider this point as the smallest cut-off frequency if one wants to preserve as much signal as possible.

In contrast, we also used a 30 km cut-off filter (SNR = 10) to illustrate the difference. Figure A shows that we better smoothed out noise (red region = 0.12 cm RMS). Unfortunately we also smoothed the true signal up to 50 km (pink region, 0.3 cm RMS). The net result is therefore negative (hence 0.93 cm RMS after smoothing). Figure B shows that we did lost many small scale features that were partially observed and that could have been used.

The point where both effects (oversmoothing of true signal in pink zone VS. lack of smoothing of noise in red zone) balance out is between SNR=1.2 and SNR=1.3 (here 11-12 km). The 10 km filter (SNR=1) is slightly better if one wants to preserve the signal, and the 11-14 km filter is better is removing noise is more important. In practice, in a sample like Figure B, the difference is very difficult to see. Using a larger cut-off wavelength should be used only if perfect noise removal is the highest priority (read: to

get a very pure signal estimate by sacrificing the higher wavenumbers).

———

13. Referee : Page 11, line 7: It should be clarified that the 55 km mesoscale resolution capability corresponds to wavelength resolution. And the apparent contradiction between 55 km on line 7 and 65 km on line 12 should be clarified.

Authors: Additionally to the points listed in response to comment #12, the paragraph was simplified. The 55km value does not appear anymore. The 65km value is defined as the spatial mean wavelength value for which the signal-to-noise ratio of 1 is observed. As said in the manuscript, "It was defined with 1 year of Jason-2 measurement, over the global ocean excluding latitudes between -20°S and 20°N as in Dufau et al (2016).

———

14. Referee : Page 14, lines 18 and 25: I am very surprised, and indeed skeptical, that changing the gridding from the original 1/3°1/3°to 1/4°1/4°is a bigger factor than the improved processing in determining the increased SSH variance of DT2014 compared with DT2010. I would have thought that the primary explanation for the increased variance in DT2014 is the reduced smoothing applied to the along-track data that are used in the OI procedure. I wonder if the authors can somehow assess the impact of the different along-track smoothing independently of other changes to the gridded products.

Authors: The smoothing applied on the along-track data that are used in the mapping process was indeed slightly reduced between DT2010 and DT2014 version, as explained in the first paragraph of "Multi-mission mapping" chapter of the revised manuscript. The correlation scales were also modified. The cumulative impact of these two changes is a +1.5% increase of the variance. The effect of the along-track filtering change was not assessed separately from the correlation scales change, but the

expected impact would however be limited and mainly visible in the near equatorial latitude band, where the filtering cut-off wavelength change is maximum: ~250km in DT2010 vs ~200km in DT2014.

The authors agree that the grid resolution change from 1/3°x1/3° to 1/4°x1/4° does not impact the variance of the products. However as explained the authors considered the differences between DT2014 product, directly computed on the 1/4°x1/4° grid, and DT2010 "QD" product that the users could download at that time. This "QD" was interpolated from native Mercator 1/3°x1/3° grid as mentioned in sections 2.3.1 and 3.2.1 and this interpolation step explains 3.6% SLA variance increase in DT2014, as explained in the section 3.2.1.

———

15. Referee : Page 15, lines 11-21: I find the discussion in this paragraph to be rather confusing. But in any case, I disagree with the statement on line 14 that wavelength scales of 80- 100 km are "fully observable". With DT2010, SSH is fully resolved only for scales longer than 200 km (Chelton et al., 2011, Progress in Oceanography). It is asserted later (Page 24, line 3) that DT2014 has a wavelength resolution of 150 km (but see comment #21 below). The statement on lines 11-21 that scales of 80-100 km are fully resolved is therefore incorrect.

Authors: The discussion is indeed not clear and confusing mixing along-track and gridded products resolution. It was removed from this section and reported in the discussion section where the limitations of the DT2014 in terms of spatial resolution are discussed. The paragraph was also rephrased in order to clearly distinguish along-track and grid products. The 80-100km refers to along-track products (as discussed in Dufau et al, 2016). The authors refers to Chelton et al (2011; 2014) for the resolution of the gridded products which is estimated to be "nearly 1.7° i.e. slightly less than 200 km at mid latitudes"

———

16. Referee : Page 15, lines 29-30: It is not intuitive that the variance in DT2014 can be 20% smaller than the variance in DT2010 in low variability areas and near the eastern boundaries. It would be good to include some discussion of why this happens.

Authors: The paragraph, which indeed was confusing, has been rephrased clearly pointing at the EKE instead of SLA variance: The EKE deduced from DT2014 is globally higher than the EKE deduced from DT2010 (see Fig 9 and 10 of the reviewed manuscript). This increase in EKE observed with DT2014 reaches 20% (up to 80%) of the DT2010 EKE signal in high variability areas (Eastern boundary currents).

———

17. Referee : Page 16, lines 2-5: Same comment as #14 above for SSH variance: I am very surprised that the finer gridding has a bigger impact on geostrophic current variance than do all of the other changes in the processing.

Authors: As explained in page 14 (see comment #14), the authors chose to intercompare the products delivered with the cartesian $\frac{1}{4}$°x1/4° resolution, i.e. DT2014 vs "QD" DT2010 dataset the users could access. The interpolation process to obtain QD from the Mercator 1/3°x1/3° induces a smoothing leading to an EKE reduction.

To clarify, the sentence "Nearly 10% additional energy is the signature of the direct computation of the SLA on the 1/4°x1/4° Cartesian grid for DT2014 (see Sect. 2.3.1)" was replaced by "Nearly 10% additional energy is the signature of the interpolation of the native DT2010 SLA 1/3°x1/3° Mercator grid on the 1/4°x1/4° Cartesian grid (see Sect. 2.3.1)".

To quantify the effect of the DT2014 filtering, subsampling and correlation parameters, two gridded SLA experimental datasets were specifically constructed over 1 year period:

1) The first one, DT2010 like dataset, was constructed using SSH, filtering, subsampling and correlation parameters corresponding to the DT2010 processing but directly

at the DT2014 grid resolution. Centered difference methodology was used for EKE computation.

2) The DT2014 like dataset was constructed as for 1), but using filtering, subsampling and correlation parameters corresponding to the DT2014 processing

–> The comparison between the two dataset shows that the temporal mean EKE from 2) is slightly less than 6% higher than EKE from 1) over the global ocean, excluding equatorial region ($\pm 5°$N) and high latitudes ($> 60°$).

————

18. Referee : Page 16, lines 27-31: The error estimates are stated as 4.9 cm2 in low variability areas, 32.5 cm2 in high variability areas and 1.4 cm2 in very low-variability areas. These numerical values seem to differ from the numbers given elsewhere in the manuscript. For example, the three numbers that are stated on lines 21-23 of the abstract on page 1. There are perhaps similar inconsistencies elsewhere in the paper.

Authors: The numbers were corrected and now should be consistent in the whole manuscript.

————

19. Referee : Page 18, lines 1-15: It would probably be easier to interpret comparisons of speeds rather than velocity components since the velocity fields are very anisotropic so that analysis of each velocity component separately is difficult to interpret.

Authors: We agree that it is also interesting to analyze the current in terms of speeds. We however prefer to show differences with drifters separating the two components as the. It's of interest notably to assess the maps in each direction.

————

20. Referee : Page 22, lines 5-7: Same comment as #14 and #17 above.

Authors: The sentence was changed in order to point out the DT2010 interpolation processing rather than the DT2014 finer gridding. (see response comment #17)

————

21. Referee : Page 24, lines 3-8: This discussion of the resolution capability of DT2014 versus DT2010 is important but confusing. On line 3, the authors assert that the resolution of DT2014 is 150 km, which can be compared with the 200 km resolution estimated for DT2010 by Chelton et al. (2011, Progress in Oceanography). The authors cite the OSTST presentation by Chelton et al (2014) for evidence of the 150 km resolution for DT2014. If they want to include this level of detail in this paper, then I think they should include a figure showing this result, rather than pointing the readers to an abstract for the OSTST meeting. On line 6, the authors again state the resolution of the along-track data to be 65 km, but as I have noted above, I believe this is a global average number. So they should refer to the cutoff as less than about 65 km, rather than a rigid < 65 km. In line 8, I can't figure out where the 300 km end of the range 300-65 km comes from.

Authors: The reference to the Chelton et al. 2011 was added. The authors chose to keep the Chelton et al OSTST presentation (2014) reference in complement. The link to the slides showing the figure is given in the reference. The discussion chapter was also modified (see also comment #15).

The discussion of the comparison between SLA PDS deduced from along-track and gridded products was simplified. The range 300-65km range initially considered was replaced by the 200-65km range, and the estimation of the energy loss with gridded products was consequently adjusted. The 200km end of the range corresponds to the wavelength for which energy observed from gridded SLA is twice less important as the energy deduced from along-track measurements. This part of the manuscript was changed by: "The comparison with the spectral content deduced from full resolution AL 1Hz along-track measurements (not shown) shows that nearly 60% of the energy observed from along-track measurements on wavelength ranging 200-65km is missing

in the SLA gridded products. In other words, nearly 3/5 of the small-mesoscale variability is missed with DT2014 gridded products. This is clearly linked to the mapping methodology, combined with the altimeter constellation sampling capabilities"

———

Minor Comments (mostly editorial in nature)

———

1. Referee : The word "restitution" on page 1, line 26 and on page 4, line 10 does not seem to be the correct word. My dictionary defines "restitution" as "the restoration of something to its original state." I think "representation" is a better choice.

Authors: corrected

———

2. Referee : The word "homogeneous" is used frequently throughout the paper. While I believe I know what the authors have in mind by this, it should be defined unambiguously the first time this description is used. In most instances, I think the word "consistency" is better than "homogeneous".

———

3. Referee : Page 2, line 26: The phrase "large panel of ocean signals" is unclear.

Authors: The sentence was rephrased as follow : "It upgrades the previous version (called DT2010; Dibarboure et al., 2011) and still pursues the same objectives that first consist in generating time series as homogeneous in terms of altimeter standards and processing with an optimal content at both mesoscales and large scales"

———

4. Referee : Page 3, line 2: Chelton et al. (2011) is missing from the list of references.

Authors: The reference was added

———

5. Referee : Page 3, line 7: The phrase "large panel the AVISO's users" is unclear.

Authors: "large panel of" was replaced by "different"

———

6. Referee : Page 3, line 16: I think the authors mean "derived products" rather than "derivated products". My dictionary doesn't include the word "derivated". This shows up also on page 4, line 26, and on page 24, line 19.

Authors: corrected

———

7. Referee : Page 4, line 12: Here and at various places later in the manuscript, "ERA Interim" should be replaced with "ERA Interim Reanalysis".

Authors: corrected

———

8. Referee : Page 5, line 24: The word "homogene" should be "homogeneous". But see minor comment #2 above.

Authors: corrected

———

9. Referee : Page 5, lines 15-16: I am confused about how close to land the various altimeter measurements are retained. Here it is stated that "detection of erroneous measurements was strongly restricted in coastal areas", but I can't find a clear description of what these strong restrictions are. On line 17, it is stated that non-repeat data are omitted within 20 km of land. But then on page 7, line 14, it is stated that the proximity to can range anywhere from 0 to 15 km. I guess that this is for exact repeat data. What are the details on how it is decided whether an along-track measurement near

land is erroneous?

Authors: The restriction of the data selection described on this section (page 5) only concerns non-repetitive missions. The section was modified in order to:

1) Give a reference giving all the details of the data selection processing

2) Clearly list the missions for which the selection processing was changed in DT2014

As deduced by the reviewer, the description given on page 7 is for repetitive tracks. This part of the manuscript indeed focuses on MPs description. As explained at the beginning of this section ("Along-track SLA generation") section, use of MPs is possible only with repetitive tracks.

———

10. Referee : Page 8, line 4: "Keep one point over two" is awkward wording. I think the authors mean "keep every second point along track".

Authors: Yes. This was corrected

———

11. Referee : Page 12, line 21: The use of the word "traduce" here and in other places in the manuscript is not correct. I had never seen this word before. My dictionary defines it as "to speak badly of or tell lies about". I don't think this is what the authors have in mind, but I'm not quite sure what word to substitute. Perhaps "introduces" works, at least in some cases.

Authors: the word "traduce" was replaced by "express", "show", "is the signature of" according to the situation.

———

12. Referee : Page 13, line 10: What reference level was used for the dynamic height calculations?

Authors: the DHA was referenced to the 900dbar level. This information was added in the text.

———

13. Referee : Page 13, line 24: As noted above from Page 11, lines 11-12, I believe the 65 km is a global average, rather than a fixed value everywhere.

Authors: as explained in Major comment #13, the 65km value is a mean average of the wavelength characterized by a signal-to-noise ratio of 1. This value was retained as the low-pass filter cut-off wavelength applied on along-track data in order to reduce the signature of the 1Hz measurement noise. At this time, the filtering applied on along-track measurements is uniform over the global ocean. This 65km cut-off wavelength is thus constant and unique over the global ocean. (note that we describe in this section the impact of the filtering applied on L3 along-track product disseminated then to the users; as explained in sect 2.2.1 "Along-track SLA generation", it is different from the filtering applied on along-track SLA that are then used in the mapping process (and that are not disseminated to the users)).

———

14. Referee : Page 14, lines 1-2: Define "short wavelength signal".

Authors: done. It corresponds to the wavelength < 65km

———

15. Referee : Page 15, line 11: I'm not sure what "twice more important" means here. I think the authors mean to say "twice as energetic". Same comment on page 15, line 26 where it says "EKE is more important".

Authors: "twice more important" was replaced by "twice as high as"

———

16. Referee : Page 15, line 30: The phrase "can represent below 80%" is awkward. I think this should be "can represent less than 80%".

Authors: the correct formulation is "it can represent up to 80%". It was corrected in the text.

———

17. Referee : Page 16, line 23: The along-track data are not entirely independent of the gridded SSH fields since the mapping procedure is based on spatial and temporal smoothing of the along-track data from the various altimeters.

Authors: by "independent" the authors means "that were not used in the mapping process". This definition is given in section 2.3.2 "Comparison between gridded products and independent along-track measurements"

———

18. Referee : Page 18, line 5: Change "meridian" to "meridional".

Authors: done

———

19. Referee : Page 18, lines 25-26: I can't figure out what is intended by the phrase "a thinner data selection".

Authors: the word "thinner" was replaced by "tuned more finely"

———

20. Referee : Page 19, line 31: Define GIA.

Authors: The acronym was replaced by its definition: glacial isostatic adjustment

———

21. Referee : Page 20, lines 15 and 16: The use of the word "regional MSL trend" and

"at such hemispheric scales" seems contradictory. Regional usually applies to much smaller than hemispheric.

Authors: the sentence was changed by "In order to highlight the improved MSL trend estimation between Eastern and Western hemisphere with DT2014 product, the trend deduced from altimeter products were compared to the trend deduced from in-situ T/S profiles"

———

22. Referee : Page 20, lines 23-29: The word "underline" appears three times here and in many other places in the manuscript. This is not the correct word, but I don't think a single word can capture every instance of "underline". In some places, "emphasize" can work. In other places, "underscore" can work.

Authors: the term "underline" was replaced by the appropriate word (ie. underscore, emphasize, show, present, . . . according to the situation)

———

23. Referee : Page 21, line 4: The acronym LWE is defined on page 6, line 16, but I had long forgotten its definition by the time I got to page 21. Since it only appears three times in the whole manuscript, two of them on page 21, maybe it doesn't need an acronym that will require readers to go back 15 pages to find the definition.

Authors: The LWE acronym was replaced by its definition

———

24. Referee : Page 21, line 13: The phrase "was quite two times more stronger" is unclear. I think the authors mean "was not quite two times stronger".

Authors: the sentence was replaced by "The same comparison done with DT2010 products shows that this residual error was almost twice as high as in the DT2014 version [. . .]"

25. Referee : Page 21, line 18: Change "the last altimeter standards" to "the most up-to-date altimeter standards". The standards used in this paper are most assuredly not the last standards that will ever be established.

Authors: corrected

———

26. Referee : Page 22, line 1: Describing the dataset as more energetic is not correct. It is the SSH variability in the DT2014 dataset that is more energetic than in the DT2010 dataset.

Authors: corrected

———

27. Referee : Page 22, line 16: The phrase "is quite 10 times less important than" is unclear. I'm not sure what the authors mean. Perhaps "is not quite 10 times as important as"?

Authors: the phrase was replaced by "is almost 10 times less important than"

———

28. Referee : Page 23, line 26: I believe that "between eastern and western basin" should be changed to "between eastern and western hemispheres."

Authors: Done

———

29. Referee : Page 24, line 9: I think that "liked" should be "linked".

Authors: corrected

[Figure]

Figure A : K$^{-2}$ spectrum (dashed) is added to white noise (dotted) to simulate noisy data (black, plain). The green curve shows the PSD after a 10 km cut-off lanczos filter is applied. The blue curve shows the PSD after a 30 km cut-off lanczos filter is applied.

**Fig. 1.**

[Figure]

Figure B : K$^{-2}$ spectrum (black dots) is added to white noise to simulate noisy data (grey). The green curve shows the smoothed result after a 10 km cut-off lanczos filter is applied. The blue curve shows the smoothed result after a 30 km cut-off lanczos filter is applied.

**Fig. 2.**

---

## Author Response (AR1)

[revised manuscript text omitted]
 PSD when all-sat-merged (UPD; thick red line) and two-sat-merged (REF; thin bleue line) are considered: zonal (b) and meridionalan (d) components.

[Figure]

Figure 9: Difference of the mean EKE  computed from DT2014 and DT2010  SLA over the [1993, 2012] period. Gridded  SLA merging  all the altimeters available are considered (i.e., "all-sat-merged" in DT2014; "UPD" in DT2010). DT2010  SLA was referenced to the 20-year altimeter reference period and interpolated onto the ¼°x1/4° Cartesian grid for comparison with DT2014. The same methodology (centered differences) was used for geostrophic current computations for  DT2010 and DT2014.

[Figure]

Figure 10: Evolution of the mean EKE over the global ocean (selection of latitudes lower than 60°), computed from the DT2014  (black line) and  DT2010  SLA gridded products referenced to the 20-year period (black dots lines) or to the 7-year period (grey s lines). The same methodology (finite differences) was used for the geostrophic current computation for  DT2010 and DT2014.

[Figure]

Figure 1112: a)  Variance of the differences between gridded DT2014 two-sat-merged  SLA and independent TPN along-track SLA measurements. Statistics are presented for wavelengths ranging from 65-500 km. (unit = cm²). b) Differences  with the results obtained with the DT2010 SLA products.  Negative values indicate a reduction of the differences between gridded and along-track SLA when DT2014 products are considered.

[Figure]

Figure 12: Maps of the difference of the variances of the altimeter geostrophic currents minus drifter measurement differences, using successively DT2014 and DT2010 SLA gridded products. The  difference of variance is expressed in % of the drifter variance. Zonal (a) and meridional (b) component differences. Negative values mean

that the variance of the differences between geostrophic currents deduced from altimetry and from drifter measurement is reduced when considering the DT2014 product.

[Figure]

Figure 13: Difference of the variance of the altimeter SLA minus tide gauge SLA differences, using successively DT2014 and DT2010 SLA gridded products. Monthly TG from PSMSL. Negative values mean that the SLA differences between altimetry and tide gauges is reduced when considering DT2014 products.

[Figure]

Figure 14: a) Map of the differences of the local MSL trend  estimated from the  DT2014 and DT2010 gridded SLA products. MSL estimated over the [1993, 2012]

period. b) Temporal evolution of the differences of the global MSL differences estimated frombetween DT2014 and DT2010.

[Figure]

Figure 1516.: Difference of the mean SLA over the year 2000, measured with TP only, and with the merged TP+ERS-2 product. Comparison done for the DT2010 (a) and DT2014 (b) products.

[Figure]

Figure 4C1 : Left : Difference between two successive grid points on a meridionalaan section as a function of the latitude, atfor a 1/4°x1/4° Cartesian resolution (blue) and 1/3°x1/3° Mercator resolution (red). Right: same as left but for a zonal section.

---

## Referee Report (RR1)

[referee-annotated manuscript omitted]

---

## Referee Report (RR2)

**Review of DUACS DT2014 : the new multi-mission altimeter dataset reprocessed over 20 years by Pujol et al.**

**Reviewer: Graham Quartly**

**General**

As the other reviewers have said before me, the DUACS datasets are very widely used, and a paper giving a complete and clear description of the methodology (data selection, editing, filtering and interpolation) would be very useful. Unfortunately this is by no means as clear as it should be.

**Structure of Paper**

There were a number of references to sections by their title e.g. 'the "Along-track SLA generation" paragraph' (bottom of p.6). If Ocean Sciences permits a 4th level of sub-heading i.e. sect 2.2.1.5 then this should be used; otherwise some re-organization to permit enumeration of these sub-sections using three levels. The situation was made much worse by the present section numbering being awry, which felt like the sections had had a last minute re-ordering without anyone re-reading the manuscript to check for continuity.

There seems to be some repetition of information, which might originate from a late decision to re-order the structure of the paper. The authors should read the paper in its entirety removing duplication, unless it is felt necessary.

There is also a confusing mix of different structures within the paper. Roughly up to p.15 there is description of method intermingled with some results (illustrative figures, percentages of data affected), which works well; subsequently the subsections are all methodology (p. 16-18) followed by separate results (p.19 onwards).

**Writing style**

In general I found the English in this version very understandable, although at times the sentences were too long e.g. "In recent papers .... (L33 and L4)." (p.3, l.13-17) and "Although small wavelengths ... Dussurget et al., 2011)." (p.30, l.2-8).

**Summarising Processing (and changes)**

The article aims to describe fully the DT2014 processing, and to compare its results to an earlier version (DT2010). The changes include new satellite data, new editing, improved corrections and orbits and changes to the interpolation scheme. Although I accept that these are all aspects of the methodology and analysis that should be covered, I did feel that the text felt very long. Some parts felt obvious e.g. a smaller correction scale for the interpolation leads to more short-scale variability being passed and currents determined via centre difference method being larger. Similarly better recovery of data near to the coast will result from a direct OI to chosen grid ('qd') rather than OI to Mercator 1/3° and then simple interpolation to qd. These conclusions seem obvious, so the main merit is in quantifying the difference between DT2010 and DT2014 in these regards. This paper has the potential to become a standard reference for all papers using DT2014, which will no doubt be many. However it needs to

clearly detail what has been done, rather than simply refer to improved editing. The details on the filtering seem quite clear; other aspects contain very little specific information.

i) There is a lot of useful information mentioned in the paragraphs on corrections. Could these be pulled out into a simple text table listing the corrections (wet trop, dry trop, iono, SSB, tides etc.) with specification of which corrections are applied in DT2014 and in DT2010. (Obviously some of these corrections are different for different satellite missions or vary with time; however I feel a tabular form would make the changes between DUACS versions more instantly understandable.

ii) Please provide more details of the editing criteria, rather than simply referring to AVISO/SALP (2015). I believe this is the sort of information users expect from this publication, rather than having all the details available somewhere in an unrefereed publication. Again, if this can be efficiently put in a table, it would be much clearer.

iii) I would like information on how the latitude-dependent biases and long-wavelength biases are removed, and also specifically on what their causes are believed to be e.g. time-tag bias, sea state bias?

iv) How are propagation speeds taken into account (p.12 l.9-12)? Eastward signal are only accepted if a few cm/s: what about Kelvin waves, which are much faster? Is it the case that long wavelength signals do not need such propagation effects to be explicitly included in the interpolation? Does the inclusion of expected values of propagation in the interpolation make it more likely that derived estimates for movement of eddies and Rossby waves will match the pre-conceived expectations, and that data matching anomalous propagation events are suppressed?

v) What is the "tuning of the grid definition near the coast" (p.23 l.27)?

**Summary of Missions**
Fig. 1 is a very familiar image from countless OSTST presentations, but still useful to keep. However, it could be improved with a slight use of colour, with one hue for the "reference missions" and another for those used in the "two-sat" solution, and another for the "tandem" phase. (Personally I always regard the 6-month intercalibration phase as "tandem' and the subsequent one as "interleaved phase", but accept that usage in the altimetry community is confused on this point.) Thus please i) define "tandem" at first use, so your usage is clear, and ii) explain TPN.

**References**
The bibliography list does contain a lot of CLS reports and OSTST presentations. Given that some of these are several years ago, is there not a refereed publication that covers these points? I believe your Fernandes citation should now be as

below: is this correct?  If so, I would have thought that one of the authors of the current paper would have known!!

Fernandes, MJ , C Lazaro,  M Ablain,  and N Pires, Improved wet path delays for all ESA and reference altimetric missions, Remote Sensing of Environment 169, 50-74, doi 10.1016/j.rse.2015.07.023, 2015.

There is also a little inconsistency in the styling of references  e.g. whether 2nd-nth authors should have initial before or after surname, a few have journal name in full whereas it is shortened for most, and some have capitalization for every word in title of paper, whereas most do not; two have the year in the wrong position.  Reference list should have "Marcos", not "Marco".  Finally Ablaiin (2009) and Aviso/DUACS (2014a) do not seem to be cited, and should thus be removed.

**Percentages**
I appreciated the authors' decision to express many things as percentages, since for many it is not clear whether a change of 1.4 cm² for example is large or small. However sometimes it is unclear what is a result or what is an artefact e.g. p.20-21 there is mention of 10% additional energy due to interpolation and +6% due to less filtering.  Do these together constitute the 15% additional EKE?   Is the change in EKE from using 20-year reference instead of 7-year negligible or is this a roughly 0.5-1% reduction (Fig. 10)?

**Specific Questions**
1) The interpolation to a regular grid (qd) takes notice of all observations within a certain distance.  Is there a special adjustment for the isthmus of Panama, or can observations just to the north of Panama affect the gridded SLA just to the south (and vice versa)?
2) Why are ERA-interim data only used up till 2001 (p.5), with ERA Operational thereafter?   There should be some explanation in the text.

**Minor Points**
1) Reviewer 2 pointed out that 'meridian' should be replaced by 'meridional'. This still needs to be done, including in figure labels.
2) Offset between J1 and J2 in "tandem" intercalibration mission is a few minutes not a few hours (p.7 l. 14).
3) Should be "DT2010" on p.7 l.19.
4) Is ERS-1 not used in MP generation (p.8 l.19)?  If not, please add a sentence of explanation.
5) Not "-20°S"!  (p.10 l.3)
6) Is ERS-1 geodetic not included (p.13 l.4)?
7) What is the Lagerloef methodology (p.14 l.5)?   A sentence or two of explanation plus a reference would be useful.
8) Consistency in the expression of timespans would make this easier to read. On p.14 l.23-25, one period is given as "[1993, 1999]", which reads like a pair of references and the other as "1993-2012]".  The latter is much more clearly a timespan than simply 2 dates.  I suggest the latter format is used throughout this paper not just in this section.

9) It is not clear how energy "falls drastically" (p.20 l.13) -- is this with product change, or with wavelength i.e. spectral slope. Fig. 8 would be clearer if x-axis only spanned 30-3000 km.

10) Does 'degradation' (p.23 l.14-15) simply mean an increase in EKE or something else? Possibly change the term 'degradation'.

11) 'which is more prominent' (p.23 l.20)

12) 'They reached' (p.24 l.17).

13) What is meant by the "polar equatorial band" (p.26 l.27)? Is it just an "and" missing?

14) Drop "globally" from "globally ... within the tropics" (p.29 l.1-2).

15) Should 'AL' be 'AltiKa' (p.30 l.15)?

16) 4th letter of CMEMS stands for "Monitoring" (p.31 l.10).

17) Change "approximate range" for "band" (p.34 l.3) -- 41.5 does not seem to be an approximate value!

18) The authors often use "important" when they mean large, and "less important" when "smaller" is intended.

19) Different definitions of low latitude band are used -- ±15° (p.12 l.2) and ±10° (p.23 l.10) -- could these be harmonised?

---

## Author Response (AR2)

**Table of contents**

**1  Detailed answer to review #1**

**Suggestions for revision or reasons for rejection (will be published if the paper is accepted for final publication)**

Review of "DUACS DT2014: the new multi-mission altimeter dataset reprocessed over 20 years" (V2) by M.-I. Pujol, et al. - 2016-05-31.

This second revision of the manuscript, describing the new multi-mission DT2014 altimetry data set, has improved significantly in response to the first round of reviewer comments. As emphasized in those first reviews, the AVISO/DUACS data set is widely used both within and outside of the altimetry community. Therefore, this paper is an important publication that documents the methodology of and improvements in DT2014, particularly when compared to the previous DT2010.

In spite of the improvements between V1 & V2, my biggest complaint is that the article still suffers from structural problems and a lack of focus. Some of the structural problems include: a mixture of referring to previous sections by name, while at other times by number; introducing comparison methodologies in one section, while presenting the comparison itself in another (e.g. "Comparison between gridded products and independent along-track measurements" and later "DT2014 gridded product error estimates at the mesoscale and error reduction compared to DT2010"); the figure numbering is out of order; and the section numbering (as least in the PDF I reviewed) was inconsistent with section numbers 'resetting' several times. The lack of focus comes about due to the overly long article duration as well as the inclusion of too many detailed comparisons. The writing still needs to be streamlined and the points to be made better organized. One specific recommendation I would make is to be more forthcoming about the details of the processing procedures, and spend less time on the minutae of the comparisons. For example, it would be very helpful for users of the gridded product to see maps of the temporal and spatial correlation parameters for the gridding process, as well as the estimated measurement errors and propagation parameters.

This article has a lot of potential, and deserves at least one more significant editing pass to tell a clearer story about how the data set is derived, and how it compares to the earlier version. There is no lack of available figures to present, but in fact the weakness now is that the most thoughtful choice of figures to include hasn't been done. Again, this is an important data set to document, and the article should do it justice. I feel that a major revision is still required to achieve that goal. I've attached a PDF with markup (pop-up notes) to indicate specific edits and comments

**2  Authors:**

We partially and respectfully disagree with this reviewer's comment. The current manuscript
provide extensive details about the processing used, and generally more detailed descriptions
than all previous papers on this topic (Le Traon et al, 1992; 1995; 1998; 1999; 2003; Ducet et
al, 2000; Pujol et al, 2005; Dibarboure et al., 2011). The main purpose of the paper is to
describe, to demonstrate, and to discuss the improvements from the latest DT2014 release, and
not to provide a line-by-line algorithm description with all implementation details and all
parameters.

Still, we tried to take into account the reviewer's comment for the sake of clarity and
concision. To that extent, we have restructured the paper in order to avoid the mixture of
referring to previous sections by name, while at other times by number, and in order to merge
the comparison methodologies in the section presenting the results of the comparison. We
have added various clarifications throughout the manuscript following specific comments of
the reviewer (e.g. table with details of the altimeter standards; precisions in MP computation).
We also tried to compress unnecessary details in the validation chapters (e.g. description and
discussion of the methodology of the comparison between maps and along-track SLA).

Lastly, we have added the new figure 4 which shows the mean zonal and temporal correlation
scales as a function of latitude. This figure is simpler and less confusing than a complex series
detailed high-resolution maps. With this figure, the user has an overview of all critical
parameters and they do not need to be familiar with all the technical computation and implementations details about the high-resolution maps discussed by Ducet et al (e.g. balance between the constellation sampling capability VS observed ocean scales).

Answer to the specific edits and comments are given here after.

**Specific comments:**

Page: 2 : perhaps 'retain' vs. 'present additional'?
Authors: corrected

Page: 2 : remove 'products' here and after L4
Authors: corrected

Page: 2 : You should mention the delay in availability for DT vs. the hours-day for NRT, and separately state that a complete DT reprocessing is done every 4 years. This combination of 'latencies' is confusing.
Authors: done

Page: 3 : You should make this ERS-1 and ERS-2 as done for Jason-1 & Jason-2. The "1/2" is misleading. Also should an 'E1' and 'E2' be indicated or is it all 'EN'?
Authors: corrected

Page: 3 :need to complete the sentence with 'are used'.
Authors: corrected

Page: 3 : change to 'is limited'
Authors: corrected

Page: 3 : change ';' to 'and'
Authors: corrected

Page: 4 : Perhaps move this to section 3 to keep the description of processing on track...
Authors: done

Page: 5 : but nothing about orbits is mentioned for GFO, S/A, C2, HY-2?
Authors: "all" was replaced by "different".
DT2014 GFO and C2 orbit solution is unchanged compared to DT2010. S/A and HY-2 were not in the DT2010 solution so not commented in the text.
As suggested after by the reviewer, the table 1 was introduced in the paper. It gives the details of the orbits used for all the altimeter missions.

Page: 5 : I believe the 'N' in NIC09 stands for 'NOAA'.
Authors: Corrected

Page: 5 :Actually, it should be helpful for all the single-f altimeters: ERS-2, GFO, C2...
Authors: GIM model was rather used for ERS-2, part of EN period, GFO, C2, H2 and AL. This choice was leaded by the validation results obtained during SL_cci project. The NIC09 model indeed induce a degradation of the signal (ERS-2) or has few impact (EN) compared to GIM model solution. The SL_cci validation report is available here for details: www.esa-sealevel-cci.org/webfm_send/178

Page: 5 : It might be helpful to have a side-by-side comparison table showing DT2010 vs. 2014 standards.

Authors: The table 1 listing the standards used for DT2014 was added. We underline in the table the standards that have been changed compared to the DT2010 dataset.

Page: 5 :This isn't really part of the 'processing' and could probably be skipped.

Authors: this part was removed

Page: 6 : again ERS-1 and ERS-2 (E1 3-day ice phase also?)

Authors: corrected. The 3-day phase was not processed in the DT2014.

Page: 6 :This seems incorrect. The geophysical corrections are based on e.g. ECMWF grids which aren't better/worse depending upon the satellite repeat track!

Authors: The authors referred here to the ocean tide model correction. The GOT4v8 model used is an empirical model, based on altimeter repetitive measurements. The errors of this model are thus possibly increased far from a repetitive track position. However, we agree with the reviewer that this is not applicable for all the other geophysical corrections. In order to reduce possible confusion, this part was removed.

Page: 7 :The separation was only minutes on the same track for J1/J2 (and now J2/J3), not hours...

Authors: corrected

Page: 7 :What is the time difference used? +/- 5 days for all missions vs. TP/J1/J2?

Authors: the maximal time interval considered at crossover is +/-10 days.

Page: 8 :and SA?

Authors: AL (AltiKa) was added to this list, since it indeed follows the same theoretical track

Page: 10 :Everywhere? So in DT2014 there is no latitude dependence to the along-track noise filtering?

Authors: Yes, the filtering applied on L3 products is uniform and no latitude dependant. It is different from the latitude dependant filtering applied in view of the L4 processing described in the Sect  2.2.6 "Gridded product (L4) generation: multimission mapping"

Page: 10 :This is confusing. Perhaps say 'every other 1-second point, leading to 14 km...

Authors: the second part of the sentence "leading to a nearly 14 km distance between successive points" clears up possible confusion.

Page: 10 :You might want to clarify that up till this point the description has been largely if not entirely level-3. This section moves on to level-4 products.

Authors: The title was changed for "Gridded product (L4) generation: multi-mission mapping"

Page: 11 :A figure to show these length/time scales on a global map would make the description shorter and easier to understand. This is really a key part of the L4 products that needs to be well explained.

Authors: The figure 4 was added. It shows the mean zonal and temporal correlation scales as a function of latitude.

Page: 12 : wet, not dry
Authors: corrected

Page: 13 :What is the order of magnitude of this difference between all-sat & 2-sat?
Authors: The impact of the altimeter constellation was described by Pascual et al, 2006: the impact is mostly visible in high variability areas. In these areas, changing from 4-altimeter to 2-altimeter constellation lead to a reduction of SLA variability between 5 and 10 cm; the EKE rms differences can reach values higher than 400 cm²/s² within the TP/ERS ground-track diamonds.

Page: 13 Subject: :Given the length of the paper already, I'm not sure it's wise to even mention these products. The focus should remain on the SLA.
Authors: This section was added after the first review (see reviewer #2 comments). As this paper will be the reference for the DUACS DT2014 processing (that include derived products generation) for some years, we considered that it is pertinent to add this section.

Page: 14 :reference?
Authors: the reference was added

Page: 14 : The section identifiers from here to the end of the document are wrong, and appear to have reset a few times.
Authors: corrected

Page: 15 :La Niña
Authors: corrected

Page: 15 :current.
Authors: corrected

Page: 15 :These sections on how the comparisons are made should be part of Section 3, and should be streamlined as much as possible.
Authors: the structure of the paper was changed. this section is now merged with the different subsections of the section 3.

Page: 16 :missions?
Authors: yes. corrected

Page: 18 : (L3)
Authors: corrected

Page: 18 :Why is the figure introduced AFTER Figure 5?
Authors: the inconsistency in the order of the figures was corrected.

Page: 18 :Is this referring to sigma-0 blooms etc? Please clarify.

Authors: this refers to inhomogeneities within the altimeter footprint. They can be induced for instance by surface roughness changes or rain cells.
The sentence was changed to clarify this point. The reference Dibarboure et al was also corrected (2014 instead of 2011 reference)

Page: 19 :Again, a figure showing maps of these scales would be welcome!
Authors: see previous comment: Figure 4 added

Page: 22 :are only?
Authors: not taken into account

Page: 23 :internal tidal
Authors: corrected

Page: 23 : is more
Authors: corrected

Page: 23 :This seems to be out of order with Fig 8-12 coming first.
Authors: see previous comment: the figures were reordered

Page: 24 :reach
Authors: corrected

Page: 26 :This makes it sound like ERS-2 COULD have used dual-f mode, which wasn't possible as it was a single-f altimeter. That should be made clear.
Authors: reformulated. ERS-2 is a nomo-frequency mode altimeter

Page: 27 :This makes it sound like the analysis was for lat>60, whereas I think you mean to say those were excluded. Perhaps just add "excluded" after 60N.
Authors: This is a typography error. The sentence was changed.

Page: 29 :And yet it was discussed in the previous section and in the next paragraph? Perhaps simple remove this sentence and the 'However' at the start of
the next one...
Authors: this sentence was removed

Page: 30 :alongtrack?
Authors: Saral/AltiKa. The acronym is defined in introduction. However, following the recommendation of a first review, we replaced the acronym by the full mission name (the last use of the acronym is in section 2).

**Detailed answer to review #2**

**Review of DUACS DT2014 : the new multi---mission altimeter dataset reprocessed over 20 years by Pujol et al.**

**Reviewer: Graham Quartly**

**General**

As the other reviewers have said before me, the DUACS datasets are very widely used, and a paper giving a complete and clear description of the methodology (data selection, editing, filtering and interpolation) would be very useful. Unfortunately this is by no means as clear as it should be.

**Structure of Paper**

There were a number of references to sections by their title e.g. 'the "Along--- track SLA generation" paragraph' (bottom of p.6). If Ocean Sciences permits a 4th level of sub---heading i.e. sect 2.2.1.5 then this should be used; otherwise some re---organization to permit enumeration of these sub---sections using three levels. The situation was made much worse by the present section numbering being awry, which felt like the sections had had a last minute re---ordering without anyone re---reading the manuscript to check for continuity.

Authors: the paper was reorganized with an enumeration of the "DUACS DT processing" subsections. The wrong section numbering was corrected.

There seems to be some repetition of information, which might originate from a late decision to re---order the structure of the paper. The authors should read the paper in its entirety removing duplication, unless it is felt necessary.

Authors: repetition in sections "Mesoscale signals in the along-track (L3) products" "Coastal areas and high latitudes" (respectively Sect. 3.1 and 3.4 of the revised version) were corrected. Others minor repetitions were also corrected.

There is also a confusing mix of different structures within the paper. Roughly up to p.15 there is description of method intermingled with some results (illustrative figures, percentages of data affected), which works well; subsequently the subsections are all methodology (p. 16---18) followed by separate results (p.19 onwars).

Authors: the paper was restructured. Section "DT2014 gridded SLA product validation protocol" (wrongly numbered 1.2 in the previous version) does not longer exist. The content was merged in the section "3. DT2014 products analysis" " (wrongly numbered 2 in the previous version)

**Writing style**

In general I found the English in this version very understandable, although at times the sentences were too long e.g. "In recent papers .... (L33 and L4)." (p.3, l.13---17) and "Although small wavelengths ... Dussurget et al., 2011)." (p.30, l.2--- 8).

Authors: the long sentences were cut in shorter sentences.

**Summarising Processing (and changes)**

The article aims to describe fully the DT2014 processing, and to compare its results to an earlier version (DT2010). The changes include new satellite data, new editing, improved corrections and orbits and changes to the interpolation scheme. Although I accept that these are all aspects of the methodology and analysis that should be covered, I did feel that the text felt very long. Some parts felt obvious e.g. a smaller correction scale for the interpolation leads to more short---scale variability being passed and currents determined via centre difference method being larger. Similarly better recovery of data near to the coast will result from a direct OI to chosen grid ('qd') rather than OI to Mercator $1/3°$ and then simple interpolation to qd. These conclusions seem obvious, so the main merit is in quantifying the difference between DT2010 and DT2014 in these regards. This paper has the potential to become a standard reference for all papers using DT2014, which will no doubt be many. However it needs to clearly detail what has been done, rather than simply refer to improved editing. The details on the filtering seem quite clear; other aspects contain very little specific information.

i) There is a lot of useful information mentioned in the paragraphs on corrections. Could these be pulled out into a simple text table listing the corrections (wet trop, dry trop, iono, SSB, tides etc.) with specification of which corrections are applied in DT2014 and in DT2010. (Obviously some of these corrections are different for different satellite missions or vary with time; however I feel a tabular form would make the changes between DUACS versions more instantly understandable.

Authors: The details of the standards used in DT2014 products are described in the table 1. As it is not the focus of this paper we did not describe the standards used in DT2010. However, the standards changes compared with the DT2010 are underlined in the table 1.

ii) Please provide more details of the editing criteria, rather than simply referring to AVISO/SALP (2015). I believe this is the sort of information users expect from this publication, rather than having all the details available somewhere in an unrefereed publication. Again, if this can be efficiently put in a table, it would be much clearer.

Authors: The authors agree on the importance on the editing procedure. However, it is a complex processing and several pages would be necessary to fully describe it for all the missions. As it is not the focus of this paper we prefer not to add such a dedicated chapter. Note that Reviewer 1 also mentioned an "overly long" article duration, so we also have to take into account this aspect. Meanwhile, we added some new reference in the text in order to give more sources of information for the users.

iii) I would like information on how the latitude---dependent biases and long--- wavelength biases are removed, and also specifically on what their causes are believed to be e.g. time---tag bias, sea state bias?
Authors: Latitude dependant biases are estimated using an along-track multi-polynomial adjustment of the SLA differences between the two missions considered. An additional corrective map is applied in order to reduce the large-scale residual regional biases after polynomial correction. This corrective map is estimated using boxed mean and spatial
smoothing in order to extract only large scale regional biases and reduce tracks effects.
The methodology was described with the SL_cci project. (see chapter 3.2.1 of the document
http://www.esa-sealevel-cci.org/webfm_send/246).
The paragraph was modified to introduce this reference.
The long-wavelength biases observed between Topex/Poseidon and Jason-1 or Jason-1 and
Jason-2 are mainly induced by :
− differences in the orbit estimation
− altimeter instrumental biases, impacting the range, ionospheric and sea state
bias estimation
− radiometer instrumental biases, impacting the wet tropospheric correction
estimation
iv) How are propagation speeds taken into account (p.12 l.9---12)? Eastward signal are
only accepted if a few cm/s: what about Kelvin waves, which are much faster? Is it
the case that long wavelength signals do not need such propagation effects to be
explicitly included in the interpolation? Does the inclusion of expected values of
propagation in the interpolation make it more likely that derived estimates for
movement of eddies and Rossby waves will match the pre--- conceived expectations,
and that data matching anomalous propagation events are suppressed?

Authors: The propagation speed are a component of the spatio-temporal correlation scales
definition (see Appendix A for formulation). As said in the paper (Sect 2.2.6 of the revised
version), the correlation scales and the propagation speed are a compromise between the
characteristic of the physical signal and the sampling capabilities of the altimeter
constellation. The DUACS processing is based on a mean approximation of the correlation
scales of the signal.  This is a limitation for the reconstruction of some scales, as small spatial
scales as shown for instance by Escudier et al (2013) and for high frequency signal in some
areas. However, the main limiting factor is not the propagation speed, but rather the special
and/or temporal correlation scales considered.  They do not necessarily lead to the total
suppression of part of the signal in the gridded products, but contribute to smooth (spate and
time) part of the signal.

In the case of the extreme propagating events, they are present in the DUACS gridded
products with a possible attenuation of the propagation speed due to the smoothing induced
by OI processing. An example is given in the following figure. It represents the temporal
evolution over year 1997 of the SLA along a section in the equatorial Pacific. It shows wave
signal crossing the Pacific Ocean in nearly 2-3 months (see for instance between day 50 and
120)

[Figure]

Fig: Temporal evolution over year 1997 of the SLA along the section defined along the
latitude 0.125°N and longitude ranging between 120 and 280°E. (unit cm)

v) What is the "tuning of the grid definition near the coast" (p.23 l.27)?

Authors: the tuning simply consists in an improved definition of the ocean/land areas. SLA is computed for grid points considered over ocean, while default value is defined for grid points considered over land.

**Summary of Missions**

Fig. 1 is a very familiar image from countless OSTST presentations, but still useful to keep. However, it could be improved with a slight use of colour, with one hue for the "reference missions" and another for those used in the "two---sat" solution, and another for the "tandem" phase. (Personally I always regard the 6--- month intercalibration phase as "tandem' and the subsequent one as "interleaved phase", but accept that usage in the altimetry community is confused on this point.) Thus please i) define "tandem" at first use, so your usage is clear, and ii) explain TPN.

Authors: The figure 1 was improved following the recommendation of the referee. "tandem" was replaced by "interleaved" and definition is given in Section 2.2.1 "Acquisition". TPN corresponds to the interleaved orbit of TP. This notation was replaced by "TP interleaved".

**References**

The bibliography list does contain a lot of CLS reports and OSTST presentations. Given that some of these are several years ago, is there not a refereed publication that covers these points? I believe your Fernandes citation should now be as below: is this correct? If so, I would have thought that one of the authors of the current paper would have known!!

Fernandes, MJ , C Lazaro, M Ablain, and N Pires, Improved wet path delays for all ESA and reference altimetric missions, Remote Sensing of Environment 169, 50---74, doi 10.1016/j.rse.2015.07.023, 2015.

There is also a little inconsistency in the styling of references e.g. whether 2nd--- nth authors should have initial before or after surname, a few have journal name in full whereas it is shortened for most, and some have capitalization for every word in title of paper, whereas most do not; two have the year in the wrong position. Reference list should have "Marcos", not "Marco". Finally Ablaiin (2009) and Aviso/DUACS (2014a) do not seem to be cited, and should thus be removed.

Authors: the reference list was corrected. Unfortunately, few referred publications are available instead of the reference to CLS reports and OSTST presentations.

**Percentages**

I appreciated the authors' decision to express many things as percentages, since for many it is not clear whether a change of 1.4 $cm_2$ for example is large or small. However sometimes it is unclear what is a result or what is an artefact e.g. p.20---21 there is mention of 10% additional energy due to interpolation and +6% due to less filtering. Do these together constitute the 15% additional EKE? Is the change in EKE from using 20---year reference instead of 7---year negligible or is this a roughly 0.5---1% reduction (Fig. 10)?

Authors: The methodology used to separate the contribution of the different processing on EKE can indeed induce some artifacts that can explain an imprecision in the EKE budget. The change of the reference period should not affect this estimation since as explained in section "DT2010 and DT2014 gridded products intercomparison methodology" (i.e. Sect 1.2.1 of the previous version), the DT2010 products were referenced to the 20-year period for comparison with DT2014.

The 1% observed can be rather explained by the change of the altimeter standards. They indeed contributes to reduce the SLA variance and EKE in DT2014. As discussed in section 4, its contribution is almost 10 times less important than the effect of the mapping procedure changes. A sentence was added in order to introduce this effect on the EKE budget.

**Specific Questions**

1) The interpolation to a regular grid (qd) takes notice of all observations within a certain distance. Is there a special adjustment for the isthmus of Panama, or can observations just to the north of Panama affect the gridded SLA just to the south (and vice versa)?

Authors: The "land-sea mask" processing, allowing to optimize the data selection near small island, narrow strait and land strip (like Panama), was not used in the DT2014 Global ocean processing. At this time, this processing is activated for regional production only (i.e. Mediterranean Sea and Black Sea). This processing will be activated in the next version of the Global products (expected in 2018).

2) Why are ERA---interim data only used up till 2001 (p.5), with ERA Operational thereafter? There should be some explanation in the text.

Authors: ERA-interim reanalysis is available after 2001. However, as explained in section 2.1 of the revised version and in Carrere et al (2016), the use of the ERA-interim based corrections (dry troposphere and DAC) after 2001 leads to a slight degradation of coastal/mesoscale, explaining why this solution was not used after 2001.

**Minor Points**

1) Reviewer 2 pointed out that 'meridian' should be replaced by 'meridional'. This still needs to be done, including in figure labels.

authors: corrected

2) Offset between J1 and J2 in "tandem" intercalibration mission is a few minutes not a few hours (p.7 l. 14).

authors: corrected

3) Should be "DT2010" on p.7 l.19.

authors: yes. corrected

4) Is ERS---1 not used in MP generation (p.8 l.19)? If not, please add a sentence of explanation.

Authors: ERS-1 35-days repetitive measurements (corresponding to year 1993) are indeed not used in the MP estimation. The quality of the ERS-1 measurements is slightly reduced compared to ERS-2 and Envisat. The temporal period covered by ERS-2 and Envisat was estimated long enough to compute an accurate MP and allowed us to give a priority to the
quality of the measurements rather than the quantity. The explanation was added in the paper.

5) Not "---20˚S"! (p.10 l.3)

authors: corrected

6) Is ERS---1 geodetic not included (p.13 l.4)?

Authors: Additional MSS errors during ERS-1 geodetic phase were not considered. Errors for
this mission are constant during the period of the mission, including repetitive and geodetic
phase.

7) What is the Lagerloef methodology (p.14 l.5)? A sentence or two of explanation
plus a reference would be useful.

Authors:

The Lagerloef methodology is used near the equator were the classic geostrophic formulation
cannot be applied (f parameter close to 0). It consists in using a β plane approximation that
implies the use of a second derivate of the sea surface. A Gaussian weight function is used to
ensure the transition with the classic geostrophic formulation.

The Lagerloef methodology is presented in :

Lagerloef, G.S.E., G.Mitchum, R.Lukas and P.Niiler: Tropical Pacific near-surface currents
estimated from altimeter, wind and drifter data,J. Geophys. Res., 104, 23,313-23,32, 1999

This reference was added in the paper.

8) Consistency in the expression of timespans would make this easier to read. On
p.14 l.23---25, one period is given as "[1993, 1999]", which reads like a pair of
references and the other as "1993---2012]". The latter is much more clearly a
timespan than simply 2 dates. I suggest the latter format is used throughout this paper
not just in this section.

authors: corrected

9) It is not clear how energy "falls drastically" (p.20 l.13) ------ is this with product
change, or with wavelength i.e. spectral slope. Fig. 8 would be clearer if x---axis only
spanned 30---3000 km.

Authors: the X axis of the figure was changed

10) Does 'degradation' (p.23 l.14---15) simply mean an increase in EKE or something
else? Possibly change the term 'degradation'.

Authors: here "degradation" mean increase of the variance of the differences between
altimetry and drifters measurements (geostrophic current comparison). The term was
changed.by "increase of the variance of the differences between altimetry and drifter
measurement" (or simply increase)

11) 'which is more prominent' (p.23 l.20)

authors: corrected

12) 'They reached' (p.24 l.17).

authors: corrected

13) What is meant by the "polar equatorial band" (p.26 l.27)? Is it just an "and" missing?

Authors: This is an error. The "magnetic equator band" are the correct words.

14) Drop "globally" from "globally ... within the tropics" (p.29 l.1---2).

authors: corrected

15) Should 'AL' be 'AltiKa' (p.30 l.15)?

Authors: yes. The acronym is defined in introduction. However, the full mission name was used in this section since the last use of the acronym was in section 2 (recommendation of a first review).

16) 4th letter of CMEMS stands for "Monitoring" (p.31 l.10).

authors: corrected

17) Change "approximate range" for "band" (p.34 l.3) ------ 41.5 does not seem to be an approximate value!

authors: corrected

18) The authors often use "important" when they mean large, and "less important" when "smaller" is intended.

Authors: the term "[less] important" was replaced by large or smaller where this was appropriate.

19) Different definitions of low latitude band are used ------ ±15° (p.12 l.2) and ±10° (p.23 l.10) ------ could these be harmonised?

Authors: Done. ±15° was finally retained

**Revised manuscript with detail of the changes**

[revised manuscript text omitted]